# STATELESS MEAN-FIELD GAMES: A FRAMEWORK FOR INDEPENDENT LEARNING WITH LARGE POPULATIONS

## ABSTRACT

Competitive games played by thousands or even millions of players are omnipresent in the real world, for instance in transportation, communications, or computer networks. However, learning in such large-scale multi-agent settings is known to be challenging due to the so-called "curse of many agents". In order to tackle large population independent learning in a general class of such problems, we formulate and analyze the Stateless Mean-Field Game (SMFG): we show that SMFG is a relevant and powerful special case of certain mean-field game formulations and a generalization of other interaction models. Furthermore, we show that SMFG can model many real-world interactions, and we prove explicit finite sample complexity guarantees with independent learning under different feedback models with repeated play. Theoretically, we contribute techniques from variational inequality (VI) literature to analyze independent learning by showing that SMFG is a VI problem at the infinite agent limit. We formulate learning and exploration algorithms that converge efficiently to approximate Nash equilibria even with finitely many agents. Finally, we validate our theoretical results in numerical examples as well as in the real-world problems of city traffic and network access.

## 1 INTRODUCTION

Multi-agent RL (MARL) has been an active area of research, with a very broad range of successful applications in games such as Chess, Shogi (Silver et al., 2017), Go (Silver et al., 2018), Stratego (Perolat et al., 2022), as well as real-world applications for instance in robotics (Matignon et al., 2007). However, the applications of MARL to games of a much larger scale involving thousands or millions of agents still remain a theoretical and experimental challenge (Wang et al., 2020).

Despite this limitation, competitive games with many players are ubiquitous and typically high-stakes. In many real-world games, extremely large-scale competitive multiagency is the rule rather than the exception: for instance when commuting every morning between cities using infrastructure shared with millions of other commuters, when periodically accessing an internet resource by querying servers used simultaneously by many users, or when sending information over common communication channels. A strong but common and powerful assumption in such games is that of *statelessness*. For instance, one can assume that the capacity of an intercity highway or an internet server is only a function of its (and other highways'/servers') immediate load and not a function of time [1].

With the aim of modeling learning in such games, we propose and analyze the Stateless Mean-Field Game (SMFG). In our SMFG model, $N$ players (drivers, internet users, ...) choose among $K$ actions (highway to take, server to access, ...) yielding an empirical distribution of players over actions (i.e., the load of each action) $\boldsymbol{\mu} \in \Delta_K$, where $\Delta_K$ is the $K$ dimensional probability simplex. The vector of expected payoffs then of each action is a function $\mathbf{F}(\boldsymbol{\mu}) \in [0,1]^K$ of the load over actions. Note that $\mathbf{F}$ allows complex dependencies of the payoff of an action on the occupancy of itself and all other actions as well, hence is a powerful and generic model. Furthermore, we assume players are *selfish*, i.e., without regard to global welfare each agent has the goal of maximizing their expected reward. Hence the natural solution concept is a Nash equilibrium where agents do not have incentive to unilaterally deviate. Finally, in our setting, we are interested in *independent learning (IL) using repeated play*, that is, algorithms where each agent learns from their own (noisy) interactions without

---

[1]Unless the timescale is years/decades, in which long-term degradation effects will become significant.

observing others or a centralized coordinator. IL, despite being theoretically challenging, is natural for such games as centralized control can be an unrealistic assumption for large populations.

From a theoretical perspective, our setting is novel but closely related to certain mean-field game (MFG) formulations in RL literature, and yields non-trivial results for cases where $N$ is large. As we present later, the assumptions we introduce are related to Lasry-Lions conditions in MFG literature (Perrin et al., 2020), although not a strict subclass. Furthermore, our formulation is sufficiently general to be relevant in many real-world problems while admitting finite-time, finite-sample IL guarantees, which is absent in MFG literature to the best of our knowledge. Before we formalize SMFG and present the main theoretical contribution, we first motivate its relevance with three examples.

**Network resource management.** Assume there are $K$ resources (e.g., servers/computational nodes) available on a computer network shared by a large number of (say, $N \gg K$) users. These resources might have varying capabilities and load tolerances, as well as correlations in performance due effects of virtualization. At each time step, each user tries to access a resource and experiences a delay that increases with the number of users trying to access the same node. The expected delays of each server might be a highly nonlinear function of the distribution of users over servers that can not be directly modeled, as many complex interactions in infrastructure usage (power, connectivity, CPU/memory resources) typically exist.

**Repeated commuting with large populations.** Every morning, $N$ commuters from city A to city B choose among $K$ routes to drive to their target (typically $N \gg K$), observing only how long it takes them to commute. The distribution of choices in the population affects how much time each person spends traveling. A simplistic model would be increased waiting times as more people choose the same (or intersecting) routes. However, modern road infrastructure can be very complex (Hoogendoorn & Knoop, 2013) due to non-trivial feedback loops and adaptive systems such as load-dependent traffic lights.

**Multi-player multi-armed bandits with soft collisions.** Multiplayer MAB have already been studied in the special case where collisions (i.e., multiple players choosing the same arm) result in zero returns. In many real-world applications, arms used by multiple players yield diminished (but non-zero) utilities when occupied by multiple players. For instance, in many radio communications applications (Bluetooth, Wi-Fi), common frequencies are automatically used via *time-sharing*, yielding delayed but successful communication when collisions occur. Similarly, when accessing online resources, servers will be able to serve multiple clients albeit with slower response times.

Overall, in this paper we introduce the SMFG, its Nash equilibrium as the intuitive solution concept, and discuss its real-world relevance (Section 2). We theoretically analyze the SMFG at the limit $N \to \infty$ and make connections to variational inequalities in optimization literature (Section 3.1), leading to the formulation and finite-time analysis of *independent learning* in full and bandit feedback cases (Sections 3.2, 3.3). Finally, we experimentally verify our theory in numerical examples and two real-world use cases in city traffic and access to the Tor network (Section 4).

## 1.1 RELATED WORK

This work is situated at the intersection of multiple areas of research, we first present each.

**Mean-field games.** Mean-field games (MFG), originally proposed independently by Lasry & Lions (2007) and Huang et al. (2006), have been an active research area in multi-agent RL literature. MFG is a useful theoretical tool for analyzing a specific class of MARL problems consisting of a large number of players with symmetric (but competitive) interests by formulating the infinite agent limit. Such games have been analyzed under various assumptions, reward models and solution concepts, such as Lasry-Lions games (Perrin et al., 2020; Pérolat et al., 2022), stationary MFG (Anahtarci et al., 2022; Xie et al., 2021; Zaman et al., 2023; Yardim et al., 2022), linear quadratic MFG (Guo et al., 2022; Fu et al., 2019), and MFGs on graphs (Yang et al., 2017; Gu et al., 2021). However, finite-sample guarantees for mean-field games only exist under specific assumptions. For Lasry-Lions games, convergence guarantees exist only in continuous time dynamics with an infinite agent oracle (Perrin et al., 2020) and discrete time iteration complexity guarantees only have been shown in the case rewards admit a concave potential (Geist et al., 2021), while IL guarantees with finite agents in this setting do not exist to the best of our knowledge. For stationary MFGs, many recent works require strong regularization, smoothness, and access to infinite-agent oracles (Anahtarci et al., 2022;

Xie et al., 2021; Zaman et al., 2023; Mao et al., 2022). IL in stationary MFGs has been studied either under a subjective equilibrium solution concept (Yongacoglu et al., 2022) or with strong regularization bias and arbitrarily poor sample complexity (Yardim et al., 2022).

**Multi-player MAB (MMAB).** Our model is related to the MMAB problem. The standard reward model for MMAB is collisions (Anandkumar et al., 2011), where agents receive a reward of 0 if more than one pulls the same arm. Results have focused on the so-called no-communications setting, establishing regret guarantees for the cooperative setting without a centralized coordinator (Avner & Mannor, 2014; Rosenski et al., 2016; Bubeck et al., 2021). In our competitive setting, our primary metric is exploitability or proximity to NE rather than regret as typically employed in the collaborative setting of MMAB (similar to the NE for MMAB result established in (Lugosi & Mehrabian, 2022)).

**Variational inequalities.** Variational inequalities (VI) and algorithms, which we use as theoretical tools, have been an active research in classical (Nemirovski, 2004; Facchinei & Pang, 2003; Nesterov, 2007) and recent (Lin et al., 2022; Kotsalis et al., 2022) optimization literature. A recent survey for methods for solving VIs can be found in (Beznosikov et al., 2023).

**Other work.** A setting that has a similar set of keywords is mean-field bandits Gummadi et al. (2013); Wang & Jia (2021), as mentioned in the survey by (Laurière et al., 2022). However, these models do not analyze a *competitive (Nash)* equilibrium, but rather a population steady-state reached by an infinite population under a prescribed, unknown agent behavior. In these works, optimality or exploitability is not a concern, hence a direct comparison with our work is not possible (differences detailed in appendix, Section A). IL has also been investigated in zero-sum games (Daskalakis et al., 2020; Sayin et al., 2021; Ozdaglar et al., 2021) and potential games (Ding et al., 2022; Heliou et al., 2017). A related literature is population games & evolutionary dynamics (Sandholm, 2015; Quijano et al., 2017), where competitive populations are analyzed with differential equations. While solution concepts overlap, we are interested in IL with repeated play and not the continuous-time dynamic system.

## 1.2 SUMMARY OF CONTRIBUTIONS

From the perspective of MFG literature, we (1) formulate the SMFG, a new class of MFGs in which learning occurs without central coordination and with restricted (bandit) feedback, and (2) propose an algorithm that converges to a solution with finite-sample, finite-agent, IL guarantees under limited feedback models. Compared to MMAB, the fundamental contribution of our model is the incorporation of a large class of reward interactions beyond collision models and analysis for *large* ($N \gg K$) populations of players. SMFG can be also seen as an extension of VIs to the setting where a stochastic operator oracle is absent, and can only be evaluated in aggregate by a population. From a technical point of view, we further present the following summary of our novelty and roadmap:

1. We use techniques from optimization and VIs to analyze independent learning with $N$-agents, proving finite sample bounds with regularized learning. Our analyses are partially inspired by Rakhlin et al. (2011), although significantly diverging from their techniques due to (1) having an arbitrary monotone operator rather than a convex minimization problem, (2) the lack of access to a noisy gradient/operator oracle since we are analyzing IL with $N$-agents rather than stochastic optimization, and (3) restricted (bandit) feedback. As a technical contribution, our work demonstrates that VI and operator theory can be used to understand IL.

2. We analyze the learning algorithm under a bandit feedback model and prove finite sample guarantees. In this case, we propose a probabilistic exploration scheme that enables agents to obtain low-variance estimates of the payoff operator $\mathbf{F}$ in the absence of centralized coordination.

3. We verify our theoretical results with synthetic and real-world experiments, demonstrating the usefulness of the mean-field formulation in real-world problems.

## 2 PROBLEM FORMALIZATION AND MOTIVATION

**Notation.** $\Delta_{\mathcal{A}}$ denotes the probability simplex over a finite set $\mathcal{A}$. For a set $\mathcal{A}$ and a map $\mathbf{F} : \Omega \to \mathbb{R}^{\mathcal{A}}$ defined on $\Omega$, for $\omega \in \Omega$, $a \in \mathcal{A}$, we denote the entry of $\mathbf{F}(\omega) \in \mathbb{R}^{\mathcal{A}}$ corresponding to $a$ as $\mathbf{F}(\omega, a) \in \mathbb{R}$. For a vector $\mathbf{u} \in \mathbb{R}^{\mathcal{A}}$, $\mathbf{u}(a) \in \mathbb{R}$ denotes its entry corresponding to $a \in \mathcal{A}$. $\mathbf{1}_K \in \mathbb{R}^K$ denotes a vector with all entries 1. For $N$-tuple $\mathbf{v} \in \Omega^N$ and $v \in \Omega$, $(v, \mathbf{v}^{-i})$ is the $N$-tuple with the $i$-th entry replaced by $v$. $\mathbf{e}_a \in \mathbb{R}^{\mathcal{A}}$ is the standard unit vector with coordinate $a$ set to 1.

## 2.1 MATHEMATICAL FORMULATION

We formalize the stateless mean-field game (SMFG), the main mathematical object of interest of this work. The SMFG problem with repeated plays consists of the following.

1. Finite set of players $\mathcal{N} = \{1, \ldots, N\}$ with $|\mathcal{N}| = N \in \mathbb{N}_{>0}$,
2. Set of finitely many actions $\mathcal{A}$, with $|\mathcal{A}| = K \in \mathbb{N}_{>0}$,
3. A payoff function $\mathbf{F} : \Delta_{\mathcal{A}} \to [0, 1]^K$, which maps the empirical occupancy measure of the population over actions to a corresponding payoff in $[0, 1]$ for each action.

SMFG is played across multiple rounds where players are allowed to alter their strategies in between observations, where at each round $t \in \mathbb{N}_{\geq 0}$,

1. Each player $i \in \mathcal{N}$ picks an action $a_t^i \in \mathcal{A}$,
2. The empirical occupancy measure over actions is set to be $\widehat{\boldsymbol{\mu}}_t := \frac{1}{N} \sum_{i=1}^{N} \mathbf{e}_{a_t^i}$
3. Depending on the feedback model, each agent $i \in \mathcal{N}$ observes:
   **Full feedback:** The (noisy) payoffs for *each* action $\mathbf{r}_t^i := \mathbf{F}(\widehat{\boldsymbol{\mu}}_t) + \mathbf{n}_t^i \in \mathbb{R}^K$ or,
   **Bandit feedback:** The (noisy) payoff for their chosen action $r_t^i := \mathbf{r}_t^i(a_t^i) \in \mathbb{R}$.
4. Each agent receives the payoff $r_t^i$.

Intuitively, SMFG models games where the payoff obtained from choosing an action depends on how the population is statistically distributed over actions, without distinguishing between particular players. We assume the $K$-dimensional noise vectors $\mathbf{n}_t^i$ are i.i.d. for each $i, t$ and entry-wise have zero mean and variance $\sigma^2$. Hence each agent observes a noisy version of the payoff of their action (or the payoff of all actions with full feedback) under the current empirical occupancy over actions.

We assume the game is competitive, that is, each agent aims to maximize their personal expected payoff without regard to social welfare. We allow agents to play randomized actions (mixed strategies), where each agent $i$ randomly chooses their actions at time $t$ with respect to their mixed strategy $\boldsymbol{\pi}_t^i \in \Delta_{\mathcal{A}}$. The primary solution concept for such a game will be the Nash equilibrium (NE).

**Definition 1 (Expected payoff, exploitability, Nash equilibrium)** *For an $N$-tuple of mixed strategies $(\boldsymbol{\pi}^1, \ldots, \boldsymbol{\pi}^N) \in \Delta_{\mathcal{A}}^N$, we define the expected payoff $V^i$ of an agent $i \in \{1, \ldots, N\}$ as*

$$V^i(\boldsymbol{\pi}^1, \ldots, \boldsymbol{\pi}^N) := \mathbb{E}\left[\mathbf{F}\left(\frac{1}{N} \sum_{j=1}^{N} \mathbf{e}_{a^j}\right)(a^i) \middle| a^j \sim \boldsymbol{\pi}^j, \forall j = 1, \ldots, N\right].$$

*An $N$-tuple $(\boldsymbol{\pi}^1, \ldots, \boldsymbol{\pi}^N) \in \Delta_{\mathcal{A}}^N$ is called a NE if for all $i$, $V^i(\boldsymbol{\pi}^1, \ldots, \boldsymbol{\pi}^N) = \max_{\boldsymbol{\pi} \in \Delta_{\mathcal{A}}} V^i(\boldsymbol{\pi}, \boldsymbol{\pi}^{-i})$. For any $\delta > 0$, we call $(\boldsymbol{\pi}^1, \ldots, \boldsymbol{\pi}^N) \in \Delta_{\mathcal{A}}$ a $\delta$-NE if $V^i(\boldsymbol{\pi}^1, \ldots, \boldsymbol{\pi}^N) \geq \max_{\boldsymbol{\pi} \in \Delta_{\mathcal{A}}} V^i(\boldsymbol{\pi}, \boldsymbol{\pi}^{-i}) - \delta$ for any $i$. We also define the exploitability of agent $i$ for the tuple as $\mathcal{E}_{exp}^i(\{\boldsymbol{\pi}^j\}_{j=1}^N) := \max_{\boldsymbol{\pi}' \in \Delta_{\mathcal{A}}} V^i(\boldsymbol{\pi}', \boldsymbol{\pi}^{-i}) - V^i(\boldsymbol{\pi}^1, \ldots, \boldsymbol{\pi}^N)$.*

Intuitively, under a mixed strategy profile $(\boldsymbol{\pi}^1, \ldots, \boldsymbol{\pi}^N)$ that is a Nash equilibrium, no agent has the incentive to deviate from their mixed strategy as in expectation each agent is playing optimally with respect to the rest. The task of finding a NE is equivalent to finding policies with low exploitability, therefore $\mathcal{E}_{\exp}^i$ is a natural error metric. Note that our definition of exploitability can be seen as the $N$-player version of the mean-field game exploitability defined in literature (Perrin et al., 2020).

**Goal.** With the SMFG problem definition and the solution concept introduced, we state our objective: in both feedback models, we would like to find *sample efficient* algorithms which learn an *approximate NE* (in the sense of low exploitability) *independently* from repeated plays by $N$ agents when $N$ is *large*. To avoid clutter, we rigorously formalize the notion of an algorithm in the different feedback models in the appendices (Section C).

## 2.2 ASSUMPTIONS ON PAYOFFS AND EXAMPLES

Finding an NE in general is challenging, and possibly computationally intractable. We analyze SMFG under certain assumptions on $\mathbf{F}$ that also correspond closely to real-world problems. Our first assumption is the Lipschitz continuity of the payoff. In many applications, action payoffs should not change too much with small variations in population behavior, Definition 2 formalizes this intuition.

**Definition 2 (Lipschitz continuous payoffs)** *The payoff map* $\mathbf{F} : \Delta_{\mathcal{A}} \to [0,1]^K$ *is called Lipschitz continuous with parameter* $L > 0$ *if for any* $\boldsymbol{\mu}, \boldsymbol{\mu}' \in \Delta_{\mathcal{A}}$, $\|\mathbf{F}(\boldsymbol{\mu}) - \mathbf{F}(\boldsymbol{\mu}')\|_2 \leq L\|\boldsymbol{\mu} - \boldsymbol{\mu}'\|_2$.

The next fundamental assumption, monotonicity, is standard in variational inequality literature (Facchinei & Pang, 2003) and is somewhat similar in form to Lasry-Lions conditions in mean-field games literature (Perrin et al., 2020; Pérolat et al., 2022), apart from the dependence on distributions over actions rather than states.

**Definition 3 (Monotone/Strongly monotone payoff)** *The vector map* $\mathbf{F} : \Delta_{\mathcal{A}} \to [0,1]^K$ *is called monotone if for some* $\lambda \geq 0$, *for all* $\forall \boldsymbol{\mu}_1, \boldsymbol{\mu}_2 \in \Delta_{\mathcal{A}}$, *it holds that*

$$(\mathbf{F}(\boldsymbol{\mu}_1) - \mathbf{F}(\boldsymbol{\mu}_2))^\top (\boldsymbol{\mu}_1 - \boldsymbol{\mu}_2) = \sum_{a \in \mathcal{A}} (\boldsymbol{\mu}_1(a) - \boldsymbol{\mu}_2(a))(\mathbf{F}(\boldsymbol{\mu}_1, a) - \mathbf{F}(\boldsymbol{\mu}_2, a)) \leq -\lambda \|\boldsymbol{\mu}_1 - \boldsymbol{\mu}_2\|_2^2.$$

*Furthermore, if the above holds with some* $\lambda > 0$, $\mathbf{F}$ *is called* $\lambda$*-strongly monotone.*

Monotone payoffs, as in the case of Lasry-Lions conditions, can be intuitively thought as modeling problems where payoff of an action on average decreases as the occupancy increases. However, one fundamental difference from existing MFG works is that our monotonicity assumption is on *action occupancy*, and not on *state occupancy* as in (Perrin et al., 2020). While monotonicity is somewhat technical, the following examples indicate that the assumptions coincide with practically relevant cases (explicit proofs deferred to the appendix, Section B).

**Example 1 (Non-increasing payoffs)** *Let* $F_a : [0,1] \to [0,1]$ *for* $a \in \mathcal{A}$ *be Lipschitz continuous and non-increasing functions, define* $\mathbf{F}(\boldsymbol{\mu}) := \sum_a F_a(\boldsymbol{\mu}(a))\mathbf{e}_a$. *Then* $\mathbf{F}$ *is monotone and Lipschitz. If* $F_a$ *is also* strictly *decreasing and there exists* $\lambda > 0$ *such that* $|F_a(x) - F_a(x')| \geq \lambda|x - x'|$ *for all* $x, x' \in [0,1], a \in \mathcal{A}$, *then* $\mathbf{F}$ *is* $\lambda$*-strongly monotone.*

While the above example is expected, the monotone assumption can also model a large class of payoffs with complicated cross-dependencies on the occupations, as the next example shows.

**Example 2** *Let* $\mathcal{A} = \{1, \ldots, K\}$, *and* $\alpha_k > 0, \lambda_{k,k'} \in \mathbb{R}$ *constants for all* $k, k' \in \mathcal{A}, k > k'$. *The payoff given by* $\mathbf{F}(\boldsymbol{\mu}, k) := -\exp\{\alpha_k \boldsymbol{\mu}(k)\}/\sum_{k'} \exp\{\alpha_{k'}\boldsymbol{\mu}(k')\} + \sum_{k'<k} \lambda_{k,k'}\boldsymbol{\mu}(k') - \sum_{k'>k} \lambda_{k',k}\boldsymbol{\mu}(k')$ *for all* $k \in \mathcal{A}$ *is Lipschitz and monotone.*

As a richer class of examples, for any concave potential function $\Phi : \Delta_{\mathcal{A}} \to \mathbb{R}$, the payoff vector $\mathbf{F} = \nabla\Phi$ is monotone (see Example 5 in the appendix). The class of monotone payoffs is also strictly richer than payoffs with potential, and the MMAB problem with collisions can be also formulated as a monotone operator. Hence, one interpretation of our setting is that we allow soft collisions for action payoffs, as actions yield non-zero payoffs when chosen by multiple players. We discuss these further in the appendix (Section B). We also show that monotone payoffs are more general than congestion games (Rosenthal, 1973) and potential games (Monderer & Shapley, 1996).

## 3 MAIN THEORETICAL RESULTS

We present the main theoretical results of this paper, with actual proofs deferred to the appendices.

### 3.1 THEORETICAL TOOL: THE MEAN-FIELD GAME LIMIT $N \to \infty$

In this section, we present the main theoretical machinery of our approach that permits efficient learning: we formulate and introduce the mean-field limit as the limit when the number of players goes to infinity. For this purpose, we introduce the following MF-NE concept.

**Definition 4 (MF-NE)** *A mean-field Nash equilibrium (MF-NE)* $\boldsymbol{\pi}^* \in \Delta_{\mathcal{A}}$ *associated with payoff operator* $\mathbf{F}$ *is a probability distribution over actions such that* $\sum_a \boldsymbol{\pi}^*(a)\mathbf{F}(\boldsymbol{\pi}^*, a) = \max_{\boldsymbol{\pi} \in \Delta_{\mathcal{A}}} \sum_a \boldsymbol{\pi}(a)\mathbf{F}(\boldsymbol{\pi}^*, a)$. *If it holds that* $\sum_a \boldsymbol{\pi}^*(a)\mathbf{F}(\boldsymbol{\pi}^*, a) \geq \max_{\boldsymbol{\pi} \in \Delta_{\mathcal{A}}} \sum_a \boldsymbol{\pi}(a)\mathbf{F}(\boldsymbol{\pi}^*, a) - \delta$ *for some* $\delta > 0$, *we call* $\boldsymbol{\pi}^*$ *a* $\delta$*-MF-NE.*

Intuitively, the mean-field limit simplifies the SMFG problem by assuming each agent follows the same policy and replacing the notion of $N$ independent agents with a single agent playing against a

continuum of infinitely many identical agents. We show in the appendix (Proposition 2) that the price paid for using the MF-NE solution concept scales with $\mathcal{O}(1/\sqrt{N})$ additional exploitability, which will become insignificant in games with large $N$. Furthermore, finding MF-NE is equivalent to solving the following *variational inequality* problem corresponding to the operator $\mathbf{F}$ and domain $\Delta_{\mathcal{A}}$:

$$\text{Find } \boldsymbol{\pi}^* \in \Delta_{\mathcal{A}} \text{ s.t. } \mathbf{F}(\boldsymbol{\pi}^*)^\top (\boldsymbol{\pi}^* - \boldsymbol{\pi}) \geq 0, \forall \boldsymbol{\pi} \in \Delta_{\mathcal{A}}. \tag{MF-VI}$$

This formulation enables the use of theoretical results from VI literature. For instance, the MF-NE always exists for continuous $\mathbf{F}$, and is unique for strongly monotone $\mathbf{F}$ (see Section E.1). Finally, regularizing the MF-VI problem will play a crucial role in the IL setting, as it will prevent the policies of agents from diverging when there is no centralized controller synchronizing the policies of agents. We formulate the related $\tau$-Tikhonov regularized VI problem:

$$\text{Find } \boldsymbol{\pi}^* \in \Delta_{\mathcal{A}} \text{ s.t. } (\mathbf{F} - \tau \mathbf{I})(\boldsymbol{\pi}^*)^\top (\boldsymbol{\pi}^* - \boldsymbol{\pi}) \geq 0, \forall \boldsymbol{\pi} \in \Delta_{\mathcal{A}}. \tag{MF-RVI}$$

Note that $\mathbf{F} - \tau \mathbf{I}$ is strongly monotone if $\mathbf{F}$ is monotone, hence (MF-RVI) always admits a unique solution. In the following sections, we will present IL algorithms in the more concrete finite $N$-player setting and the associated guarantees, using as a theoretical tool.

**Is it possible to simply solve MF-RVI in our setting?** Past works in MFG have already proved approximation results of $N$-agent games by MFG albeit in different settings (e.g. Saldi et al. (2019)), but these results do not consider when *learning itself* is carried out with $N$ agents. If $N$ agents can not communicate, it is theoretically challenging to approximate the MF-RVI and to tackle bandit feedback. Hence, in the next sections, we prove the more challenging result of convergence with IL.

### 3.2 CONVERGENCE IN THE FULL FEEDBACK CASE

We first present an IL algorithm for full feedback. In this setting, while there is no centralized controller, independent noisy reports of all action payoffs are available to each agent after each round. Our analysis builds up on Tikhonov regularized projected ascent (TRPA). The TRPA is defined as

$$\Gamma^{\eta,\tau}(\boldsymbol{\pi}) := \Pi_{\Delta_{\mathcal{A}}}(\boldsymbol{\pi} + \eta(\mathbf{F} - \tau \mathbf{I})(\boldsymbol{\pi})) = \Pi_{\Delta_{\mathcal{A}}}((1 - \eta\tau)\boldsymbol{\pi} + \eta\mathbf{F}(\boldsymbol{\pi})), \tag{TRPA}$$

for a learning rate $\eta > 0$ and regularization $\tau > 0$. Intuitively, $\Gamma^{\eta,\tau}$ uses $\mathbf{F}$ evaluated at $\boldsymbol{\pi}$ to modify action probabilities in the direction of the greatest payoff, incorporating an $\ell_2$ regularizer of $\tau$. The analysis of TRPA is standard and known to converge for monotone $\mathbf{F}$ (Facchinei & Pang, 2003; Nemirovski, 2004), when (stochastic) oracle access to $\mathbf{F}$ is assumed. Naturally, the main complication in applying the method above will be the fact that in the IL setting, agents can not evaluate the operator $\mathbf{F}$ arbitrarily, but rather can only observe (a noisy) estimate of $\mathbf{F}$ as a function of the empirical population distribution and not of their policy $\boldsymbol{\pi}$. Hence in the full feedback setting, we analyze the following update rule:

$$\boldsymbol{\pi}_0^i := \text{Unif}(\mathcal{A}) = \frac{1}{K}\mathbf{1}_K, \quad \boldsymbol{\pi}_{t+1}^i = \Pi_{\Delta_{\mathcal{A}}}((1 - \tau\eta_t)\boldsymbol{\pi}_t^i + \eta_t \mathbf{r}_t^i), \tag{TRPA-Full}$$

for a time varying learning rate $\eta_t$, for each agent $i = 1, \ldots, N$. The extraneous $\ell_2$ regularization incorporated each agent running TRPA-Full is critical for the analysis and convergence in IL, as it allows explicit synchronization of policies of agents without communication.

**Theorem 1 (Convergence, full feedback)** *Assume $\mathbf{F}$ is Lipschitz, monotone. Assume $N$ agents run the TRPA-Full update rule for $T$ time steps with learning rates $\eta_t := \frac{\tau^{-1}}{t+1}$ and regularization $\tau > 0$. Then it holds for any agent $i = 1, \ldots, N$ that $\mathbb{E}\left[\mathcal{E}_{exp}^i(\{\boldsymbol{\pi}_T^j\}_{j=1}^N)\right] \leq \mathcal{O}(\frac{\tau^{-2}}{\sqrt{T}} + \frac{\tau^{-1}}{\sqrt{N}} + \tau)$. Furthermore, if $\mathbf{F}$ is $\lambda$-strongly monotone, then $\mathbb{E}\left[\mathcal{E}_{exp}^i(\{\boldsymbol{\pi}_T^j\}_{j=1}^N)\right] \leq \mathcal{O}(\frac{\tau^{-3/2}\lambda^{-1/2}}{\sqrt{T}} + \frac{\tau^{-1/2}\lambda^{-1/2}}{\sqrt{N}} + \tau)$.*

**Proof:** *(sketch)* We show that under TRPA-Full, (i) the deviations of policies $\|\boldsymbol{\pi}_t^i - \boldsymbol{\pi}_t^j\|_2$ can be bounded, (ii) consequently, the bias in evaluating $\mathbf{F}$ in aggregate as a population is bounded, and (iii) the resulting recursion yields the stated result. See Theorem 3 in the appendix for details. $\square$

We note that in the choice of learning rate $\eta_t$ above, no intrinsic problem parameter is assumed to be known. Furthermore, due to (1) the regularization $\tau$ and (2) a finite population, a non-vanishing exploitability of $\mathcal{O}(\tau + \tau^{-1}/\sqrt{N})$ will be induced in terms of the NE (see Lemma 7) in the monotone

case. Since for finite population SMFG, there will be always be a non-vanishing exploitability in terms of NE due to the mean-field approximation, in practice $\tau$ could be chosen to incorporate an acceptable loss. Alternatively, if the exact value of the number of players $N$ is known by each agent, one could choose $\tau$ optimally, to obtain the following corollary.

**Corollary 1 (Optimal $\tau$, full feedback)** *Assume the conditions of Theorem 1. For monotone $\mathbf{F}$, choosing regularization parameter $\tau = {}^1/\sqrt[4]{N}$ yields $\mathbb{E}\left[\mathcal{E}_{exp}^i(\{\boldsymbol{\pi}_T^j\}_{j=1}^N)\right] \leq \mathcal{O}(\frac{\sqrt{N}}{\sqrt{T}} + \frac{1}{\sqrt[4]{N}})$ for any $i$. For $\lambda$-strongly monotone $\mathbf{F}$, choosing $\tau = {}^1/\sqrt[3]{N}$ yields $\mathbb{E}\left[\mathcal{E}_{exp}^i(\{\boldsymbol{\pi}_T^j\}_{j=1}^N)\right] \leq \mathcal{O}(\frac{\lambda^{-1/2}\sqrt{N}}{\sqrt{T}} + \frac{\lambda^{-1/2}}{\sqrt[3]{N}})$.*

Even though TRPA-Full solves the regularized (hence strongly monotone) problem, compared to the $\mathcal{O}({}^1/T)$ rate in classical strongly monotone VI (Kotsalis et al., 2022) or strongly convex optimization (Rakhlin et al., 2011), our worse $\mathcal{O}({}^1/\sqrt{T})$ time dependence is due to independent learning. Intuitively, additional time is required to ensure the policies of independent learners are sufficiently close when "collectively" evaluating $\mathbf{F}$. The additional dependence of the time-vanishing term on $\sqrt{N}$ is also a result of this fact. Furthermore, when learning itself is performed by $N$ agents, we note that the bias as a function of $N$ decreases with $\mathcal{O}({}^1/\sqrt[4]{N})$ (or $\mathcal{O}({}^1/\sqrt[3]{N})$ for strongly monotone problems), and not with $\mathcal{O}({}^1/\sqrt{N})$ as Proposition 2 might suggest. We leave the question of whether this gap can be improved and whether knowledge of $N$ is required in Corollary 1, as future work.

### 3.3 CONVERGENCE IN THE BANDIT FEEDBACK CASE

We now move on to the more challenging and realistic bandit feedback case, where agents can only observe the payoffs of the actions they have chosen. Once again, we analyze the IL setting (or in bandits terminology, the "no communications" setting) where agents can not interact or coordinate with each other. One of the main challenges of bandit feedback with IL in our setting is that it is difficult for each agent to identify itself (i.e., assign itself a unique number between $1, \ldots, N$) so that exploration of action payoffs can be performed in turns. For instance, in MMAB algorithms, this is typically achieved using variants of the so-called musical chairs algorithm (Lugosi & Mehrabian, 2022), which is not available in our formulation. Instead, we adopt a *probabilistic* exploration scheme where each agent probabilistically decides it is its turn to explore payoffs while the rest of the agents induce the required empirical population distribution on which $\mathbf{F}$ should be evaluated.

Our algorithm, which we call TRPA-Bandit, is straightforward and relies on exploration occurring over epochs, where policies are updated once inbetween epochs using the estimate of action payoffs constructed during the exploration phase. While we formally present TRPA-Bandit in the appendix (Algorithm 1), the procedure informally is as follows for each agent, fixing an exploration parameter $\varepsilon \in (0, 1)$:

1. At each epoch $h$, for $T_h = \lceil \varepsilon^{-1} \log(h + 1) \rceil$ time steps, repeat the following:
    (a) With probability $\varepsilon$, sample uniformly an action $a_{h,t}$, observe the payoff $r_{h,t}$, and keep the importance sampling estimate $\widehat{\mathbf{r}}_h \leftarrow K r_{h,t} \mathbf{e}_{a_{h,t}}$.
    (b) Otherwise (with probability $1 - \varepsilon$), sample action according to current policy $\boldsymbol{\pi}_h$.
2. Update the policy using TRPA, $\boldsymbol{\pi}_{h+1} = \Pi_{\Delta_{\mathcal{A}}}((1 - \tau\eta_h)\boldsymbol{\pi}_h + \eta_h\widehat{\mathbf{r}}_h)$, with learning rate $\eta_h := \frac{\tau^{-1}}{h+1}$. If agent did not explore this epoch, use $\widehat{\mathbf{r}}_h = \mathbf{0}$.

Intuitively, the probabilistic sampling scheme allows some agents to build a low-variance estimate of $\mathbf{F}$, while others simply sample actions with their current policy in order to induce the empirical population distribution at which $\mathbf{F}$ should be evaluated. The result in this setting is as follows.

**Theorem 2 (Convergence, bandit feedback)** *Assume $\mathbf{F}$ is Lipschitz, monotone. Assume $N$ agents run TRPA-Bandit (Algorithm 1) for $T$ time steps with regularization $\tau > 0$ and exploration parameter $\varepsilon > 0$, and agents return policies $\{\boldsymbol{\pi}^i\}_i$ after executing Algorithm 1. Then, for any agent $i = 1, \ldots, N$ that $\mathbb{E}\left[\mathcal{E}_{exp}^i(\{\boldsymbol{\pi}^j\}_{j=1}^N)\right] \leq \widetilde{\mathcal{O}}(\frac{\tau^{-2}\varepsilon^{-1/2}}{\sqrt{T}} + \tau^{-1}\varepsilon + \tau + \frac{\tau^{-1}}{\sqrt{N}} + \frac{\tau^{-3/2}}{N})$. If $\mathbf{F}$ is $\lambda$-strongly monotone, then $\mathbb{E}\left[\mathcal{E}_{exp}^i(\{\boldsymbol{\pi}^j\}_{j=1}^N)\right] \leq \widetilde{\mathcal{O}}(\frac{\tau^{-3/2}\lambda^{-1/2}\varepsilon^{-1/2}}{\sqrt{T}} + \tau^{-1/2}\lambda^{-1/2}\varepsilon + \tau + \frac{\tau^{-1/2}\lambda^{-1/2}}{\sqrt{N}} + \frac{\tau^{-1}\lambda^{-1/2}}{N})$.*

**Proof:** *(sketch)* In addition to the full feedback case, we show that (i) for each agent the $\widehat{\mathbf{r}}_h$ are low-variance, low-bias (scaling $\mathcal{O}(\varepsilon)$) estimators of $\mathbf{F}$ evaluated at the mean policy, and (ii) the bias due to no exploration occuring can be controlled. See Theorem 4 in the appendix for a full proof. $\square$

Once again, the values of $\tau$ and exploration probability $\varepsilon$ can be chosen to incorporate a tolerable exploitability. In the case where the number of participants $N$ in the game is known, the following corollary indicates the asymptotically optimal choices for the hyperparameters.

**Corollary 2 (Optimal $\varepsilon, \tau$, bandit feedback)** *Assume the conditions of Theorem 2 for $N$ agents running TRPA-Bandit. For monotone $\mathbf{F}$, choosing $\tau = 1/\sqrt[4]{N}$ and $\varepsilon = 1/\sqrt{N}$ yields $\mathbb{E}\left[\mathcal{E}_{exp}^i(\{\boldsymbol{\pi}^j\}_{j=1}^N)\right] \leq \widetilde{\mathcal{O}}(\frac{N^{3/4}}{\sqrt{T}} + \frac{1}{\sqrt[4]{N}})$ for any $i$. For strongly monotone $\mathbf{F}$, choosing $\tau = 1/\sqrt[3]{N}$ and $\varepsilon = 1/\sqrt{N}$ yields $\mathbb{E}\left[\mathcal{E}_{exp}^i(\{\boldsymbol{\pi}^j\}_{j=1}^N)\right] \leq \widetilde{\mathcal{O}}(\frac{N^{3/4}\lambda^{-1/2}}{\sqrt{T}} + \frac{\lambda^{-1/2}}{\sqrt[3]{N}})$.*

The dependence of $N$ of the sample complexity in the bandit case is worse compared to the full feedback setting as expected: intuitively the agents must take turns to estimate the payoffs of each action in bandit feedback. Furthermore, while our problem framework is different and a direct comparison is difficult in terms of bounds, we point out that classical MMAB results such as (Lugosi & Mehrabian, 2022) have a linear dependence on $N$, while in our case the dependence on $N$ scales with $N^{3/4}$. We emphasize that the time-dependence is sublinear in terms of $N$, up to the non-vanishing finite population bias. As in the full feedback case, the non-vanishing finite population bias in the bandit feedback case scales with $\mathcal{O}(1/\sqrt[4]{N})$ or $\mathcal{O}(1/\sqrt[3]{N})$, rather than $\mathcal{O}(1/\sqrt{N})$ which would match Proposition 2. Note that the dependence of the bias on $N$ varies in various mean-field game results (Saldi et al., 2019), but asymptotically is known to converge to zero as $N \to \infty$, as our explicit finite-agent bound also demonstrates.

## 4 EXPERIMENTAL RESULTS

We validate the theoretical results of our work on numerical and two real-world experiments. The details of the setup are presented in greater detail in the appendix (Section G) and the code is provided in supplementary materials. Our experiments assume bandit feedback, although in the appendix we also include results under the full feedback model. We provide an overview of our setup first.

**Numerical problems.** Firstly, we formulate three numerical problems, which are based on examples suggested in Sections 2.2, B. We randomly generate monotone payoff operators that are linear (LINEAR) and a payoff function that admits a KL divergence potential (KL). We also analyze a particular "beach bar process" (BB), a stateless version of the example presented in (Perrin et al., 2020). For numerical examples, we use $K = 5$, and vary between various population sizes in $N = \{20, 50, 100, 200, 500, 1000\}$ to quantify the effect of finite $N$ and compare with theoretical bias bounds. We set the parameters $\varepsilon, \tau$ using the theoretical values from Corollary 2.

**Traffic flow simulation.** Using the UTD19 dataset (Loder et al., 2019), we evaluate our algorithms on traffic congestion data on three different routes for accessing the city center of Zurich (UTD). The UTD19 dataset features many closed loop sensors across various urban roads in Zurich, providing granular measurements of road occupancy and traffic flow. We use these real-world stationary detector measurements to approximate traveling times as a function of route occupancy with a kernelized regression model on three routes. We then evaluate our algorithms on estimated traveling times.

**Access to the Tor network.** We also run experiments on accessing the Tor network, which is a large decentralized network for secure communications and an active area of research in computer security (Jansen et al., 2014; McCoy et al., 2008). The Tor network consists of many decentralized servers and access to the network is achieved by communicating with one of many entry guard servers (a list advertised publicly (The Tor Network), some hosted by universities). As the entry guards serve a wide public of users and each user is free to choose an entry point, the network is an ideal setting to test our algorithms. We simulate 100 independent agents accessing Tor by choosing among $K = 5$ entry servers every minute and use the real-world ping delays for bandit feedback.

Overall, our experiments in both models and real-world use cases support our theory. Firstly, our experiments verify learning in the sense of decreasing maximum exploitability $\max_{i \in \mathcal{N}} \mathcal{E}_{exp}^i(\{\boldsymbol{\pi}^j\}_{j=1}^N)$ and mean distance to MF-NE given by $\frac{1}{N}\sum_i \|\boldsymbol{\pi}^* - \boldsymbol{\pi}_t^i\|_2$ during IL (Figure1-b,c). As expected from Corollary 2, the maximum exploitability oscillates but remains bounded (due to the effect of finite $N$). Furthermore, the agents converge to policies closer to the MF-NE as $N$ increases (Figure1-c), empirically demonstrating that the MF-NE is an accurate description of the limiting behaviour of

SMFG as $N \to \infty$. The scale of the excess exploitability in our experiments decreases rapidly in $N$, allowing our results to be significant even for hundreds or thousands of agents.

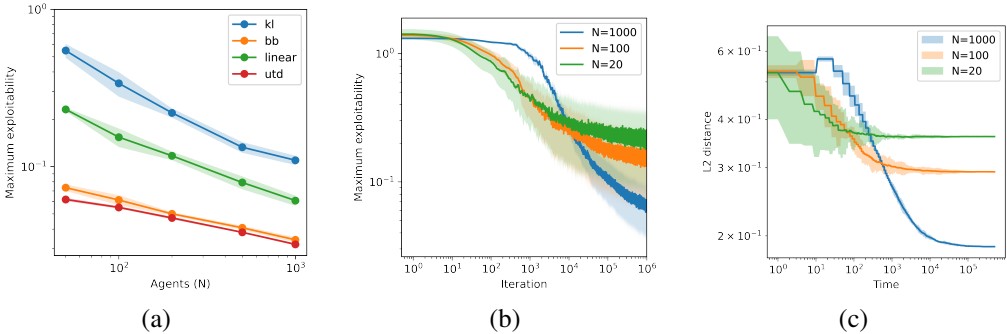

(a)  (b)  (c)

Figure 1: Results for numerical problems KL, BB, LINEAR, UTD. (a) Maximum exploitability of $N$ agents at convergence as a function of $N$ for different problems, (b) The max. exploitability among $N$ agents during training with linear payoff (LINEAR), for different $N$, (c) The mean $\ell_2$ distance of agent policies during training to the MF-NE in the Zurich traffic flow simulation problem (UTD).

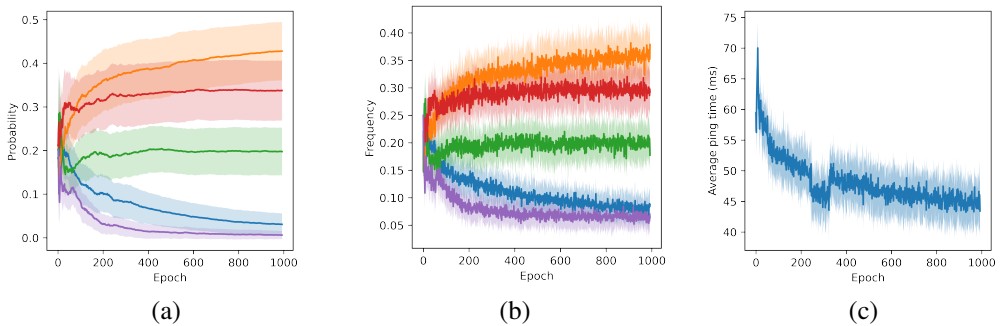

(a)  (b)  (c)

Figure 2: Results for the Tor network experiment. (a) Average policies (probability distribution) over 5 servers of the 100 agents in the Tor network access experiment, (b) Empirical distribution of agents over Tor entry servers during training on 5 servers (different colors indicate different entry servers), (c) Average waiting times for Tor network access during training.

Our experimental results in traffic congestion and server access also support our theoretical claims. Despite having no *a priori* assumption of monotonicity, our methods efficiently converge. We note that in the case of the network access experiments, we refrain from simulating thousands of agents to minimize network impact. This implies that the delays have a high dependence on external factors as the behavior of other users will be dominant. Nevertheless, our experiment indicates that our algorithm can produce competitive results in the wild (Figure 2).

Finally, we note that while there are no algorithms that have theoretical guarantees in the $N$ agent IL setting, we compare our results with a heuristic extension of online mirror descent (Pérolat et al., 2022) in the numerical problems, see Appendix G.4. Our results suggest a naive extension of OMD to our setting will lead to divergence or cycling.

## 5 DISCUSSION AND CONCLUSION

Overall, we proposed and analyzed a finite-agent, independent learning algorithm under bandit feedback with finite sample guarantees, demonstrating efficient IL is possible in a new class of games. Our theoretical results prove explicit guarantees for a class of games under IL, corresponding to open questions in MFG literature. We leave the establishment of lower bounds as well as considerations of weaker notions of monotonicity (Kotsalis et al., 2022) as future work.

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

## A   A DETAILED COMPARISON TO THE SETTING IN GUMMADI ET AL. (2013)

Since specific keywords seem to correspond to the works on mean-field approximations with bandits, we provide a greater discussion of our setting and the results by Gummadi et al. (2013). In general, our settings and models are very different, hence almost none of the results between our work and Gummadi et al. are transferable to the other. Our problem formulation, analysis, and results are fundamentally different from their setting due to the following points.

**Stationary equilibrium vs Nash equilibrium.** The most critical difference between the two works is the solution concepts. Our setting is competitive, as a natural extension, the solution concept is that of a Nash equilibrium where each agent has no incentive to change their policy. On the other hand, the setting of Gummadi et al. need not be competitive or collaborative and this distinction is not significant for their framework, their goal is to characterize convergence of the population to a stationary distribution. Their main results show that if a particular policy map $\sigma : \mathbb{Z}_{\geq 0}^{2n} \to \Delta_{\mathcal{A}}$ is prescribed on agents, the population distribution will converge to a steady state. The equilibrium concept Gummadi et al. (2013) is not *Nash*, rather stationarity.

**Optimality considerations.** As a consequence of analyzing stationarity, the results in Gummadi et al. (2013) do not analyze or aim to characterize optimality. In their analysis, a fixed map $\sigma : \mathbb{Z}_{\geq 0}^{2n} \to \Delta_{\mathcal{A}}$ is assumed to be the policy/strategy of a continuum of (i.e., infinitely many) agents, which maps observed loss/win counts (from Bernoulli distributed arm rewards) to arm probabilities. The stationary distribution in general obtained from $\sigma$ in Gummadi et al. (2013) does not have optimality properties, for instance, a fixed agent will can have arbitrary large exploitability. The main goal of Gummadi et al. (2013) is to prove the convergence of the population distribution to a steady state behaviour.

**Algorithms.** As a consequence of the previous points, Gummadi et al. abstract away any algorithmic considerations to the fixed map $\sigma$ and the particular algorithms employed by agents do not directly have significance in terms of their theoretical conclusions. Since we analyze optimality in our setting, we require a specific algorithm to be employed (TRPA and Algorithm 1).

**Independent learning.** In our setting, the notion of learning and independent learning become significant since we are aiming to obtain an approximate NE. Hence, our theoretical results bound the expected exploitability (Theorems 1, 2) in terms of number of samples. In Gummadi et al. (2013), the main aim is convergence to steady state rather than learning.

**Population regeneration.** Finally, to be able to obtain a contractive mapping yielding a population stationary distribution/steady state, Gummadi et al. (2013) assumes that the population regenerates at a constant rate $\beta$, implying agents are constantly being replaced by oblivious agents that have not observed the game dynamics. This smooths the dynamics by introducing a forgetting mechanism to game participants. Our results on the other hand are closer to the traditional bandits/independent learning setting. For instance, this would correspond to non-vanishing exploitability scaling with $\mathcal{O}(\beta)$ in our system as agents constantly "forget" what they learned.

**Other (minor) model differences.** In our setting, we assume general noisy rewards while in Gummadi et al. (2013), the rewards are Bernoulli random variables with success probability dependent on the population.

## B   EXAMPLES OF MONOTONE PAYOFFS

We first prove the monotonicity of the example provided in the main body.

**Example 3 (Monotone decreasing congestion payoffs)** *Assume there exists Lipschitz continuous functions* $F_a : [0, 1] \to [0, 1]$ *such that* $F_a$ *is non-increasing, and let* $\mathbf{F}(\boldsymbol{\mu}) := \sum_a F_a(\boldsymbol{\mu}(a))\mathbf{e}_a$. *Since we have for* $\boldsymbol{\mu}, \boldsymbol{\mu}' \in \Delta_{\mathcal{A}}$,

$$(\mathbf{F}(\boldsymbol{\mu}) - \mathbf{F}(\boldsymbol{\mu}'))^{\top}(\boldsymbol{\mu} - \boldsymbol{\mu}') = \sum_a (F_a(\boldsymbol{\mu}(a)) - F_a(\boldsymbol{\mu}'(a)))(\boldsymbol{\mu}(a) - \boldsymbol{\mu}'(a)) \leq 0,$$

*the payoff map* $\mathbf{F}$ *is monotone. This payoff map corresponds to a congestion model as more agents using a particular action leads to diminished payoffs.*

*Assume further there exists functions $F_a : [0,1] \to [0,1]$ and a constant $\lambda' > 0$ such that $|F_a(x) - F_a(x')| \geq \lambda'|x - x'|$ for all $a, x, x'$. Since we have for $\boldsymbol{\mu}, \boldsymbol{\mu}' \in \Delta_A$,*

$$(\mathbf{F}(\boldsymbol{\mu}) - \mathbf{F}(\boldsymbol{\mu}'))^\top (\boldsymbol{\mu} - \boldsymbol{\mu}') = \sum_a (F_a(\boldsymbol{\mu}(a)) - F_a(\boldsymbol{\mu}'(a)))(\boldsymbol{\mu}(a) - \boldsymbol{\mu}'(a))$$

$$\leq \sum_a \lambda'(\boldsymbol{\mu}'(a) - \boldsymbol{\mu}(a))(\boldsymbol{\mu}(a) - \boldsymbol{\mu}'(a))$$

$$= -\lambda'\|\boldsymbol{\mu} - \boldsymbol{\mu}'\|_2^2,$$

*$\mathbf{F}$ is $\lambda$-strongly monotone. This map also corresponds to payoffs decreasing with occupancy, but ruling out flat regions.*

As a special case of the example above, the classical multi-player multi-armed bandits setting with collisions also yields a monotone Lipschitz continuous payoff. We state this in the following example.

**Example 4 (Multi-player MAB with Collisions)** *For the $N$-player game, for each action $a \in \mathcal{A}$, take the functions $F_{col}^a : [0,1] \to [0,1]$ such that*

$$F_{col}^a(x) = \begin{cases} \alpha^a, \text{if } x \leq \frac{1}{N}, \\ \alpha^a N(\frac{2}{N} - x), \text{if } \frac{1}{N} \leq x \leq \frac{2}{N}, \\ 0, \text{if } x \geq \frac{2}{N}. \end{cases}$$

*where $\alpha^a \in [0,1]$ is the expected payoff of the action $a$ when no collision occurs. Take the payoff map $\mathbf{F}(\boldsymbol{\mu}) := \sum_{a \in \mathcal{A}} F_{col}^a(\mu_a)\mathbf{e}_a$. The payoff map $\mathbf{F}$ is Lipschitz continuous and monotone, and corresponds to the classical multi-player MAB with collisions.*

While the classical MMAB is a special case of SMFG, the analysis methods and solution concepts are different (for instance, our results in SMFG consider the regime when $N \to \infty$), hence classical algorithms are still preferable in the MMAB case. Instead, we emphasize that the above connection to MMAB indicates our monotonicity assumption on $\mathbf{F}$ can be seen as a model with "soft-collisions" observed in many real-world applications.

We also re-iterate the payoffs with potential as outlined in the main text as a large class of examples satisfying monotonicity.

**Example 5 (Payoffs with potential)** *If there exists a potential function $\Phi : \Delta_A \to \mathbb{R}$ that is concave such that $\mathbf{F} = \nabla\Phi$, the payoff operator $\mathbf{F}$ is monotone. Furthermore, if the potential $\Phi$ is $\lambda$-strongly concave, then $\mathbf{F}$ is $\lambda$-strongly monotone.*

### B.1 MONOTONE PAYOFFS WITHOUT POTENTIAL FUNCTION

The examples in the previous section are special cases of payoffs with a potential function. We emphasize that the problem class of monotone operators is much large than that of potential payoffs with the following example that does not admit any potential function.

**Example 6 (A monotone payoff operator without a potential)** *Let $\mathbf{S} \in \mathbb{S}_{++}^{D \times D}$ be a symmetric positive definite matrix, $\mathbf{b} \in \mathbb{R}^D$ be an arbitrary vector, and $\mathbf{X} \in \mathbb{M}^{D \times D}$ be a anti-symmetric matrix such that $\mathbf{X} = -\mathbf{X}^\top$. Take the payoff operator*

$$\mathbf{F}(\boldsymbol{\mu}) = (-\mathbf{S} + \mathbf{X})\boldsymbol{\mu} + \mathbf{b}.$$

*Then, $-\mathbf{S} + \mathbf{X}$ is not symmetric in general, therefore there exists no potential $\Phi$ such that $\mathbf{F} = \nabla\Phi$ since the Jacobian satisfies $\mathbf{J}(\mathbf{F}) = -\mathbf{S} + \mathbf{X} = \nabla^2\Phi$ and $\nabla^2\Phi$ must be symmetric. Furthermore, $(-\mathbf{S} + \mathbf{X}) + (-\mathbf{S} + \mathbf{X})^\top = -2\mathbf{S} \preceq 0$, so $\mathbf{r}$ is monotone.*

Note that in the above Example 6, $(-\mathbf{S} + \mathbf{X})$ must exclusively have negative entries on its diagonal, but can have positive/negative non-diagonal entries. Hence, this would model a problem where action payoffs have non-trivial interactions/correlations with the occupancies. Such problems can be wide-spread in complicated real-world problems which can not be modeled just by simple collisions. For instance, due to memory caching effects or power-of-two effects on virtual servers sharing

hardware such as RAM/CPU it might be the case that in certain regimes increased load on one server introduces a small benefit to the other. Likewise, systems that **adapt** to population behavior (e.g., red light durations that depend on number of cars waiting, queueing systems that incorporate non-trivial priority rules) might exhibit such behavior: where on average increased load of options leads to worse payoffs for everyone, but certain positive reinforcement regimes exist.

Finally, we prove that Example 2 in the main body of the paper is indeed monotone.

**Example 7** *Let $\mathcal{A} = \{1, \ldots, K\}$, and $\alpha_k > 0, \lambda_{k,k'} \in \mathbb{R}$ constants for all $k, k' \in \mathcal{A}, k > k'$. Take the payoff given by $\mathbf{F}(\boldsymbol{\mu}, k) := -\exp\{\alpha_k \boldsymbol{\mu}(k)\}/\sum_{k'} \exp\{\alpha_{k'} \boldsymbol{\mu}(k')\} + \sum_{k'<k} \lambda_{k,k'} \boldsymbol{\mu}(k') - \sum_{k'>k} \lambda_{k',k} \boldsymbol{\mu}(k')$.*

*The fact that $\mathbf{F}$ is Lipschitz follows from straightforward properties of the softmax function. For monotonicity, define the two operators $\mathbf{F}_1, \mathbf{F}_2 : \Delta_{\mathcal{A}} \to [0, 1]^{\mathcal{A}}$ given by*

$$\mathbf{F}_1(\boldsymbol{\mu}, a) = -\frac{\exp\{\alpha_k \boldsymbol{\mu}(k)\}}{\sum_{k'} \exp\{\alpha_{k'} \boldsymbol{\mu}(k')\}}$$

$$\mathbf{F}_2(\boldsymbol{\mu}, a) = \sum_{k'<k} \lambda_{k,k'} \boldsymbol{\mu}(k') - \sum_{k'>k} \lambda_{k',k} \boldsymbol{\mu}(k').$$

*Firstly, note that $\mathbf{F}_2(\boldsymbol{\mu}) = \mathbf{X}\boldsymbol{\mu}$ for the antisymmetric matrix $\mathbf{X}$, hence is monotone by Example 6. For $\mathbf{F}_1$, take the negative log-sum-exp potential function*

$$\Phi(\boldsymbol{\mu}) = -\log\left(\sum_{a \in \mathcal{A}} \exp\{\boldsymbol{\mu}(a)\}\right),$$

*which is concave, and note that $\mathbf{F}_1 = \nabla\Phi$, implying that $\mathbf{F}_1$ is monotone by Example 5. Finally, since both $\mathbf{F}_1, \mathbf{F}_2$ are monotone, the operator $\mathbf{F} = \mathbf{F}_1 + \mathbf{F}_2$ is also monotone.*

### B.2 MONOTONE PAYOFFS, POTENTIAL GAMES AND CONGESTION GAMES

A natural comparison of our framework is with congestion games first proposed by Rosenthal (1973). In such a setting, a monotonically decreasing payoff function (in terms of occupancy) is assigned to each action. We present the following example to emphasize that monotone operators are more general than such payoff models.

**Example 8 (Monotonicity without congestion)** *In Example 6, take the three action game $\mathcal{A} = \{a_1, a_2, a_3\}$ and the operator $\mathbf{F}(\boldsymbol{\mu}) = \mathbf{A}\boldsymbol{\mu} + \mathbf{b}$, where*

$$\mathbf{A} = \begin{pmatrix} -1 & -1 & 0 \\ 1 & 0 & 0 \\ 0 & 0 & -1 \end{pmatrix}, \mathbf{b} = \begin{pmatrix} 0 \\ 0 \\ 0 \end{pmatrix}.$$

*Hence by simple computation,*

$$\mathbf{F}\begin{pmatrix} \mu_1 \\ \mu_2 \\ \mu_3 \end{pmatrix} = \begin{pmatrix} -\mu_1 - \mu_2 \\ \mu_1 \\ -\mu_3 \end{pmatrix}.$$

*As before, $\mathbf{F}$ is monotone, but it is not a congestion game as the payoff of the action $a_2$ increases as the occupancy of $a_1$ increases.*

A related class of games in game theory that admits provable guarantees with independent learning is potential games Monderer & Shapley (1996). We show that the $N$-player SMFG is in general not a potential game, hence independent learning with best response dynamics is not trivial to analyze. We provide a particular SMFG that is a counter-example (i.e. does not admit a game potential).

**Example 9** *The SMFG is a* potential game *if there exists a map $P : \mathcal{A}^N \to \mathbb{R}$ such that for all $i \in \mathcal{N}, \mathbf{a} = (a_1, \ldots, a_N) \in \mathcal{A}^N, a \in \mathcal{A}$, it holds that*

$$V^i(a^i, \mathbf{a}^{-i}) - V^i(a, \mathbf{a}^{-i}) = P(a^i, \mathbf{a}^{-i}) - P(a, \mathbf{a}^{-i}), \tag{1}$$

where $V^i(\mathbf{a})$ denotes the expected reward of player $i$ when each player $j \in \mathcal{N}$ (deterministically) plays action $a^j$. Note that $V^i(\mathbf{a}) = \mathbf{F}(\boldsymbol{\mu_a})(a^i)$ where $\boldsymbol{\mu_a} \in \Delta_{\mathcal{A}}$ is the empirical distribution of actions over actions induced by action profile $\mathbf{a}$.

We provide an example of a SMFG where no such $P$ exists. Take $N = 3, K = 3, \mathcal{A} = \{a_1, a_2, a_3\}$, and the monotone reward $\mathbf{F}$ is defined as

$$\mathbf{F}\begin{pmatrix}\mu_1 \\ \mu_2 \\ \mu_3\end{pmatrix} = \begin{pmatrix}-\mu_1 - \mu_2 \\ \mu_1 \\ -\mu_3\end{pmatrix},$$

which is monotone. Assume a potential $P$ exists satisfying Equation 1. Then, it follows by simple computation that:

$$V^1(a_2, a_1, a_1) - V^1(a_3, a_1, a_1) = P(a_2, a_1, a_1) - P(a_3, a_1, a_1),$$
$$V^2(a_3, a_1, a_1) - V^2(a_3, a_2, a_1) = P(a_3, a_1, a_1) - P(a_3, a_2, a_1),$$
$$V^1(a_3, a_2, a_1) - V^1(a_2, a_2, a_1) = P(a_3, a_2, a_1) - P(a_2, a_2, a_1),$$
$$V^2(a_2, a_2, a_1) - V^2(a_2, a_1, a_1) = P(a_1, a_2, a_1) - P(a_1, a_1, a_1).$$

Hence, adding the inequalities above, it must hold that

$$
\begin{aligned}
0 =& V^1(a_2, a_1, a_1) - V^1(a_3, a_1, a_1) + V^2(a_3, a_1, a_1) - V^2(a_3, a_2, a_1) \\
& V^1(a_3, a_2, a_1) - V^1(a_2, a_2, a_1) + V^2(a_2, a_2, a_1) - V^2(a_2, a_1, a_1) \\
=& \mathbf{F}((2/3, 1/3, 0), a_2) - \mathbf{F}((2/3, 0, 1/3), a_3) + \mathbf{F}((2/3, 0, 1/3), a_1) - \mathbf{F}((1/3, 1/3, 1/3), a_2) \\
& + \mathbf{F}((1/3, 1/3, 1/3), a_3) - \mathbf{F}((1/3, 2/3, 0), a_2) + \mathbf{F}((1/3, 2/3, 0), a_2) - \mathbf{F}((2/3, 1/3, 0), a_1) \\
=& 2/3 - (-1/3) + (-2/3) - 1/3 \\
& + (-1/3) - 1/3 + 1/3 - (-1) \\
\neq& 0,
\end{aligned}
$$

leading to a contradiction. Hence no such potential $P$ exists, and this (monotone) SMFG is not a potential game.

## C  FORMALIZING LEARNING ALGORITHMS

In this section, we formalize the concept of an independent learning algorithm in the full feedback and bandit feedback setting. In general, we formalize the notion of an algorithm as a map $A_t^i : \mathcal{H}_t^i \to \Delta_{\mathcal{A}}$ that maps the set of past observations of agent $i$ at time $t$ to action selection probabilities. The definition of the set $\mathcal{H}_t^i$ varies in the full feedback and bandit feedback, we define these rigorously.

**Definition 5 (Learning algorithm with full feedback)** *An independent learning algorithm with full information $\mathbf{A} = \{A_t^i\}_{i,t}$ is a sequence of mappings for each player with*

$$A_t^i : \Delta_{\mathcal{A}}^{t-1} \times \mathcal{A}^{t-1} \times [0,1]^{(t-1) \times K} \to \Delta_{\mathcal{A}},$$
$$A_0^i \in \Delta_{\mathcal{A}},$$

*that maps past $t - 1$ observations from previous rounds to a mixed strategy on actions $\mathcal{A}$ at time $t$ for each agent $i$.*

Naturally, we are interested in algorithms that yield the NE at the limit with high probability or in expectation.

**Definition 6 (Rational learning algorithm with full feedback)** *Let $\mathbf{A}$ be an algorithm with full feedback as defined in Definition 5. We call $\mathbf{A}$ $\delta$-rational if it holds that for all $i$, the induced mixed strategies $\boldsymbol{\pi}_t^i$ under $\boldsymbol{\pi}_0^i = A_0^i, \boldsymbol{\pi}_t^i = A_t^i(\boldsymbol{\pi}_0^i, \ldots, \boldsymbol{\pi}_{t-1}^i, a_0^i, \ldots, a_{t-1}^i, \mathbf{r}_0^i, \ldots, \mathbf{r}_{t-1}^i)$ satisfy*

$$\lim_{t \to \infty} \mathbb{E}[\mathcal{E}_{exp}^i(\{\boldsymbol{\pi}_t^j\}_{j=1}^N)] \leq \delta, \text{ for all } i \in \mathcal{N}.$$

Note that while not specified in the definition above, we will also be interested in the rate of convergence of the exploitability term for a consistent algorithm. Finally, we also formalize the bandit setting.

**Definition 7 (Algorithm with bandit beedback)** *An algorithm with bandit feedback* $\mathbf{A} = \{A_t^i\}_{i,t}$ *is a sequence of mappings for each player with*

$$A_t^i : \Delta_{\mathcal{A}}^{t-1} \times \mathcal{A}^{t-1} \times [0,1]^{(t-1)} \to \Delta_{\mathcal{A}},$$
$$A_0^i \in \Delta_{\mathcal{A}},$$

*that maps past* $t - 1$ *observations from previous rounds (only including the payoffs of the* played *actions) to a probability distribution on actions* $\mathcal{A}$.

**Definition 8 (Rational algorithm with bandit feedback)** *Let* $\mathbf{A}$ *be an algorithm with bandit feedback as defined in Definition 7. We call* $\mathbf{A}$ $\delta$-*rational if it holds that for all* $i$, *the induced (random) mixed strategies* $\boldsymbol{\pi}_t^i$ *under* $\boldsymbol{\pi}_0^i = A_0^i, \boldsymbol{\pi}_t^i = A_t^i(\boldsymbol{\pi}_0^i, \dots, \boldsymbol{\pi}_{t-1}^i, a_0^i, \dots, a_{t-1}^i, r_0^i, \dots, r_{t-1}^i)$ *satisfy*

$$\lim_{t \to \infty} \mathbb{E}[\mathcal{E}_{exp}^i(\{\boldsymbol{\pi}_t^j\}_{j=1}^N)] \leq \delta, \text{ for all } i \in \mathcal{N}.$$

## D  BASIC INEQUALITIES

In our proofs, we will need to repeatedly bound certain recurrences and sums. In this section, we present useful inequalities to this end.

**Lemma 1 (Harmonic partial sum bound)** *For any integers* $s, \bar{s}$ *such that* $1 \leq \bar{s} < s$ *and* $p \neq -1$, *it holds that*

$$\log s - \log \bar{s} + \frac{1}{\bar{s}} \leq \sum_{n=\bar{s}}^{s} \frac{1}{n} \leq \frac{1}{s} + \log s - \log \bar{s},$$

$$\frac{s^{p+1}}{p+1} - \frac{\bar{s}^{p+1}}{(p+1)} + \bar{s}^p \leq \sum_{n=\bar{s}}^{s} n^p \leq \frac{s^{p+1}}{p+1} - \frac{\bar{s}^{p+1}}{p+1} + s^p, \text{ if } p \geq 0$$

$$\frac{s^{p+1}}{p+1} - \frac{\bar{s}^{p+1}}{p+1} + s^p \leq \sum_{n=\bar{s}}^{s} n^p \leq \frac{s^{p+1}}{p+1} - \frac{\bar{s}^{p+1}}{p+1} + \bar{s}^p, \text{ if } p \leq 0$$

**Proof:**  Let $f_1 : \mathbb{R}_{\geq 0} \to \mathbb{R}_{\geq 0}$ be a non-decreasing positive function and $f_2 : \mathbb{R}_{\geq 0} \to \mathbb{R}_{\geq 0}$ be a non-increasing positive function. Then it holds that

$$\int_{x=\bar{s}}^{s} f_1(x)dx + f_1(\bar{s}) \leq \sum_{n=\bar{s}}^{s} f_1(n) \leq \int_{x=\bar{s}}^{s} f_1(x)dx + f_1(s),$$

$$\int_{x=\bar{s}}^{s} f_2(x)dx + f_2(s) \leq \sum_{n=\bar{s}}^{s} f_2(n) \leq \int_{x=\bar{s}}^{s} f_2(x)dx + f_2(\bar{s}).$$

The result follows from a simple integral bound with $\int \frac{1}{x} dx = \log x$ and $\int x^p dx = \frac{x^{p+1}}{p+1}$. $\square$

We state a certain recurrence inequality that appears several times in our analysis as a lemma, in order to shorten some proofs.

**Lemma 2 (General error recurrence)** *Let* $c_0 \geq 0, c_1 \geq 0, \gamma > 1$ *be arbitrary constants. Furthermore, let* $\{u_t\}_{t=0}^{\infty}$ *be a sequence of non-negative numbers such that for all* $t \geq 0$, *it holds that*

$$u_{t+1} \leq \frac{c_0}{t+1} + \frac{c_1}{(t+1)^2} + \left(1 - \frac{\gamma}{t+1}\right) u_t.$$

*Then, for all values of* $t \geq 0$, *it holds that:*

$$u_{t+1} \leq \frac{c_0}{t+1} + \frac{c_1}{(t+1)^2} + \frac{u_0}{(t+1)^{\gamma}} + \gamma^{-1} c_0 + \frac{c_1}{(t+1)(\gamma-1)} + \frac{c_1}{(t+1)^{\gamma}}.$$

**Proof:** We note that inductively, we have

$$u_{t+1} \leq \frac{c_0}{t+1} + \frac{c_1}{(t+1)^2} + u_0 \prod_{s=0}^{t} \left(1 - \frac{\gamma}{s+1}\right) + \sum_{s=0}^{t-1} \left(\frac{c_0}{s+1} + \frac{c_1}{(s+1)^2}\right) \prod_{s'=s+1}^{t} \left(1 - \frac{\gamma}{s'+1}\right).$$

Using the inequality $1 + x \leq e^x$ and Lemma 1, we obtain

$$u_{t+1} \leq \frac{c_0}{t+1} + \frac{c_1}{(t+1)^2} + u_0 \prod_{s=0}^{t} \exp\left\{-\frac{\gamma}{s+1}\right\} + \sum_{s=0}^{t-1} \left(\frac{c_0}{s+1} + \frac{c_1}{(s+1)^2}\right) \prod_{s'=s+1}^{t} \exp\left\{-\frac{\gamma}{s'+1}\right\}$$

$$\leq \frac{c_0}{t+1} + \frac{c_1}{(t+1)^2} + u_0 \exp\left\{-\sum_{s=0}^{t} \frac{\gamma}{s+1}\right\} + \sum_{s=0}^{t-1} \left(\frac{c_0}{s+1} + \frac{c_1}{(s+1)^2}\right) \exp\left\{-\sum_{s'=s+1}^{t} \frac{\gamma}{s'+1}\right\}$$

$$\leq \frac{c_0}{t+1} + \frac{c_1}{(t+1)^2} + u_0 \exp\left\{-\gamma \log(t+1)\right\} + \sum_{s=0}^{t-1} \left(\frac{c_0}{s+1} + \frac{c_1}{(s+1)^2}\right) \exp\left\{-\gamma \log\left(\frac{t+1}{s+1}\right)\right\}$$

$$\leq \frac{c_0}{t+1} + \frac{c_1}{(t+1)^2} + \frac{u_0}{(t+1)^\gamma} + \sum_{s=0}^{t-1} \left(\frac{c_0}{s+1} + \frac{c_1}{(s+1)^2}\right) \left(\frac{s+1}{t+1}\right)^\gamma$$

$$\leq \frac{c_0}{t+1} + \frac{c_1}{(t+1)^2} + \frac{u_0}{(t+1)^\gamma} + (t+1)^{-\gamma} \sum_{s=0}^{t-1} \left(\frac{c_0}{s+1} + \frac{c_1}{(s+1)^2}\right) (s+1)^\gamma$$

$$\leq \frac{c_0}{t+1} + \frac{c_1}{(t+1)^2} + \frac{u_0}{(t+1)^\gamma} + (t+1)^{-\gamma} \sum_{s=0}^{t-1} \left(c_0 (s+1)^{\gamma-1} + c_1 (s+1)^{\gamma-2}\right)$$

The last term can be bound with the corresponding integral (see Lemma 1), yielding (since $\gamma - 1 > 0$):

$$\sum_{s=0}^{t-1} (s+1)^{\gamma-1} \leq \frac{(t+1)^\gamma}{\gamma}.$$

For the term $\sum_{s=0}^{t-1} (s+1)^{\gamma-2}$, we analyze to cases. If $1 < \gamma \leq 2$, we have

$$\sum_{s=0}^{t-1} (s+1)^{\gamma-2} \leq \frac{(t+1)^{\gamma-1}}{\gamma-1} + 1$$

otherwise, if $\gamma \geq 2$, then

$$\sum_{s=0}^{t-1} (s+1)^{\gamma-2} \leq \frac{(t+1)^{\gamma-1}}{\gamma-1}.$$

Hence the two inequalities combined yield the stated bound,

$$u_{t+1} \leq \frac{c_0}{t+1} + \frac{c_1}{(t+1)^2} + \frac{u_0}{(t+1)^\gamma} + \frac{1}{(t+1)^\gamma} \sum_{s=0}^{t-1} \left(c_0 (s+1)^{\gamma-1} + c_1 (s+1)^{\gamma-2}\right)$$

$$\leq \frac{c_0}{t+1} + \frac{c_1}{(t+1)^2} + \frac{u_0}{(t+1)^\gamma} + \gamma^{-1} c_0 + \frac{c_1}{(t+1)(\gamma-1)} + \frac{c_1}{(t+1)^\gamma}.$$

$\square$

# E  GENERAL RESULTS FOR MF-NE AND VIS

## E.1  EXISTENCE AND UNIQUENESS OF MF-NE

The problem of finding a MF-NE is equivalent to finding a distribution over actions $\boldsymbol{\pi}^* \in \Delta_{\mathcal{A}}$

$$\forall \boldsymbol{\pi} \in \Delta_{\mathcal{A}}, \mathbf{F}(\boldsymbol{\pi}^*)^\top (\boldsymbol{\pi}^* - \boldsymbol{\pi}) \geq 0.$$

that is, it is the solution of the VI corresponding to $\mathbf{F}$. The existence of solutions to VIs is a well-studied problem, and we state the main existence and uniqueness theorem.

**Proposition 1 (Existence and Uniqueness of MF-NE)** *Let* $\mathbf{F} : \Delta_{\mathcal{A}} \to [0,1]^K$ *be a continuous function. Then* $\mathbf{F}$ *has at least one MF-NE* $\boldsymbol{\pi}^*$, *and the set of MF-NE is compact. Furthermore, if* $\mathbf{F}$ *is also* $\lambda$-*strongly monotone for some* $\lambda > 0$, *then the MF-NE is unique.*

**Proof:** The MF-NE corresponds to solutions of the VI:

$$\forall \boldsymbol{\pi} \in \Delta_{\mathcal{A}}, \mathbf{F}(\boldsymbol{\pi}^*)^\top (\boldsymbol{\pi}^* - \boldsymbol{\pi}) \geq 0.$$

The domain set $\Delta_{\mathcal{A}}$ is compact and convex, and the assumption that $\mathbf{F}$ is continuous yields the existence of a solution using Corollary 2.2.5 of Facchinei & Pang (2003).

For uniqueness in the case of strong monotonicity, see Theorem 2.3.3 of Facchinei & Pang (2003). $\square$

### E.2 MF-NE, NE AND EXPLOITABILITY BOUNDS

The critical theoretical building block of the paper is the relationship between MF-NE, NE and explicit bounds in terms of the former on exploitability. We restate several results from the main body and provide the full proofs, as well as proving certain useful intermediary lemmas.

**Lemma 3** *For any $\boldsymbol{\pi}^1, \ldots, \boldsymbol{\pi}^N \in \Delta_{\mathcal{A}}$, it holds that*

$$\left| V^i(\boldsymbol{\pi}_1, \ldots, \boldsymbol{\pi}_N) - \boldsymbol{\pi}^{i,\top} \, \mathbb{E} \left[ \mathbf{F}(\widehat{\boldsymbol{\mu}}) \Big|_{\widehat{\boldsymbol{\mu}} := \frac{1}{N} \sum_{j=1}^N \mathbf{e}_{a^j}}^{a^j \sim \boldsymbol{\pi}^j, \forall j} \right] \right| \leq \frac{L\sqrt{2}}{N}.$$

**Proof:** We introduce a random variable $\bar{a}^i$ which is independent from other players' actions $\{a^j\}_{j=1}^N$ and has distribution $\boldsymbol{\pi}^i$. Then, it holds by simple computation that

$$
\begin{aligned}
V^i(\boldsymbol{\pi}_1, \ldots, \boldsymbol{\pi}_N) &= \mathbb{E}\left[ \mathbf{F}(\widehat{\boldsymbol{\mu}}, a^i) \right] = \mathbb{E}\left[ \mathbf{F}(\widehat{\boldsymbol{\mu}}, a^i) \right] = \mathbb{E}\left[ \mathbf{F}\left( \frac{1}{N} \sum_{j=1}^N \mathbf{e}_{a^j}, a^i \right) \right] \\
&= \mathbb{E}\left[ \mathbf{F}\left( \frac{1}{N}\left( \sum_{j=1, j\neq i}^N \mathbf{e}_{a^j} + \mathbf{e}_{\bar{a}^i} \right), a^i \right) \right] \\
&\quad + \mathbb{E}\left[ \mathbf{F}\left( \frac{1}{N}\sum_{j=1}^N \mathbf{e}_{a^j}, a^i \right) - \mathbf{F}\left( \frac{1}{N}\left( \sum_{j=1, j\neq i}^N \mathbf{e}_{a^j} + \mathbf{e}_{\bar{a}^i} \right), a^i \right) \right].
\end{aligned}
$$

For the first term above, we observe that

$$
\begin{aligned}
\mathbb{E}\left[ \mathbf{F}\left( \frac{1}{N}\left( \sum_{j=1, j\neq i}^N \mathbf{e}_{a^j} + \mathbf{e}_{\bar{a}^i} \right), a^i \right) \right] &= \mathbb{E}\left[ \mathbb{E}\left[ \mathbf{F}\left( \frac{1}{N}\left( \sum_{j=1, j\neq i}^N \mathbf{e}_{a^j} + \mathbf{e}_{\bar{a}^i} \right), a^i \right) \Big| a^i \right] \right] \\
&= \mathbb{E}\left[ \mathbb{E}\left[ \mathbf{e}_{a^i}^\top \mathbf{F}\left( \frac{1}{N}\left( \sum_{j=1, j\neq i}^N \mathbf{e}_{a^j} + \mathbf{e}_{\bar{a}^i} \right) \right) \Big| a^j \right] \right] \\
&= \mathbb{E}\left[ \mathbf{e}_{a^i}^\top \mathbb{E}\left[ \mathbf{F}\left( \frac{1}{N}\left( \sum_{j=1, j\neq i}^N \mathbf{e}_{a^j} + \mathbf{e}_{\bar{a}^i} \right) \right) \Big| a^i \right] \right] \\
&= \mathbb{E}\left[ \mathbf{e}_{a^i}^\top \right] \mathbb{E}\left[ \mathbf{F}\left( \frac{1}{N}\left( \sum_{j=1, j\neq i}^N \mathbf{e}_{a^j} + \mathbf{e}_{\bar{a}^i} \right) \right) \right] \\
&= \boldsymbol{\pi}^{i,\top} \mathbb{E}[\mathbf{F}(\widehat{\boldsymbol{\mu}})],
\end{aligned}
$$

since $\{a^j\}_i^N$ and $(\bar{a}^i, \mathbf{a}^{-i})$ are identically distributed. The second term above can be bounded using

$$
\left| \mathbf{F}\left( \frac{1}{N} \sum_{j=1}^N \mathbf{e}_{a^j}, a^i \right) - \mathbf{F}\left( \frac{1}{N} \left( \sum_{j=1, j\neq i}^N \mathbf{e}_{a^j} + \mathbf{e}_{\bar{a}^i} \right), a^i \right) \right|
$$

$$
= \left| \mathbf{e}_{a^i}^\top \mathbf{F}\left( \frac{1}{N} \sum_{j=1}^N \mathbf{e}_{a^j} \right) - \mathbf{e}_{a^i}^\top \mathbf{F}\left( \frac{1}{N} \left( \sum_{j=1, j\neq i}^N \mathbf{e}_{a^j} + \mathbf{e}_{\bar{a}^i} \right) \right) \right|
$$

$$
\leq \| \mathbf{e}_{a^i} \|_2 \left\| \mathbf{F}\left( \frac{1}{N} \sum_{j=1}^N \mathbf{e}_{a^j} \right) - \mathbf{F}\left( \frac{1}{N} \left( \sum_{j=1, j\neq i}^N \mathbf{e}_{a^j} + \mathbf{e}_{\bar{a}^i} \right) \right) \right\|_2
$$

$$
\leq L \left\| \frac{1}{N} (\mathbf{e}_{a^i} - \mathbf{e}_{\bar{a}^i}) \right\|_2 \leq \frac{L\sqrt{2}}{N}.
$$

The last inequality follows from the fact that $\mathbf{F}$ is $L$-Lipschitz. Hence the result follows. $\square$

**Proposition 2** *Let $\mathbf{F}$ be $L$-Lipschitz, $\delta \geq 0$, and $\boldsymbol{\pi}^*$ be a $\delta$-MF-NE. Then, the strategy profile $(\boldsymbol{\pi}^*, \ldots, \boldsymbol{\pi}^*) \in \Delta_{\mathcal{A}}^N$ is a $\mathcal{O}\left( \delta + \frac{L}{\sqrt{N}} \right)$-NE.*

**Proof:** Firstly, define the independent random variables $a^j \sim \boldsymbol{\pi}_j$ for all $j \in \mathcal{N}$ for some $\boldsymbol{\pi}_j \in \Delta_{\mathcal{A}}$, and let $\bar{\boldsymbol{\pi}} = 1/N \sum_{j=1}^N \boldsymbol{\pi}_j$. As before, define the random variable $\widehat{\boldsymbol{\mu}} := 1/N \sum_{j=1}^N \mathbf{e}_{a^j}$. It is straightforward that $\mathbb{E}\left[ \frac{1}{N} \sum_{j=1}^N \mathbf{e}_{a^j} \middle| a^j \sim \boldsymbol{\pi}_j \right] = \bar{\boldsymbol{\pi}}$, furthermore, by independence of the random vectors $\mathbf{e}_{a^j}$, we have

$$
\mathbb{E}\left[ \| \widehat{\boldsymbol{\mu}} - \boldsymbol{\pi}_j \|_2 \middle| a^j \sim \boldsymbol{\pi}_j \right] \leq \sqrt{ \mathbb{E}\left[ \left\| \frac{1}{N} \sum_{j=1}^N \mathbf{e}_{a^j} - \boldsymbol{\pi}_j \right\|_2^2 \middle| a^j \sim \boldsymbol{\pi}_j \right] }
$$

$$
\leq \sqrt{ \frac{1}{N^2} \sum_{j=1}^N \mathbb{E}\left[ \| \mathbf{e}_{a^j} - \boldsymbol{\pi}_j \|_2^2 \middle| a^j \sim \boldsymbol{\pi}_j \right] }
$$

$$
\leq \frac{2}{\sqrt{N}}.
$$

Hence, we have that

$$
\| \mathbb{E}[\mathbf{F}(\widehat{\boldsymbol{\mu}}) | a_j \sim \boldsymbol{\pi}_j] - \mathbf{F}(\bar{\boldsymbol{\pi}}) \|_2 \leq \mathbb{E}[\| \mathbf{F}(\widehat{\boldsymbol{\mu}}) - \mathbf{F}(\bar{\boldsymbol{\pi}}) \|_2] \leq \frac{2L}{\sqrt{N}}.
$$

Now let $i \in \mathcal{N}$ be arbitrary, and let $\boldsymbol{\pi}' \in \Delta_{\mathcal{A}}$ be any distribution over actions that satisfies $V^i(\boldsymbol{\pi}', \boldsymbol{\pi}^{*,-i}) = \max_{\boldsymbol{\pi}} V^i(\boldsymbol{\pi}, \boldsymbol{\pi}^{*,-i})$. We also define the quantities

$$
\overline{\mathbf{F}}_1 = \mathbb{E}\left[ \mathbf{F}(\widehat{\boldsymbol{\mu}}) \middle| a^j \sim \boldsymbol{\pi}^*, \forall j \in \mathcal{N} \right],
$$

$$
\overline{\mathbf{F}}_2 = \mathbb{E}\left[ \mathbf{F}(\widehat{\boldsymbol{\mu}}) \middle| a^j \sim \boldsymbol{\pi}^j, \forall i \neq j, a^i \sim \boldsymbol{\pi}' \right].
$$

We will bound $V^i(\boldsymbol{\pi}', \boldsymbol{\pi}^{*,-i}) - V^i(\boldsymbol{\pi}^*, \boldsymbol{\pi}^{*,-i})$. Firstly, using Lemma 3 and the result above, we observe that

$$
|V^i(\boldsymbol{\pi}', \boldsymbol{\pi}^{*,-i}) - \boldsymbol{\pi}'^\top \mathbf{F}(\boldsymbol{\pi}^*)| \leq |V^i(\boldsymbol{\pi}', \boldsymbol{\pi}^{*,-i}) - \boldsymbol{\pi}'^\top \overline{\mathbf{F}}_2| + |\boldsymbol{\pi}'^\top \overline{\mathbf{F}}_2 - \boldsymbol{\pi}'^\top \mathbf{F}(\frac{N-1}{N} \boldsymbol{\pi}^* + \frac{1}{N} \boldsymbol{\pi}')|
$$

$$
+ |\boldsymbol{\pi}'^\top \mathbf{F}(\frac{N-1}{N} \boldsymbol{\pi}^* + \frac{1}{N} \boldsymbol{\pi}') - \boldsymbol{\pi}'^\top \mathbf{F}(\boldsymbol{\pi}^*)|
$$

$$
\leq \frac{L\sqrt{2}}{N} + \frac{2L}{\sqrt{N}} + \frac{2L}{N}
$$

since $\mathbf{F}$ is $L$-Lipschitz. Likewise, we have

$$|V^i(\boldsymbol{\pi}^*, \boldsymbol{\pi}^{*,-i}) - \boldsymbol{\pi}^{*,\top}\mathbf{F}(\boldsymbol{\pi}^*)| \leq \frac{L\sqrt{2}}{N} + \frac{2L}{\sqrt{N}}.$$

Finally, using the definition of a $\delta$-MF-NE, it holds that

$$
\begin{aligned}
V^i(\boldsymbol{\pi}', \boldsymbol{\pi}^{*,-i}) - V^i(\boldsymbol{\pi}^*, \boldsymbol{\pi}^{*,-i}) \leq &\mathbf{F}(\boldsymbol{\pi}^*)^\top(\boldsymbol{\pi}' - \boldsymbol{\pi}^*) \\
&+ |V^i(\boldsymbol{\pi}^*, \boldsymbol{\pi}^{*,-i}) - \boldsymbol{\pi}^{*,\top}\mathbf{F}(\boldsymbol{\pi}^*)| + |V^i(\boldsymbol{\pi}', \boldsymbol{\pi}^{*,-i}) - \boldsymbol{\pi}'^\top\mathbf{F}(\boldsymbol{\pi}^*)| \\
\leq &\delta + \frac{L(2\sqrt{2}+4)}{N} + \frac{4L}{\sqrt{N}}.
\end{aligned}
$$

$\square$

Our goal in the context of the $N$-player SMFG is to find policies $\{\boldsymbol{\pi}^j\}_{j=1}^N$ with low exploitability $\mathcal{E}_{\text{exp}}^i$ for all $i$. We will next establish a sequence of results that explicitly bound $\mathcal{E}_{\text{exp}}^i$ in terms of various quantities related to the MF-NE. The next lemma shows that for strongly monotone problems, bounding the distance $\|\boldsymbol{\pi} - \boldsymbol{\pi}^*\|_2$ to the unique MF-NE $\boldsymbol{\pi}^*$ can be used to obtain an approximate MF-NE.

**Lemma 4 (Exploitability and $\|\boldsymbol{\pi} - \boldsymbol{\pi}^*\|_2$)** *Let $\mathbf{F} : \Delta_{\mathcal{A}} \to [0,1]^{\mathcal{A}}$ be $L$-Lipschitz, $\lambda$-strongly monotone. Assume $\boldsymbol{\pi}^*$ is the (unique) MF-NE corresponding to $\mathbf{F}$, and $\boldsymbol{\pi} \in \Delta_{\mathcal{A}}$ with $\delta := \|\boldsymbol{\pi} - \boldsymbol{\pi}^*\|_2$. Then, it holds that for all $\boldsymbol{\pi}' \in \Delta_{\mathcal{A}}$,*

$$\mathbf{F}(\boldsymbol{\pi})^\top\boldsymbol{\pi} \geq \mathbf{F}(\boldsymbol{\pi})^\top\boldsymbol{\pi}' - c\delta.$$

*for $c := (2L + \sqrt{|\mathcal{A}|})$, that is, $\boldsymbol{\pi}$ is a $c\delta$-MF-NE.*

**Proof:** Let $\boldsymbol{\pi}' \in \Delta_{\mathcal{A}}$ be arbitrary. Using the definition of the MF-NE $\boldsymbol{\pi}^*$,

$$
\begin{aligned}
\mathbf{F}(\boldsymbol{\pi}^*)^\top\boldsymbol{\pi}^* \geq &\mathbf{F}(\boldsymbol{\pi}^*)^\top\boldsymbol{\pi}'. \\
\mathbf{F}(\boldsymbol{\pi})^\top\boldsymbol{\pi} + (\mathbf{F}(\boldsymbol{\pi}^*)^\top\boldsymbol{\pi}^* - \mathbf{F}(\boldsymbol{\pi})^\top\boldsymbol{\pi}) \geq &\mathbf{F}(\boldsymbol{\pi})^\top\boldsymbol{\pi}' + (\mathbf{F}(\boldsymbol{\pi}^*)^\top\boldsymbol{\pi}' - \mathbf{F}(\boldsymbol{\pi})^\top\boldsymbol{\pi}') \\
\mathbf{F}(\boldsymbol{\pi})^\top(\boldsymbol{\pi} - \boldsymbol{\pi}') \geq &\underbrace{(\mathbf{F}(\boldsymbol{\pi}^*)^\top\boldsymbol{\pi}' - \mathbf{F}(\boldsymbol{\pi})^\top\boldsymbol{\pi}') - (\mathbf{F}(\boldsymbol{\pi}^*)^\top\boldsymbol{\pi}^* - \mathbf{F}(\boldsymbol{\pi})^\top\boldsymbol{\pi})}_{\epsilon}.
\end{aligned}
$$

The quantity $\epsilon$ can be bounded by

$$
\begin{aligned}
|\epsilon| \leq &|(\mathbf{F}(\boldsymbol{\pi}^*) - \mathbf{F}(\boldsymbol{\pi}))^\top\boldsymbol{\pi}'| + |\mathbf{F}(\boldsymbol{\pi}^*)^\top\boldsymbol{\pi}^* - \mathbf{F}(\boldsymbol{\pi}^*)^\top\boldsymbol{\pi}| + |\mathbf{F}(\boldsymbol{\pi}^*)^\top\boldsymbol{\pi} - \mathbf{F}(\boldsymbol{\pi})^\top\boldsymbol{\pi}| \\
\leq &\|\boldsymbol{\pi}'\|_2\|\mathbf{F}(\boldsymbol{\pi}^*) - \mathbf{F}(\boldsymbol{\pi})\|_2 + \|\mathbf{F}(\boldsymbol{\pi}^*)\|_2\|\boldsymbol{\pi}^* - \boldsymbol{\pi}\|_2 + \|\boldsymbol{\pi}\|_2\|\mathbf{F}(\boldsymbol{\pi}^*) - \mathbf{F}(\boldsymbol{\pi})\|_2 \\
\leq &\delta(2L + \sqrt{|\mathcal{A}|}).
\end{aligned}
$$

$\square$

Finally, we will need the following result regarding Tikhonov regularized solutions to MF-NE. Namely, our approach will involve solving a Tikhonov regularized version of the VI problem for stability, that is, our algorithms will converge to the solutions of the regularized problem MF-RVI. The following result shows that as expected, the solutions of the MF-RVI are approximate MF-NE.

**Lemma 5 (MF-NE and Tikhonov regularization)** *Let $\mathbf{F} : \Delta_{\mathcal{A}} \to [0,1]^{\mathcal{A}}$ be $\lambda \geq 0$ monotone, $L$-Lipschitz. Assume further that $\boldsymbol{\pi}_{\tau,\delta} \in \Pi$ is a $\delta$-MF-NE for the $(\lambda+\tau)$-strongly monotone operator $\mathbf{F} - \tau\mathbf{I}$. Then, it holds that $\boldsymbol{\pi}_{\tau,\delta}$ is a $\delta + 2\tau$-MF-NE for the operator $\mathbf{F}$.*

**Proof:** The proof simply follow by writing down the MF-NE inequality that $\forall \boldsymbol{\pi} \in \Delta_{\mathcal{A}}$,

$$(\mathbf{F} - \tau\mathbf{I})(\boldsymbol{\pi}_{\tau,\delta})^\top\boldsymbol{\pi}_{\tau,\delta} \geq (\mathbf{F} - \tau\mathbf{I})(\boldsymbol{\pi}_{\tau,\delta})^\top\boldsymbol{\pi} - \delta,$$

$$\mathbf{F}(\boldsymbol{\pi}_{\tau,\delta})^\top\boldsymbol{\pi}_{\tau,\delta} \geq \mathbf{F}(\boldsymbol{\pi}_{\tau,\delta})^\top\boldsymbol{\pi} + \tau\boldsymbol{\pi}_{\tau,\delta}^\top(\boldsymbol{\pi}_{\tau,\delta} - \boldsymbol{\pi}) - \delta,$$

and a simple bound on the term $|\boldsymbol{\pi}_{\tau,\delta}^\top(\boldsymbol{\pi}_{\tau,\delta} - \boldsymbol{\pi})| \leq \|\boldsymbol{\pi}_{\tau,\delta}\|_2\|\boldsymbol{\pi}_{\tau,\delta} - \boldsymbol{\pi}\|_2 \leq 2$. $\square$

Before moving on to the proof of Lemma 7, we will require an additional technical result regarding the Lipschitz continuity of $\mathcal{E}_{\text{exp}}^i$.

**Lemma 6 ($\mathcal{E}^i_{\text{exp}}$ is Lipschitz)** *For any $N$, the exploitability function $\mathcal{E}^i_{exp} : \Delta^N_{\mathcal{A}} \to \mathbb{R}$ is Lipschitz continuous, that is, for any $(\boldsymbol{\pi}^1, \ldots, \boldsymbol{\pi}^N) \in \Delta^N_{\mathcal{A}}, \boldsymbol{\pi}, \bar{\boldsymbol{\pi}} \in \Delta_{\mathcal{A}}$,*

$$|\mathcal{E}^i_{exp}(\boldsymbol{\pi}, \boldsymbol{\pi}^{-j}) - \mathcal{E}^i_{exp}(\bar{\boldsymbol{\pi}}, \boldsymbol{\pi}^{-j})| \leq L_{j,i}\|\boldsymbol{\pi} - \bar{\boldsymbol{\pi}}\|_2,$$

*with the Lipschitz modulus given by*

$$L_{i,j} = \begin{cases} \sqrt{K}, & \text{if } i = j, \\ \frac{4L\sqrt{2K}}{N}, & \text{if } i \neq j. \end{cases}$$

**Proof:** We first prove the fact that $V^i$ is Lipschitz. In the proof, we denote the empirical action distribution induced by actions $\{a^j\}^N_{j=1} \in \mathcal{A}^N$ by $\widehat{\boldsymbol{\mu}}(\{a^j\}^N_{j=1}) \in \Delta_{\mathcal{A}}$.

$$
\begin{aligned}
&|V^i(\boldsymbol{\pi}, \boldsymbol{\pi}^{-i}) - V^i(\boldsymbol{\pi}', \boldsymbol{\pi}^{-i})| \\
&\leq \left|\mathbb{E}\left[\mathbf{F}\left(\widehat{\boldsymbol{\mu}}(\{a^k\}^N_{k=1}), a^i\right)\big|a^j \sim \boldsymbol{\pi}^j, \forall j \neq i, a^i \sim \boldsymbol{\pi}\right] - \mathbb{E}\left[\mathbf{F}\left(\widehat{\boldsymbol{\mu}}(\{a^k\}^N_{k=1}), a^i\right)\big|a^j \sim \boldsymbol{\pi}^j, \forall j \neq i, a^i \sim \boldsymbol{\pi}'\right]\right| \\
&\leq \left|\sum_{\substack{a^j \in \mathcal{A} \\ j \neq i}} \prod_{j \neq i} \boldsymbol{\pi}^j(a^j) \sum_{a^i \in \mathcal{A}} \boldsymbol{\pi}(a^i)\mathbf{F}(\widehat{\boldsymbol{\mu}}(\{a^k\}^N_{k=1}), a^i) - \sum_{\substack{a^j \in \mathcal{A} \\ j \neq i}} \prod_{j \neq i} \boldsymbol{\pi}^j(a^j) \sum_{a^i \in \mathcal{A}} \boldsymbol{\pi}'(a^i)\mathbf{F}(\widehat{\boldsymbol{\mu}}(\{a^k\}^N_{k=1}), a^i)\right| \\
&\leq \sum_{\substack{a^j \in \mathcal{A} \\ j \neq i}} \prod_{j \neq i} \boldsymbol{\pi}^j(a^j) \left|\sum_{a^i \in \mathcal{A}} \left[\boldsymbol{\pi}(a^i) - \boldsymbol{\pi}'(a^i)\right] \mathbf{F}(\widehat{\boldsymbol{\mu}}(\{a^k\}^N_{k=1}), a^i)\right| \\
&\leq \sum_{\substack{a^j \in \mathcal{A} \\ j \neq i}} \prod_{j \neq i} \boldsymbol{\pi}^j(a^j) \|\boldsymbol{\pi} - \boldsymbol{\pi}'\|_2 \sqrt{\sum_{a^i \in \mathcal{A}} \mathbf{F}(\widehat{\boldsymbol{\mu}}(\{a^k\}^N_{k=1}), a^i)^2} \\
&\leq \sum_{\substack{a^j \in \mathcal{A} \\ j \neq i}} \prod_{j \neq i} \boldsymbol{\pi}^j(a^j) \|\boldsymbol{\pi} - \boldsymbol{\pi}'\|_2 \sqrt{K} \\
&\leq \|\boldsymbol{\pi} - \boldsymbol{\pi}'\|_2 \sqrt{K}.
\end{aligned}
$$

where we use the Cauchy-Schwartz inequality.

Likewise, for any $k \neq i$, it holds that

$$
\begin{aligned}
&|V^i(\boldsymbol{\pi}, \boldsymbol{\pi}^{-k}) - V^i(\boldsymbol{\pi}', \boldsymbol{\pi}^{-k})| \\
&\leq \mathbb{E}\left[\mathbf{F}\left(\widehat{\boldsymbol{\mu}}(\{a^l\}^N_{l=1}), a^i\right)\big|a^j \sim \boldsymbol{\pi}^j, \forall j \neq k, a^k \sim \boldsymbol{\pi}\right] - \mathbb{E}\left[\mathbf{F}\left(\widehat{\boldsymbol{\mu}}(\{a^l\}^N_{l=1}), a^i\right)\big|a^j \sim \boldsymbol{\pi}^j, \forall j \neq k, a^k \sim \boldsymbol{\pi}'\right] \\
&\leq \left|\sum_{\substack{a^j \in \mathcal{A} \\ j \neq k}} \prod_{j \neq k} \boldsymbol{\pi}^j(a^j) \sum_{a^k \in \mathcal{A}} \boldsymbol{\pi}(a^k)\mathbf{F}(\widehat{\boldsymbol{\mu}}(\{a^l\}^N_{l=1}), a^i) - \sum_{\substack{a^j \in \mathcal{A} \\ j \neq k}} \prod_{j \neq k} \boldsymbol{\pi}^j(a^j) \sum_{a^k \in \mathcal{A}} \boldsymbol{\pi}'(a^k)\mathbf{F}(\widehat{\boldsymbol{\mu}}(\{a^l\}^N_{l=1}), a^i)\right| \\
&\leq \sum_{\substack{a^j \in \mathcal{A} \\ j \neq k}} \prod_{j \neq k} \boldsymbol{\pi}^j(a^j) \left|\sum_{a^k \in \mathcal{A}} \left[\boldsymbol{\pi}(a^k) - \boldsymbol{\pi}'(a^k)\right] \mathbf{F}(\widehat{\boldsymbol{\mu}}(\{a^l\}^N_{l=1}), a^i)\right|
\end{aligned}
$$

In this case, note that for any $a, a' \in \mathcal{A}, \mathbf{a} \in \mathcal{A}^K$, we have $|\mathbf{F}(\widehat{\boldsymbol{\mu}}(a, \mathbf{a}^{-k}), a^i) - \mathbf{F}(\widehat{\boldsymbol{\mu}}(a', \mathbf{a}^{-k}), a^i)| \leq \|\mathbf{F}(\widehat{\boldsymbol{\mu}}(a, \mathbf{a}^{-k})) - \mathbf{F}(\widehat{\boldsymbol{\mu}}(a', \mathbf{a}^{-k}))\|_2 \leq L\sqrt{2}/N$, hence there exists a constant $v_{-k} \in \mathbb{R}$ such that

$|\mathbf{F}(\widehat{\boldsymbol{\mu}}(a, \mathbf{a}^{-k}), a^i) - v_{-k}| \leq 2L\sqrt{2}/N$ for all $a$. Then,

$$|V^i(\boldsymbol{\pi}, \boldsymbol{\pi}^{-k}) - V^i(\boldsymbol{\pi}', \boldsymbol{\pi}^{-k})|$$

$$\leq \sum_{\substack{a^j \in \mathcal{A} \\ j \neq k}} \prod_{j \neq k} \boldsymbol{\pi}^j(a^j) \left| \sum_{a^k \in \mathcal{A}} \left[ \boldsymbol{\pi}(a^k) - \boldsymbol{\pi}'(a^k) \right] \left[ \mathbf{F}(\widehat{\boldsymbol{\mu}}(\{a^l\}_{l=1}^N), a^i) - v_{-k} \right] \right|$$

$$\leq \sum_{\substack{a^j \in \mathcal{A} \\ j \neq k}} \prod_{j \neq k} \boldsymbol{\pi}^j(a^j) \|\boldsymbol{\pi} - \boldsymbol{\pi}'\|_2 \sqrt{\sum_{a^k \in \mathcal{A}} \left[ \mathbf{F}(\widehat{\boldsymbol{\mu}}(\{a^l\}_{l=1}^N), a^i) - v_{-k} \right]^2}$$

$$\leq \sum_{\substack{a^j \in \mathcal{A} \\ j \neq k}} \prod_{j \neq k} \boldsymbol{\pi}^j(a^j) \|\boldsymbol{\pi} - \boldsymbol{\pi}'\|_2 2L\sqrt{2}\sqrt{K}/N$$

$$\leq \|\boldsymbol{\pi} - \boldsymbol{\pi}'\|_2 \frac{2L\sqrt{2}\sqrt{K}}{N}.$$

Finally, we establish the Lipschitz continuity of $\mathcal{E}_{\exp}^i$.

$$|\mathcal{E}_{\exp}^i(\boldsymbol{\pi}, \boldsymbol{\pi}^{-i}) - \mathcal{E}_{\exp}^i(\boldsymbol{\pi}', \boldsymbol{\pi}^{-i})|$$

$$\leq \left| \max_{\overline{\boldsymbol{\pi}} \in \Delta_{\mathcal{A}}} V^i(\overline{\boldsymbol{\pi}}, \boldsymbol{\pi}^{-i}) - V^i(\boldsymbol{\pi}, \boldsymbol{\pi}^{-i}) - \max_{\overline{\boldsymbol{\pi}} \in \Delta_{\mathcal{A}}} V^i(\overline{\boldsymbol{\pi}}, \boldsymbol{\pi}^{-i}) + V^i(\boldsymbol{\pi}', \boldsymbol{\pi}^{-i}) \right|$$

$$\leq \left| V^i(\boldsymbol{\pi}', \boldsymbol{\pi}^{-i}) - V^i(\boldsymbol{\pi}, \boldsymbol{\pi}^{-i}) \right|$$

$$\leq \sqrt{K} \|\boldsymbol{\pi} - \boldsymbol{\pi}'\|_2.$$

Similarly, for $k \neq i$,

$$|\mathcal{E}_{\exp}^i(\boldsymbol{\pi}, \boldsymbol{\pi}^{-k}) - \mathcal{E}_{\exp}^i(\boldsymbol{\pi}', \boldsymbol{\pi}^{-k})|$$

$$\leq \left| \max_{\overline{\boldsymbol{\pi}} \in \Delta_{\mathcal{A}}} V^i(\overline{\boldsymbol{\pi}}, \boldsymbol{\pi}, \boldsymbol{\pi}^{-k,i}) - V^i(\boldsymbol{\pi}, \boldsymbol{\pi}^{-k}) - \max_{\overline{\boldsymbol{\pi}} \in \Delta_{\mathcal{A}}} V^i(\overline{\boldsymbol{\pi}}, \boldsymbol{\pi}', \boldsymbol{\pi}^{-k,i}) + V^i(\boldsymbol{\pi}', \boldsymbol{\pi}^{-k}) \right|$$

$$\leq \max_{\overline{\boldsymbol{\pi}} \in \Delta_{\mathcal{A}}} \left| V^i(\overline{\boldsymbol{\pi}}, \boldsymbol{\pi}, \boldsymbol{\pi}^{-k,i}) - V^i(\overline{\boldsymbol{\pi}}, \boldsymbol{\pi}', \boldsymbol{\pi}^{-k,i}) \right| + \left| V^i(\boldsymbol{\pi}, \boldsymbol{\pi}^{-k}) - V^i(\boldsymbol{\pi}', \boldsymbol{\pi}^{-k}) \right|$$

$$\leq \frac{4L\sqrt{2K}}{N} \|\boldsymbol{\pi} - \boldsymbol{\pi}'\|_2.$$

$\square$

Finally, we prove the following statement that shows the infinite agent solution to MF-RVI is an approximate $N$-player Nash equilibrium.

**Lemma 7** *Let $\mathbf{F}$ be monotone, $L$-Lipschitz. Let $\boldsymbol{\pi}_\tau^* \in \Delta_{\mathcal{A}}$ be the (unique) MF-NE of the regularized map $\mathbf{F} - \tau \mathbf{I}$. Let $\boldsymbol{\pi}^1, \ldots, \boldsymbol{\pi}^N \in \Delta_{\mathcal{A}}$ be such that $\|\boldsymbol{\pi}^i - \boldsymbol{\pi}_\tau^*\|_2 \leq \delta$ for all $i$, then it holds that $\mathcal{E}_{exp}^i(\{\boldsymbol{\pi}^j\}_{j=1}^N) = \mathcal{O}(\tau + \delta + 1/\sqrt{N})$ for all $i \in \mathcal{N}$.*

**Proof:** By the Lipschitz continuity of exploitability (Lemma 6), we have

$$\mathcal{E}_{\exp}^i(\{\boldsymbol{\pi}^j\}_{j=1}^N) \leq \mathcal{E}_{\exp}^i(\{\boldsymbol{\pi}_\tau^*\}_{j=1}^N) + \sqrt{K} \|\boldsymbol{\pi}^i - \boldsymbol{\pi}_\tau^*\|_2 + \sum_{j \neq i} \frac{4L\sqrt{2K}}{N} \|\boldsymbol{\pi}^j - \boldsymbol{\pi}_\tau^*\|_2$$

$$\leq \mathcal{E}_{\exp}^i(\{\boldsymbol{\pi}_\tau^*\}_{j=1}^N) + \delta\sqrt{K} + 4L\sqrt{2K}\delta$$

The statement follows by using the results of Lemma 4 and Lemma 5. $\square$

### E.3 THE PROJECTED ASCENT OPERATOR

**Lemma 8** *Assume $\mathbf{r}$ is $\lambda$-monotone and $L$-Lipschitz. Then $\Gamma_{pg}^\eta$ is Lipschitz with constant $\sqrt{1 - 2\lambda\eta + \eta^2 L^2}$.*

**Proof:** The proof is similar to Theorem 12.1.2 of Facchinei & Pang (2003), page 1109.

$$
\begin{aligned}
\|\Gamma_{pg}(\boldsymbol{\mu}) - \Gamma_{pg}(\boldsymbol{\mu}')\|^2 &\leq \|\Pi_{\Delta_K}(\boldsymbol{\mu} + \eta\mathbf{r}(\boldsymbol{\mu})) - \Pi_{\Delta_K}(\boldsymbol{\mu}' + \eta\mathbf{r}(\boldsymbol{\mu}'))\|^2 \\
&\leq \|\boldsymbol{\mu} + \eta\mathbf{r}(\boldsymbol{\mu}) - \boldsymbol{\mu}' - \eta\mathbf{r}(\boldsymbol{\mu}')\|^2 \\
&\leq \|\boldsymbol{\mu} - \boldsymbol{\mu}'\|^2 + \eta^2\|\mathbf{r}(\boldsymbol{\mu}) - \mathbf{r}(\boldsymbol{\mu}')\|^2 + 2\eta(\mathbf{r}(\boldsymbol{\mu}) - \mathbf{r}(\boldsymbol{\mu}'))^\top(\boldsymbol{\mu} - \boldsymbol{\mu}') \\
&\leq (1 + \eta^2 L^2)\|\boldsymbol{\mu} - \boldsymbol{\mu}'\|^2 - 2\lambda\eta\|\boldsymbol{\mu} - \boldsymbol{\mu}'\|^2 \\
&\leq (1 - 2\lambda\eta + \eta^2 L^2)\|\boldsymbol{\mu} - \boldsymbol{\mu}'\|^2
\end{aligned}
$$

$\square$

**Corollary 3** *If $0 < \eta < \frac{2\lambda}{L^2}$, then $\Gamma_{pg}^\eta$ is a contraction. The smallest contraction modulus is achieved for $\eta = \frac{\lambda}{L^2}$, for which $\Gamma_{pg}^\eta$ is a $\sqrt{1 - \kappa^2}$-contraction, where $\kappa := \frac{\lambda}{L}$.*

**Proof:** Placing the values in Lemma 8 yields the statement. $\square$

# F    PROOFS OF MAIN THEORETICAL RESULTS

In this section, we present the full statements and the proofs of the main convergence theorems, in the case of expert feedback and bandit feedback.

## F.1    CONVERGENCE RESULT FOR EXPERT FEEDBACK

**Theorem 3 (Convergence - Expert feedback)** *Assume $\mathbf{F}$ is Lipschitz and monotone. Take the algorithm with expert feedback for each player $i$ so that*

$$
A_0^i := \mathrm{Unif}(\mathcal{A})
$$

$$
A_{t+1}^i(\ldots) := \Pi_{\Delta_K}((1 - \tau\eta_t)\boldsymbol{\pi}_t^i + \eta_t\mathbf{r}_t^i).
$$

*with learning rates $\eta_t := \frac{\tau^{-1}}{t+1}$. Furthermore, let $\boldsymbol{\pi}_\tau^*$ be the solution of the $\tau$-Tikhonov regularized problem. Then, for any $i \in \mathcal{N}$, it holds that*

$$
\mathbb{E}[\mathcal{E}_{exp}^i(\{\boldsymbol{\pi}_t^j\}_{j=1}^N)] \leq \max\{1, 4L\sqrt{2}\}\frac{4\tau^{-1}K\sqrt{1 + \sigma^2} + 4\tau^{-1}(L + \tau)\sqrt{K} + 12K\tau^{-2}L\sigma^2}{\sqrt{t}}
$$

$$
+ \max\{\sqrt{K}, 4L\sqrt{2K}\}\frac{24\tau^{-1}L + \|\boldsymbol{\pi}_0^i - \boldsymbol{\pi}_\tau^*\|_2^2}{\sqrt{t}}
$$

$$
+ \frac{2\tau^{-1}L\max\{\sqrt{K}, 4L\sqrt{2K}\}}{\sqrt{N}} + 2\tau + \frac{L(2\sqrt{2} + 4)}{N} + \frac{4L}{\sqrt{N}}.
$$

*Furthermore, if $\mathbf{F}$ is $\lambda > 0$ strongly monotone, it holds that*

$$
\mathbb{E}[\mathcal{E}_{exp}^i(\{\boldsymbol{\pi}_t^j\}_{j=1}^N)] \leq \max\{1, 4L\sqrt{2}\}\frac{3\tau^{-1}K\sqrt{1 + \sigma^2} + \tau^{-1}(L + \tau)\sqrt{6K} + 5\tau^{-3/2}LK\sigma\lambda^{-1/2}}{\sqrt{t}}
$$

$$
+ \max\{\sqrt{K}, 4L\sqrt{2K}\}\frac{9\tau^{-1/2}L\lambda^{-1/2} + \|\boldsymbol{\pi}_0^i - \boldsymbol{\pi}_\tau^*\|_2}{\sqrt{t}}
$$

$$
+ \frac{2\tau^{-1/2}L\lambda^{-1/2}\max\{\sqrt{K}, 4L\sqrt{2K}\}}{\sqrt{N}} + 2\tau + \frac{L(2\sqrt{2} + 4)}{N} + \frac{4L}{\sqrt{N}}.
$$

**Proof:** Define the sigma algebra $\mathcal{F}_T := \mathcal{F}(\{\boldsymbol{\pi}_t^i\}_{t=0,\ldots,T}^{i=1,\ldots,N})$. Also define the quantities and random variables

$$
\bar{\boldsymbol{\mu}}_t := \frac{1}{N}\sum_{i=1}^N \boldsymbol{\pi}_t^i,
$$

$$
\eta_t := \frac{\tau^{-1}}{t+1},
$$

$$
e_t^i := \|\boldsymbol{\pi}_t^i - \bar{\boldsymbol{\mu}}_t\|_2^2,
$$

$$
u_t^i := \mathbb{E}\left[\|\boldsymbol{\pi}_t^i - \boldsymbol{\pi}^*\|_2^2\right].
$$

Our goal naturally is to bound the sequence or error terms $u_t^i$. The strategy is as follows: (1) bound the expected differences of each agent's action probabilities $e_t^i$, showing the deviations of the policies of the agents go to zero, (2) obtain a non-linear recursion for the expectation of the terms $u_t^i$, (3) solve the recursion to obtain the convergence rate.

**Step 1: Bounding Policy Variations.** Firstly, we control the term $e_t^i$. Note that for any $i, j$, using the non-expansiveness of the projection operator, it holds that

$$
\begin{aligned}
\|\boldsymbol{\pi}_{t+1}^i - \boldsymbol{\pi}_{t+1}^j\|_2^2 =& \|\Pi((1-\tau\eta_t)\boldsymbol{\pi}_t^i + \eta_t \mathbf{r}_t^i) - \Pi((1-\tau\eta_t)\boldsymbol{\pi}_t^j + \eta_t \mathbf{r}_t^j)\|_2^2 \\
\leq& \|(1-\tau\eta_t)\boldsymbol{\pi}_t^i + \eta_t \mathbf{r}_t^i - (1-\tau\eta_t)\boldsymbol{\pi}_t^j - \eta_t \mathbf{r}_t^j\|_2^2 \\
\leq& \|(1-\tau\eta_t)(\boldsymbol{\pi}_t^i - \boldsymbol{\pi}_t^j) + \eta_t(\mathbf{r}_t^i - \mathbf{r}_t^j)\|_2^2 \\
=& (1-\tau\eta_t)^2\|\boldsymbol{\pi}_t^i - \boldsymbol{\pi}_t^j\|_2^2 + \eta_t^2\|\mathbf{r}_t^i - \mathbf{r}_t^j\|_2^2 + 2(1-\tau\eta_t)\eta_t(\boldsymbol{\pi}_t^i - \boldsymbol{\pi}_t^j)^\top(\mathbf{r}_t^i - \mathbf{r}_t^j)
\end{aligned}
$$

Taking the conditional expectation on both sides, we obtain

$$
\begin{aligned}
\mathbb{E}\left[\|\boldsymbol{\pi}_{t+1}^i - \boldsymbol{\pi}_{t+1}^j\|_2^2|\mathcal{F}_t\right] \leq& (1-\tau\eta_t)^2\|\boldsymbol{\pi}_t^i - \boldsymbol{\pi}_t^j\|_2^2 + \mathbb{E}\left[\eta_t^2\|\mathbf{r}_t^i - \mathbf{r}_t^j\|_2^2|\mathcal{F}_t\right] \\
& + 2(1-\tau\eta_t)\eta_t(\boldsymbol{\pi}_t^i - \boldsymbol{\pi}_t^j)^\top \mathbb{E}\left[\mathbf{r}_t^i - \mathbf{r}_t^j|\mathcal{F}_t\right] \\
=& (1-\tau\eta_t)^2\|\boldsymbol{\pi}_t^i - \boldsymbol{\pi}_t^j\|_2^2 + \mathbb{E}\left[\eta_t^2\|\mathbf{r}_t^i - \mathbf{r}_t^j\|_2^2|\mathcal{F}_t\right] \\
=& (1-\tau\eta_t)^2\|\boldsymbol{\pi}_t^i - \boldsymbol{\pi}_t^j\|_2^2 + \eta_t^2\mathbb{E}\left[\|\varepsilon_t^i - \varepsilon_t^j\|_2^2|\mathcal{F}_t\right] \\
=& (1-\tau\eta_t)^2\|\boldsymbol{\pi}_t^i - \boldsymbol{\pi}_t^j\|_2^2 + \eta_t^2 K\sigma^2
\end{aligned}
$$

since we have $\mathbf{r}_t^i := \mathbf{F}(\widehat{\boldsymbol{\mu}}_t) + \varepsilon_t^i$ and $K = |\mathcal{A}|$. Then, it holds that

$$
\begin{aligned}
\mathbb{E}\left[\|\boldsymbol{\pi}_{t+1}^i - \boldsymbol{\pi}_{t+1}^j\|_2^2\right] \leq& (1-\tau\eta_t)^2\mathbb{E}\left[\|\boldsymbol{\pi}_t^i - \boldsymbol{\pi}_t^j\|_2^2\right] + \eta_t^2 K\sigma^2 \\
\leq& \left(1 - \frac{1}{t+1}\right)^2\mathbb{E}\left[\|\boldsymbol{\pi}_t^i - \boldsymbol{\pi}_t^j\|_2^2\right] + \left(\frac{\tau^{-1}}{t+1}\right)^2 K\sigma^2 \\
\leq& \left(1 - \frac{2}{t+1}\right)\mathbb{E}\left[\|\boldsymbol{\pi}_t^i - \boldsymbol{\pi}_t^j\|_2^2\right] + \frac{1}{(t+1)^2}\mathbb{E}\left[\|\boldsymbol{\pi}_t^i - \boldsymbol{\pi}_t^j\|_2^2\right] + \frac{\tau^{-2}K\sigma^2}{(t+1)^2} \\
\leq& \left(1 - \frac{2}{t+1}\right)\mathbb{E}\left[\|\boldsymbol{\pi}_t^i - \boldsymbol{\pi}_t^j\|_2^2\right] + \frac{1}{(t+1)^2}\mathbb{E}\left[2\|\boldsymbol{\pi}_t^i\|_2^2 + 2\|\boldsymbol{\pi}_t^j\|_2^2\right] + \frac{\tau^{-2}K\sigma^2}{(t+1)^2} \\
\leq& \left(1 - \frac{2}{t+1}\right)\mathbb{E}\left[\|\boldsymbol{\pi}_t^i - \boldsymbol{\pi}_t^j\|_2^2\right] + \frac{\tau^{-2}K\sigma^2 + 4}{(t+1)^2}
\end{aligned}
$$

To bound the recurrence, we can use Lemma 2 (noting $\gamma = 2, u_0 = 0, c_0 = 0, c_1 = \tau^{-2}K\sigma^2 + 4$).

$$
\mathbb{E}\left[\|\boldsymbol{\pi}_{t+1}^i - \boldsymbol{\pi}_{t+1}^j\|_2^2\right] \leq \frac{\tau^{-2}K\sigma^2 + 4}{(t+1)^2} + \frac{\tau^{-2}K\sigma^2 + 4}{t+1} + \frac{\tau^{-2}K\sigma^2 + 4}{(t+1)^2} \leq \frac{3\tau^{-2}K\sigma^2 + 12}{(t+1)}.
$$

Then, the expected values of $e_t^i$ can be bounded using:

$$
e_t^i = \|\boldsymbol{\pi}_t^i - \bar{\boldsymbol{\mu}}_t\|_2^2 = \left\|\boldsymbol{\pi}_t^i - \frac{1}{N}\sum_{j=1}^N \boldsymbol{\pi}_t^j\right\|_2^2 \leq \frac{1}{N}\sum_{j=1}^N \|\boldsymbol{\pi}_t^i - \boldsymbol{\pi}_t^j\|_2^2
$$

by an application of Jensen's inequality. Then we have $\mathbb{E}\left[e_t^i\right] \leq \frac{3\tau^{-2}K\sigma^2 + 12}{t+1}$.

**Step 2: Formulating the main recurrence.** Next, we analyze for any $i$, the error term $\|\boldsymbol{\pi}_t^i - \boldsymbol{\pi}^*\|_2^2$. We denote $\alpha_t := (1-\tau\eta_t)$. We note that for the regularized solution $\boldsymbol{\pi}^*$, we have the fixed point result

$$
\Pi((1-\tau\eta_t)\boldsymbol{\pi}^* + \eta_t\mathbf{F}(\boldsymbol{\pi}^*)) = \Pi(\boldsymbol{\pi}^* + \eta_t(\mathbf{F} - \tau\mathbf{I})(\boldsymbol{\pi}^*)) = \boldsymbol{\pi}^*.
$$

Hence, we can bound the quantity $\|\boldsymbol{\mu}_{t+1}^i - \boldsymbol{\pi}^*\|_2^2$ by:

$$
\begin{aligned}
\|\boldsymbol{\pi}_{t+1}^i - \boldsymbol{\pi}^*\|_2^2 =& \|\Pi(\alpha_t \boldsymbol{\pi}_t^i + \eta_t \mathbf{r}_t^i) - \Pi(\alpha_t \boldsymbol{\pi}^* + \eta_t \mathbf{F}(\boldsymbol{\pi}^*))\|_2^2 \\
\leq & \|\alpha_t \boldsymbol{\pi}_t^i + \eta_t \mathbf{r}_t^i - \alpha_t \boldsymbol{\pi}^* - \eta_t \mathbf{F}(\boldsymbol{\pi}^*)\|_2^2 \\
\leq & \|\alpha_t \boldsymbol{\pi}_t^i + \eta_t \mathbf{F}(\boldsymbol{\pi}_t^i) - \alpha_t \boldsymbol{\pi}^* - \eta_t \mathbf{F}(\boldsymbol{\pi}^*) + \eta_t(\mathbf{r}_t^i - \mathbf{F}(\boldsymbol{\pi}_t^i))\|_2^2 \\
=& \eta_t^2 \|\mathbf{r}_t^i - \mathbf{F}(\boldsymbol{\pi}_t^i)\|_2^2 + 2\eta_t (\alpha_t(\boldsymbol{\pi}_t^i - \boldsymbol{\pi}^*) + \eta_t(\mathbf{F}(\boldsymbol{\pi}_t^i) - \mathbf{F}(\boldsymbol{\pi}^*)))^\top (\mathbf{r}_t^i - \mathbf{F}(\boldsymbol{\pi}_t^i)) \\
& + \|\alpha_t(\boldsymbol{\pi}_t^i - \boldsymbol{\pi}^*) + \eta_t(\mathbf{F}(\boldsymbol{\pi}_t^i) - \mathbf{F}(\boldsymbol{\pi}^*))\|_2^2 \\
=& \eta_t^2 \|\mathbf{r}_t^i - \mathbf{F}(\boldsymbol{\pi}_t^i)\|_2^2 + 2\eta_t^2(\mathbf{F}(\boldsymbol{\pi}_t^i) - \mathbf{F}(\boldsymbol{\pi}^*))^\top (\mathbf{r}_t^i - \mathbf{F}(\boldsymbol{\pi}_t^i)) + 2\eta_t \alpha_t(\boldsymbol{\pi}_t^i - \boldsymbol{\pi}^*)^\top (\mathbf{r}_t^i - \mathbf{F}(\boldsymbol{\pi}_t^i)) \\
& + \|\alpha_t(\boldsymbol{\pi}_t^i - \boldsymbol{\pi}^*) + \eta_t(\mathbf{F}(\boldsymbol{\pi}_t^i) - \mathbf{F}(\boldsymbol{\pi}^*))\|_2^2 \\
\leq & \underbrace{\eta_t^2 \|\mathbf{r}_t^i - \mathbf{F}(\boldsymbol{\pi}_t^i)\|_2^2 + 2\eta_t^2(\mathbf{F}(\boldsymbol{\pi}_t^i) - \mathbf{F}(\boldsymbol{\pi}^*))^\top (\mathbf{r}_t^i - \mathbf{F}(\boldsymbol{\pi}_t^i))}_{(a)} \\
& + \underbrace{2\eta_t \alpha_t(\boldsymbol{\pi}_t^i - \boldsymbol{\pi}^*)^\top (\mathbf{r}_t^i - \mathbf{F}(\boldsymbol{\pi}_t^i))}_{(b)} + \underbrace{\|\alpha_t(\boldsymbol{\pi}_t^i - \boldsymbol{\pi}^*) + \eta_t(\mathbf{F}(\boldsymbol{\pi}_t^i) - \mathbf{F}(\boldsymbol{\pi}^*))\|_2^2}_{(c)}.
\end{aligned}
$$

We analyze the three marked terms separately. For term $(a)$, using the independence assumption of the noise vectors and Young's inequality, in expectation we obtain

$$
\begin{aligned}
\mathbb{E}[(a)] \leq & \eta_t^2 \, \mathbb{E}[\|\mathbf{r}_t^i - \mathbf{F}(\boldsymbol{\pi}_t^i)\|_2^2] + \eta_t^2 \, \mathbb{E}[\|\mathbf{F}(\boldsymbol{\pi}_t^i) - \mathbf{F}(\boldsymbol{\pi}^*)\|_2^2] + \eta_t^2 \, \mathbb{E}[\|\mathbf{r}_t^i - \mathbf{F}(\boldsymbol{\pi}_t^i)\|_2^2] \\
\leq & 2\eta_t^2 \, \mathbb{E}[\|\mathbf{r}_t^i - \mathbf{F}(\boldsymbol{\pi}_t^i)\|_2^2] + \eta_t^2 \, \mathbb{E}[\|\mathbf{F}(\boldsymbol{\pi}_t^i) - \mathbf{F}(\boldsymbol{\pi}^*)\|_2^2] \\
\leq & 2\eta_t^2 \, \mathbb{E}[\|\mathbf{r}_t^i - \mathbf{F}(\widehat{\boldsymbol{\mu}}_t)\|_2^2 + \|\mathbf{F}(\widehat{\boldsymbol{\mu}}_t) - \mathbf{F}(\boldsymbol{\pi}_t^i)\|_2^2] + \eta_t^2 K \\
\leq & 2\eta_t^2 \sigma^2 K + 2\eta_t^2 K = 2\eta_t^2 K(\sigma^2 + 1)
\end{aligned}
$$

For the term $(c)$, we obtain

$$
\begin{aligned}
(c) =& \|\alpha_t(\boldsymbol{\pi}_t^i - \boldsymbol{\pi}^*) + \eta_t(\mathbf{F}(\boldsymbol{\pi}_t^i) - \mathbf{F}(\boldsymbol{\pi}^*))\|_2^2 \\
=& \|(\boldsymbol{\pi}_t^i - \boldsymbol{\pi}^*) + \eta_t(\mathbf{F}(\boldsymbol{\pi}_t^i) - \tau\boldsymbol{\pi}_t^i - \mathbf{F}(\boldsymbol{\pi}^*) + \tau\boldsymbol{\pi}^*)\|_2^2 \\
\leq & \left(1 - 2(\lambda + \tau)\eta_t + (L + \tau)^2 \eta_t^2\right) \|\boldsymbol{\pi}_t^i - \boldsymbol{\pi}^*\|_2^2 \\
\leq & \left(1 - 2(\lambda + \tau)\eta_t\right) \|\boldsymbol{\pi}_t^i - \boldsymbol{\pi}^*\|_2^2 + 2(L + \tau)^2 \eta_t^2
\end{aligned}
$$

where the last inequality holds from the contractivity result in Lemma 8.

For the term $(b)$, first taking the strongly monotone problem $\lambda > 0$, we have that

$$
\begin{aligned}
(b) =& 2\eta_t \alpha_t(\boldsymbol{\pi}_t^i - \boldsymbol{\pi}^*)^\top (\mathbf{r}_t^i - \mathbf{F}(\boldsymbol{\pi}_t^i)) \\
=& 2\eta_t \alpha_t(\boldsymbol{\pi}_t^i - \boldsymbol{\pi}^*)^\top (\mathbf{r}_t^i - \mathbf{F}(\widehat{\boldsymbol{\mu}}_t)) + 2\eta_t \alpha_t(\boldsymbol{\pi}_t^i - \boldsymbol{\pi}^*)^\top (\mathbf{F}(\widehat{\boldsymbol{\mu}}_t) - \mathbf{F}(\bar{\boldsymbol{\mu}}_t)) \\
& + 2\eta_t \alpha_t(\boldsymbol{\pi}_t^i - \boldsymbol{\pi}^*)^\top (\mathbf{F}(\bar{\boldsymbol{\mu}}_t) - \mathbf{F}(\boldsymbol{\pi}_t^i)) \\
\leq & 2\eta_t \alpha_t \left(\frac{\lambda}{4}\|\boldsymbol{\pi}_t^i - \boldsymbol{\pi}^*\|_2^2 + \frac{1}{\lambda}\|\mathbf{F}(\widehat{\boldsymbol{\mu}}_t) - \mathbf{F}(\bar{\boldsymbol{\mu}}_t)\|_2^2\right) + 2\eta_t \alpha_t \left(\frac{\lambda}{4}\|\boldsymbol{\pi}_t^i - \boldsymbol{\pi}^*\|_2^2 + \frac{1}{\lambda}\|\mathbf{F}(\bar{\boldsymbol{\mu}}_t) - \mathbf{F}(\boldsymbol{\pi}_t^i)\|_2^2\right) + \\
& 2\eta_t \alpha_t(\boldsymbol{\pi}_t^i - \boldsymbol{\pi}^*)^\top (\mathbf{r}_t^i - \mathbf{F}(\widehat{\boldsymbol{\mu}}_t)) \\
\leq & 2\eta_t \frac{\lambda}{2}\|\boldsymbol{\pi}_t^i - \boldsymbol{\pi}^*\|_2^2 + \frac{2\eta_t}{\lambda}\|\mathbf{F}(\widehat{\boldsymbol{\mu}}_t) - \mathbf{F}(\bar{\boldsymbol{\mu}}_t)\|_2^2 + \frac{2\eta_t}{\lambda}\|\mathbf{F}(\bar{\boldsymbol{\mu}}_t) - \mathbf{F}(\boldsymbol{\pi}_t^i)\|_2^2 \\
& + 2\eta_t \alpha_t(\boldsymbol{\pi}_t^i - \boldsymbol{\pi}^*)^\top (\mathbf{r}_t^i - \mathbf{F}(\widehat{\boldsymbol{\mu}}_t)),
\end{aligned}
$$

which follows from an application of Young's inequality. For the last three terms, we have the bounds in the conditional expectation:

$$
\mathbb{E}\left[2\eta_t \alpha_t(\boldsymbol{\pi}_t^i - \boldsymbol{\pi}^*)^\top (\mathbf{r}_t^i - \mathbf{F}(\widehat{\boldsymbol{\mu}}_t)) | \mathcal{F}_t\right] = 0
$$

$$\mathbb{E}[\|\mathbf{F}(\widehat{\boldsymbol{\mu}}_t) - \mathbf{F}(\bar{\boldsymbol{\mu}}_t)\|_2^2|\mathcal{F}_t] \leq L^2 \, \mathbb{E}\left[\|\widehat{\boldsymbol{\mu}}_t - \bar{\boldsymbol{\mu}}_t\|_2^2|\mathcal{F}_t\right]$$

$$\leq L^2 \, \mathbb{E}\left[\frac{1}{N^2}\|\sum_i \boldsymbol{\pi}_t^i - \sum_i \mathbf{e}_{a_t^i}\|_2^2|\mathcal{F}_t\right]$$

$$= \frac{L^2}{N^2}\sum_i \mathbb{E}[\|\boldsymbol{\pi}_t^i - \mathbf{e}_{a_t^i}\|_2^2|\mathcal{F}_t]$$

$$\leq \frac{2L^2}{N}$$

$$\|\mathbf{F}(\bar{\boldsymbol{\mu}}_t) - \mathbf{F}(\boldsymbol{\pi}_t^i)\|_2^2 \leq L^2\|\bar{\boldsymbol{\mu}}_t - \boldsymbol{\pi}_t^i\|_2^2 = L^2 e_t^i$$

Noting that $\widehat{\boldsymbol{\mu}}_t$ is the sum of $N$ independent random variables and has expectation $\bar{\boldsymbol{\mu}}_t$.

Hence, putting in the bounds for $(a), (b), (c)$ and taking expectations, we obtain the inequality

$$\mathbb{E}\left[\|\boldsymbol{\pi}_{t+1}^i - \boldsymbol{\pi}^*\|_2^2\right] \leq 2\eta_t^2 K(1+\sigma^2) + \frac{4\eta_t L^2}{\lambda N} + \frac{2\eta_t L^2}{\lambda}\mathbb{E}\left[e_t^i\right]$$

$$+ \left(1 - 2(\frac{\lambda}{2}+\tau)\eta_t\right)\mathbb{E}\left[\|\boldsymbol{\pi}_t^i - \boldsymbol{\pi}^*\|_2^2\right] + 2(L+\tau)^2\eta_t^2$$

or equivalently, the recursion

$$u_{t+1}^i \leq \frac{2\tau^{-2}K(1+\sigma^2) + 2\tau^{-2}(L+\tau)^2}{(t+1)^2} + \frac{4\tau^{-1}L^2\lambda^{-1}}{N(t+1)} + \frac{2\tau^{-1}L^2\lambda^{-1}}{(t+1)}\mathbb{E}\left[e_t^i\right] + \left(1 - \frac{2\tau^{-1}(\lambda/2+\tau)}{t+1}\right)u_t^i.$$

However, if $\lambda = 0$, we bound the term $(a)$ as follows. Take any arbitrary $1 > \delta > 0$. Then, once again applying Young's inequality, we obtain

$$(a) = 2\eta_t\alpha_t(\boldsymbol{\pi}_t^i - \boldsymbol{\pi}^*)^\top(\mathbf{r}_t^i - \mathbf{F}(\boldsymbol{\pi}_t^i))$$

$$= 2\eta_t\alpha_t(\boldsymbol{\pi}_t^i - \boldsymbol{\pi}^*)^\top(\mathbf{r}_t^i - \mathbf{F}(\widehat{\boldsymbol{\mu}}_t)) + 2\eta_t\alpha_t(\boldsymbol{\pi}_t^i - \boldsymbol{\pi}^*)^\top(\mathbf{F}(\widehat{\boldsymbol{\mu}}_t) - \mathbf{F}(\bar{\boldsymbol{\mu}}_t))$$

$$+ 2\eta_t\alpha_t(\boldsymbol{\pi}_t^i - \boldsymbol{\pi}^*)^\top(\mathbf{F}(\bar{\boldsymbol{\mu}}_t) - \mathbf{F}(\boldsymbol{\pi}_t^i))$$

$$\leq 2\eta_t\alpha_t\left(\frac{\tau\delta}{2}\|\boldsymbol{\pi}_t^i - \boldsymbol{\pi}^*\|_2^2 + \frac{1}{2\tau\delta}\|\mathbf{r}(\widehat{\boldsymbol{\mu}}_t) - \mathbf{r}(\bar{\boldsymbol{\mu}}_t)\|_2^2\right) + 2\eta_t\alpha_t\left(\frac{\tau\delta}{2}\|\boldsymbol{\pi}_t^i - \boldsymbol{\pi}^*\|_2^2 + \frac{1}{2\tau\delta}\|\mathbf{r}(\bar{\boldsymbol{\mu}}_t) - \mathbf{r}(\boldsymbol{\pi}_t^i)\|_2^2\right) +$$

$$2\eta_t\alpha_t(\boldsymbol{\pi}_t^i - \boldsymbol{\pi}^*)^\top(\mathbf{r}_t^i - \mathbf{F}(\widehat{\boldsymbol{\mu}}_t))$$

$$\leq 2\eta_t\tau\delta\|\boldsymbol{\pi}_t^i - \boldsymbol{\pi}^*\|_2^2 + \frac{\eta_t}{\tau\delta}\|\mathbf{F}(\widehat{\boldsymbol{\mu}}_t) - \mathbf{F}(\bar{\boldsymbol{\mu}}_t)\|_2^2 + \frac{\eta_t}{\tau\delta}\|\mathbf{F}(\bar{\boldsymbol{\mu}}_t) - \mathbf{F}(\boldsymbol{\pi}_t^i)\|_2^2$$

$$+ 2\eta_t\alpha_t(\boldsymbol{\pi}_t^i - \boldsymbol{\pi}^*)^\top(\mathbf{r}_t^i - \mathbf{F}(\widehat{\boldsymbol{\mu}}_t)).$$

Then, once again repeating the analysis before, we obtain the bound

$$u_{t+1}^i \leq \frac{2\tau^{-2}K(1+\sigma^2) + 2\tau^{-2}(L+\tau)^2}{(t+1)^2} + \frac{4\tau^{-2}L^2\delta^{-1}}{N(t+1)} + \frac{\tau^{-2}L^2\delta^{-1}}{(t+1)}\mathbb{E}\left[e_t^i\right] + \left(1 - \frac{2(1-\delta)}{t+1}\right)u_t^i.$$

**Step 3: Solving the recurrence.** Note that the exploitability in the main statement of the theorem can be related to $u_t^i$ as follows using Lemma 6:

$$\mathbb{E}[\mathcal{E}_{\exp}^i(\{\boldsymbol{\pi}_t^j\}_{j=1}^N)] \leq \mathcal{E}_{\exp}^i(\{\boldsymbol{\pi}^*\}_{j=1}^N) + \sqrt{K}\,\mathbb{E}[\|\boldsymbol{\pi}_t^i - \boldsymbol{\pi}^*\|_2] + \frac{4L\sqrt{2K}}{N}\sum_{j\neq i}\mathbb{E}[\|\boldsymbol{\pi}_t^j - \boldsymbol{\pi}^*\|_2]$$

$$\leq \mathcal{E}_{\exp}^i(\{\boldsymbol{\pi}^*\}_{j=1}^N) + \sqrt{K}\sqrt{u_t^i} + \frac{4L\sqrt{2K}}{N}\sum_{j\neq i}\sqrt{u_t^j}$$

$$\leq \mathcal{E}_{\exp}^i(\{\boldsymbol{\pi}^*\}_{j=1}^N) + \frac{\max\{\sqrt{K}, 4L\sqrt{2K}\}}{N}\sum_j\sqrt{u_t^j}$$

Hence the bounds on $u_t^j$ will yield the result of the theorem by linearity of expectation, along with an invocation of Proposition 2 and Lemma 5.

Finally, we solve the recurrences for $\lambda = 0$ and $\lambda > 0$. For the case $\lambda > 0$, placing the shown before $\mathbb{E}\left[e_t^i\right] \leq \frac{3\tau^{-2}K\sigma^2 + 12}{t+1}$, we obtain

$$
\begin{aligned}
u_{t+1}^i \leq & \frac{2\tau^{-2}K(1+\sigma^2) + 2\tau^{-2}(L+\tau)^2 + 6\tau^{-3}K\sigma^2\lambda^{-1}L^2 + 24\tau^{-1}L^2\lambda^{-1}}{(t+1)^2} \\
& + \frac{4\tau^{-1}L^2\lambda^{-1}}{N(t+1)} + \left(1 - \frac{2\tau^{-1}(\lambda/2 + \tau)}{t+1}\right)u_t^i.
\end{aligned}
$$

An invocation of Lemma 2 proves the main statement of the theorem.

For the monotone case $\lambda = 0$, we have the recursion:

$$
\begin{aligned}
u_{t+1}^i \leq & \frac{2\tau^{-2}K(1+\sigma^2) + 2\tau^{-2}(L+\tau)^2 + 6K\tau^{-4}L^2\delta^{-1}\sigma^2 + 24\tau^{-2}L^2\delta^{-1}}{(t+1)^2} \\
& + \frac{2\tau^{-2}L^2\delta^{-1}}{N(t+1)} + \left(1 - \frac{2(1-\delta)}{t+1}\right)u_t^i.
\end{aligned}
$$

Another invocation of Lemma 2 concludes the proof, choosing $\delta = 1/4$. $\qquad\square$

### F.2 CONVERGENCE RESULT FOR BANDIT FEEDBACK

---

**Algorithm 1** TRPA-Bandit: Independent learning with bandit feedback algorithm for an agent.

---

**Require:** Number of actions $K$, regularization $\tau > 0$, exploration probability $\varepsilon > 0$, number of epochs $H$.

$\quad \boldsymbol{\pi}_0 \leftarrow \frac{1}{K}\mathbf{1}$

$\quad$**for** $h = 0, \ldots, H-1$ **do**

$\qquad \widehat{\mathbf{r}}_h \leftarrow \mathbf{0}$

$\qquad T_h \leftarrow \lceil \varepsilon^{-1}\log(h+1)\rceil$

$\qquad$**for** $t = 1, \ldots, T_h$ **do** $\qquad\qquad\qquad\qquad\qquad$ ▷ Exploration for $T_h$ rounds before policy update,

$\qquad\qquad$ Sample Bernoulli r.v. $u \sim \mathrm{Ber}(\varepsilon)$.

$\qquad\qquad$**if** $u = 1$ **then**

$\qquad\qquad\qquad$ Play action $a_{h,t} \sim \mathrm{Unif}(\mathcal{A})$ uniformly at random. $\quad$ ▷ playing uniformly with prob. $\varepsilon$,

$\qquad\qquad\qquad$ Observe payoff $r_{h,t}$.

$\qquad\qquad\qquad \widehat{\mathbf{r}}_h \leftarrow K r_{h,t}\mathbf{e}_{a_{h,t}}$.

$\qquad\qquad$**else if** $u = 0$ **then**

$\qquad\qquad\qquad$ Play action with current policy $a_{h,t} \sim \boldsymbol{\pi}_h$. $\qquad$ ▷ playing the current policy otherwise.

$\qquad\qquad$**end if**

$\qquad$**end for**

$\qquad \eta_h = \frac{\tau^{-1}}{h+1}$.

$\qquad \boldsymbol{\pi}_{h+1} = \Pi_{\Delta_K}((1-\tau\eta_h)\boldsymbol{\pi}_h + \eta_h\widehat{r}_h)$ $\qquad\qquad\qquad$ ▷ After each epoch, update policy.

$\quad$**end for**

$\quad$Return $\boldsymbol{\pi}_H$

---

**Theorem 4 (Convergence - Bandit feedback)** *Assume $\mathbf{F}$ is $L$-Lipschitz and monotone, and let $\boldsymbol{\pi}_\tau^*$ be the solution of the $\tau$-Tikhonov regularized problem. Assume that for $H$ rounds, each player runs Algorithm 1 with learning rates and epoch durations*

$$
\eta_h := \frac{\tau^{-1}}{(h+1)}, T_h := \varepsilon^{-1}\log(h+1),
$$

*producing policies $\{\boldsymbol{\pi}_h^j\}_{j=1}^N$ at each epoch $h \in 0, \ldots, H-1$. Then for each player $i$, it holds that*

$$\mathbb{E}[\mathcal{E}_{exp}^i(\{\boldsymbol{\pi}_h^j\}_{j=1}^N)] \leq \frac{12\sqrt{1+\sigma^2}K^{3/2}\tau^{-1}(1+\tau^{-1}L)+6\tau^{-1}(L+\tau)}{\sqrt{h}}\max\{\sqrt{K}, 4L\sqrt{2K}\}$$

$$+ \frac{8K^{3/4}\tau^{-1/2}(1+\sigma^{1/2})+15\tau^{-1}L(\tau^{-1/2}+1)+\|\boldsymbol{\pi}_0^i - \boldsymbol{\pi}_\tau^*\|_2}{\sqrt{h}}\max\{\sqrt{K}, 4L\sqrt{2K}\}$$

$$+ \left(23\tau^{-1}LN^{-1/2}+8\tau^{-1}L\varepsilon+8\tau^{-3/2}LN^{-1}\right)\max\{\sqrt{K}, 4L\sqrt{2K}\}$$

$$+ 2\tau + \frac{L(2\sqrt{2}+4)}{N} + \frac{4L}{\sqrt{N}}.$$

*Furthermore, if it holds that $\mathbf{F}$ is $\lambda$-strongly monotone,*

$$\mathbb{E}[\mathcal{E}_{exp}^i(\{\boldsymbol{\pi}_h^j\}_{j=1}^N)] \leq \frac{7\tau^{-1}K^{3/2}(\sigma+1)(1+\tau^{-1/2}\lambda^{-1/2}L)+17\tau^{-1}\lambda^{-1/2}(\tau^{-1/2}+1)}{\sqrt{h}}\max\{\sqrt{K}, 4L\sqrt{2K}\}$$

$$+ \frac{5\tau^{-1}(L+\tau)+5K^{3/4}\tau^{-1/2}\sqrt{1+\sigma}+\|\boldsymbol{\pi}_0^i - \boldsymbol{\pi}_\tau^*\|_2}{\sqrt{h}}\max\{\sqrt{K}, 4L\sqrt{2K}\}$$

$$+ \left(23\tau^{-1/2}\lambda^{-1/2}LN^{-1/2}+8\tau^{-1/2}\lambda^{-1/2}L\varepsilon+6\tau^{-1}\lambda^{-1/2}LN^{-1}\right)\max\{\sqrt{K}, 4L\sqrt{2K}\}$$

$$+ 2\tau + \frac{L(2\sqrt{2}+4)}{N} + \frac{4L}{\sqrt{N}}.$$

**Proof:** Our analysis follows that in the case of expert feedback, the difference being the errors are analyzed per epoch and randomization in the exploration probabilities. We introduce the following indicator random variables:

$$\mathbb{1}_{h,t}^i := \mathbb{1}\{\text{player } i \text{ explores at round } t \text{ of epoch } h\}$$

$$E_{h,t}^i := \{\mathbb{1}_{h,t}^i = 1\}$$

$$\mathbb{1}_h^i := \mathbb{1}\{\text{player } i \text{ explores at least once during epoch } h\} = \max_{t=1,\ldots,T_h} \mathbb{1}_{h,t}^i$$

$$E_h^i := \{\mathbb{1}_h^i = 1\} = \bigcup_{t=1}^{T_h} E_{h,t}^i$$

$$a_h^i := \text{Last explored action in epoch } h \text{ by agent } i,$$
$$\quad \text{and } a_0 \text{ if no exploration occurred.}$$

$$s_h^i := \text{Timestep when exploration last occurred in epoch } h \text{ by agent } i,$$
$$\quad \text{and } 0 \text{ if no exploration occurred.} \in \{1, \ldots, T_h\}$$

Note that $\mathbb{1}_{h,t}^i$ are independent random variables for different $(i, h, t)$ triplets, likewise for $i \neq j$, $\mathbb{1}_{h,t}^i$ and $\mathbb{1}_h^j$ are pairwise independent. Once again, we define the following sigma algebra $\mathcal{F}_h := \mathcal{F}(\{\boldsymbol{\pi}_{h'}^i\}_{h=0,\ldots,h}^{i=1,\ldots,N})$, and the random variables:

$$\bar{\boldsymbol{\mu}}_h := \frac{1}{N}\sum_{i=1}^N \boldsymbol{\pi}_h^i,$$

$$\eta_h := \frac{\tau^{-1}}{h+1}$$

$$e_t^i := \|\boldsymbol{\pi}_h^i - \bar{\boldsymbol{\mu}}_h\|_2^2,$$

$$u_h^i := \mathbb{E}\left[\|\boldsymbol{\pi}_h^i - \boldsymbol{\pi}^*\|_2^2\right].$$

We again proceed in several steps.

**Step 1: The importance sampling estimate.** By the definition of the above events and the probabilistic exploration scheme, note that we have

$$\widehat{\mathbf{r}}_h^i = K\left(\mathbf{F}(\widehat{\boldsymbol{\mu}}_{s_h^i,h}, a_h^i) + \mathbf{n}_{t,h}^i(a_h^i)\right)\mathbb{1}_h^i.$$

Firstly, we note that by law of total expectations and the fact that $E_h^i$ are independent from $\mathcal{F}_h$,

$$\mathbb{E}\left[\widehat{\mathbf{r}}_h^i \big| \mathcal{F}_h\right] = \mathbb{E}\left[\widehat{\mathbf{r}}_h^i \big| E_h^i, \mathcal{F}_h\right] \mathbb{P}(E_h^i) + \mathbb{E}\left[\widehat{\mathbf{r}}_h^i \Big| \overline{E_h^i}, \mathcal{F}_h\right] \mathbb{P}(\overline{E_h^i})$$

$$= \mathbb{E}\left[\widehat{\mathbf{r}}_h^i \big| E_h^i, \mathcal{F}_h\right] - \underbrace{\mathbb{E}\left[\widehat{\mathbf{r}}_h^i \big| E_h^i, \mathcal{F}_h\right] \mathbb{P}(\overline{E_h^i}) + \mathbb{E}\left[\widehat{\mathbf{r}}_h^i \Big| \overline{E_h^i}, \mathcal{F}_h\right] \mathbb{P}(\overline{E_h^i})}_{:=\mathbf{b}_h^i}$$

$$= \mathbb{E}\left[\widehat{\mathbf{r}}_h^i \big| E_h^i, \mathcal{F}_h\right] + \mathbf{b}_h^i$$

for $\mathbf{b}_h^i$ quantifying a bias induced due to the probability of no exploration. We have that

$$\|\mathbf{b}_h^i\|_2 \le K\sqrt{K}\sqrt{1+\sigma^2}\exp\left\{-\varepsilon T_h\right\}$$

since $\mathbb{E}\left[\widehat{\mathbf{r}}_h^i \Big| \overline{E_h^i}, \mathcal{F}_h\right] = 0$ and exploration probabilities are determined by independent random Bernoulli variables hence

$$\mathbb{P}(\overline{E_h^i}) = (1-\varepsilon)^{T_h} \le \exp\left\{-\varepsilon T_h\right\}.$$

To further characterize the bias, we introduce a coupling argument. Define independent random variables $\bar{a}_h^j \sim \varepsilon\boldsymbol{\pi}_{\text{unif}} + (1-\varepsilon)\boldsymbol{\pi}_h^j$ for all $j \in \mathcal{N}$ and $\bar{a}_{h,exp}^i \sim \boldsymbol{\pi}_{\text{unif}}$. By the definition, it holds that (where $\widehat{\boldsymbol{\mu}}(\{\bar{a}_h^j\}_j) \in \Delta_{\mathcal{A}}$ denotes the empirical distribution induced by actions $\{\bar{a}_h^j\}_j$):

$$\| \mathbb{E}\left[\widehat{\mathbf{r}}_h^i \big| E_h^i, \mathcal{F}_h\right] - \mathbb{E}\left[\mathbf{F}(\widehat{\boldsymbol{\mu}}(\{\bar{a}_h^j\}_j))\right] \|_2$$

$$\le \| \mathbb{E}\left[\mathbf{F}(\widehat{\boldsymbol{\mu}}(\bar{a}_{h,exp}^i, \bar{a}_h^{-i}))\right] - \mathbb{E}\left[\mathbf{F}(\widehat{\boldsymbol{\mu}}(\{\bar{a}_h^j\}_j))\right] \|_2$$

$$\le \mathbb{E}\left[\|\mathbf{F}(\widehat{\boldsymbol{\mu}}(\bar{a}_{h,exp}^i, \bar{a}_h^{-i})) - \mathbf{F}(\widehat{\boldsymbol{\mu}}(\{\bar{a}_h^j\}_j))\|_2\right]$$

$$\le \frac{2L}{N}.$$

**Step 2: Bounding policy variations.** Once again, we first bound the variations in policies between agents.

$$\|\boldsymbol{\pi}_{h+1}^i - \boldsymbol{\pi}_{h+1}^j\|_2^2 = \|\Pi((1-\tau\eta_h)\boldsymbol{\pi}_h^i + \eta_h\widehat{\mathbf{r}}_h^i) - \Pi((1-\tau\eta_h)\boldsymbol{\pi}_h^j + \eta_h\widehat{\mathbf{r}}_h^j)\|_2^2$$

$$\le \|(1-\tau\eta_h)\boldsymbol{\pi}_h^i + \eta_h\widehat{\mathbf{r}}_h^i - (1-\tau\eta_h)\boldsymbol{\pi}_h^j + \eta_h\widehat{\mathbf{r}}_h^j\|_2^2$$

$$\le \|(1-\tau\eta_h)(\boldsymbol{\pi}_h^i - \boldsymbol{\pi}_h^j) + \eta_h(\widehat{\mathbf{r}}_h^i - \widehat{\mathbf{r}}_h^j)\|_2^2$$

$$= (1-\tau\eta_h)^2\|\boldsymbol{\pi}_h^i - \boldsymbol{\pi}_h^j\|_2^2 + \eta_h^2\|\widehat{\mathbf{r}}_h^i - \widehat{\mathbf{r}}_h^j\|_2^2 + 2(1-\tau\eta_h)\eta_h(\boldsymbol{\pi}_h^i - \boldsymbol{\pi}_h^j)^\top(\widehat{\mathbf{r}}_h^i - \widehat{\mathbf{r}}_h^j).$$

Unlike the expert feedback proof, the last term does not vanish in expectation.

$$\mathbb{E}\left[\|\boldsymbol{\pi}_{h+1}^i - \boldsymbol{\pi}_{h+1}^j\|_2^2 \big| \mathcal{F}_h\right] \le (1-\tau\eta_h)^2\|\boldsymbol{\pi}_h^i - \boldsymbol{\pi}_h^j\|_2^2 + \mathbb{E}\left[\eta_h^2\|\widehat{\mathbf{r}}_h^i - \widehat{\mathbf{r}}_h^j\|_2^2 \big| \mathcal{F}_h\right]$$

$$+ 2(1-\tau\eta_h)\eta_h(\boldsymbol{\pi}_h^i - \boldsymbol{\pi}_h^j)^\top \mathbb{E}\left[\widehat{\mathbf{r}}_h^i - \widehat{\mathbf{r}}_h^j \big| \mathcal{F}_h\right]$$

$$\le (1-\tau\eta_h)^2\|\boldsymbol{\pi}_h^i - \boldsymbol{\pi}_h^j\|_2^2 + \mathbb{E}\left[\eta_h^2\|\widehat{\mathbf{r}}_h^i - \widehat{\mathbf{r}}_h^j\|_2^2 \big| \mathcal{F}_h\right]$$

$$+ 2\eta_h(\boldsymbol{\pi}_h^i - \boldsymbol{\pi}_h^j)^\top\left[\mathbb{E}\left[\widehat{\mathbf{r}}_h^i \big| E_h^i, \mathcal{F}_h\right] + \mathbf{b}_h^i - \mathbb{E}\left[\widehat{\mathbf{r}}_h^j \Big| E_h^j, \mathcal{F}_h\right] - \mathbf{b}_h^j\right]$$

$$\le (1-\tau\eta_h)^2\|\boldsymbol{\pi}_h^i - \boldsymbol{\pi}_h^j\|_2^2 + \mathbb{E}\left[\eta_h^2\|\widehat{\mathbf{r}}_h^i - \widehat{\mathbf{r}}_h^j\|_2^2 \big| \mathcal{F}_h\right]$$

$$+ 2\eta_h(\boldsymbol{\pi}_h^i - \boldsymbol{\pi}_h^j)^\top\left[\mathbb{E}\left[\widehat{\mathbf{r}}_h^i \big| E_h^i, \mathcal{F}_h\right] - \mathbb{E}\left[\widehat{\mathbf{r}}_h^j \Big| E_h^j, \mathcal{F}_h\right]\right] + 8\eta_h\exp\left\{-\varepsilon T_h\right\},$$

where the last line follows from the bound on $\mathbf{b}_h^i$ in step 1. Furthermore, using Young's inequality, we obtain

$$\mathbb{E}\left[\|\boldsymbol{\pi}_{h+1}^i - \boldsymbol{\pi}_{h+1}^j\|_2^2 \big| \mathcal{F}_h\right] \le (1-\tau\eta_h)^2\|\boldsymbol{\pi}_h^i - \boldsymbol{\pi}_h^j\|_2^2 + \mathbb{E}\left[\eta_h^2\|\widehat{\mathbf{r}}_h^i - \widehat{\mathbf{r}}_h^j\|_2^2 \big| \mathcal{F}_h\right] + 8\eta_h\exp\left\{-\varepsilon T_h\right\}$$

$$+ \frac{\tau\eta_h}{2}\|\boldsymbol{\pi}_h^i - \boldsymbol{\pi}_h^j\|_2^2 + \tau^{-1}\left\|\mathbb{E}\left[\widehat{\mathbf{r}}_h^i \big| E_h^i, \mathcal{F}_h\right] - \mathbb{E}\left[\widehat{\mathbf{r}}_h^j \Big| E_h^j, \mathcal{F}_h\right]\right\|_2^2$$

$$\le (1-\tau\eta_h)^2\|\boldsymbol{\pi}_h^i - \boldsymbol{\pi}_h^j\|_2^2 + \mathbb{E}\left[\eta_h^2\|\widehat{\mathbf{r}}_h^i - \widehat{\mathbf{r}}_h^j\|_2^2 \big| \mathcal{F}_h\right] + 8\eta_h\exp\left\{-\varepsilon T_h\right\}$$

$$+ \frac{\tau\eta_h}{2}\|\boldsymbol{\pi}_h^i - \boldsymbol{\pi}_h^j\|_2^2 + \frac{4\tau^{-1}L^2}{N^2}.$$

Hence with the choice of $T_h = \varepsilon^{-1}\log(h+1)$ and noting that $\|\boldsymbol{\pi}_h^i - \boldsymbol{\pi}_h^j\|_2 \leq 2$, we obtain the recurrence

$$\mathbb{E}\left[\|\boldsymbol{\pi}_{h+1}^i - \boldsymbol{\pi}_{h+1}^j\|_2^2\right] \leq \left(1 - \frac{3/2}{h+1}\right)\mathbb{E}\left[\|\boldsymbol{\pi}_h^i - \boldsymbol{\pi}_h^j\|_2^2\right] + \frac{2\tau^{-2}K^3(\sigma^2+1) + 8\tau^{-1} + 4}{(h+1)^2}$$
$$+ \frac{4\tau^{-1}L^2}{N^2}.$$

Hence, by invoking the recurrence lemma (Lemma 2, with $c_0 = \frac{4\tau^{-1}L^2}{N^2}, c_1 = 2\tau^{-2}K^3(\sigma^2+1) + 8\tau^{-1} + 4, \gamma = 3/2, u_0 = 0$), we have

$$\mathbb{E}\left[\|\boldsymbol{\pi}_{h+1}^i - \boldsymbol{\pi}_{h+1}^j\|_2^2\right] \leq \frac{8\tau^{-2}K^3(\sigma^2+1) + 32\tau^{-1} + 16}{h+1} + \frac{8\tau^{-1}L^2}{N^2}.$$

**Step 3: Formulating the main recurrence.** Next, we formulate a recurrence for the main error term of interest, $\|\boldsymbol{\pi}_{t+1}^i - \boldsymbol{\pi}^*\|_2^2$, As before, we have (noting $\alpha_h := 1 - \tau\eta_h$):

$$\|\boldsymbol{\pi}_{h+1}^i - \boldsymbol{\pi}^*\|_2^2 = \|\Pi(\alpha_h\boldsymbol{\pi}_h^i + \eta_h\widehat{\mathbf{r}}_h^i) - \Pi(\alpha_h\boldsymbol{\pi}^* + \eta_h\mathbf{F}(\boldsymbol{\pi}^*))\|_2^2$$
$$\leq \|\alpha_h\boldsymbol{\pi}_h^i + \eta_h\widehat{\mathbf{r}}_h^i - \alpha_h\boldsymbol{\pi}^* - \eta_h\mathbf{F}(\boldsymbol{\pi}^*)\|_2^2$$
$$\leq \|\alpha_h\boldsymbol{\pi}_h^i + \eta_h\mathbf{F}(\boldsymbol{\pi}_h^i) - \alpha_h\boldsymbol{\pi}^* - \eta_h\mathbf{F}(\boldsymbol{\pi}^*) + \eta_h(\widehat{\mathbf{r}}_h^i - \mathbf{F}(\boldsymbol{\pi}_h^i))\|_2^2$$
$$= \eta_h^2\|\widehat{\mathbf{r}}_h^i - \mathbf{F}(\boldsymbol{\pi}_h^i)\|_2^2 + 2\eta_h(\alpha_h(\boldsymbol{\pi}_h^i - \boldsymbol{\pi}^*) + \eta_h(\mathbf{F}(\boldsymbol{\pi}_h^i) - \mathbf{F}(\boldsymbol{\pi}^*)))^\top(\widehat{\mathbf{r}}_h^i - \mathbf{F}(\boldsymbol{\pi}_h^i))$$
$$+ \|\alpha_h(\boldsymbol{\pi}_h^i - \boldsymbol{\pi}^*) + \eta_h(\mathbf{F}(\boldsymbol{\pi}_h^i) - \mathbf{F}(\boldsymbol{\pi}^*))\|_2^2$$
$$= \eta_h^2\|\widehat{\mathbf{r}}_h^i - \mathbf{F}(\boldsymbol{\pi}_h^i)\|_2^2 + 2\eta_h^2(\mathbf{F}(\boldsymbol{\pi}_h^i) - \mathbf{F}(\boldsymbol{\pi}^*)))^\top(\widehat{\mathbf{r}}_h^i - \mathbf{F}(\boldsymbol{\pi}_h^i)) + 2\eta_h\alpha_h(\boldsymbol{\pi}_h^i - \boldsymbol{\pi}^*)^\top(\widehat{\mathbf{r}}_h^i - \mathbf{F}(\boldsymbol{\pi}_h^i))$$
$$+ \|\alpha_h(\boldsymbol{\pi}_h^i - \boldsymbol{\pi}^*) + \eta_h(\mathbf{F}(\boldsymbol{\pi}_h^i) - \mathbf{F}(\boldsymbol{\pi}^*))\|_2^2$$
$$\leq \underbrace{\eta_h^2\|\widehat{\mathbf{r}}_h^i - \mathbf{F}(\boldsymbol{\pi}_h^i)\|_2^2 + 2\eta_h^2(\mathbf{F}(\boldsymbol{\pi}_h^i) - \mathbf{F}(\boldsymbol{\pi}^*)))^\top(\widehat{\mathbf{r}}_h^i - \mathbf{F}(\boldsymbol{\pi}_h^i))}_{(a)}$$
$$+ \underbrace{2\eta_h\alpha_h(\boldsymbol{\pi}_h^i - \boldsymbol{\pi}^*)^\top(\widehat{\mathbf{r}}_h^i - \mathbf{F}(\boldsymbol{\pi}_h^i))}_{(b)} + \underbrace{\|\alpha_h(\boldsymbol{\pi}_h^i - \boldsymbol{\pi}^*) + \eta_h(\mathbf{F}(\boldsymbol{\pi}_h^i) - \mathbf{F}(\boldsymbol{\pi}^*))\|_2^2}_{(c)}$$

Once again, we will need to bound the three terms above. For the term $(a)$, we bound in expectation as before:

$$\mathbb{E}\left[(a)\right] \leq 2\eta_t^2 K^3(\sigma^2 + 1).$$

Likewise, it still holds for the term $(c)$ that

$$(c) = \|\alpha_h(\boldsymbol{\pi}_h^i - \boldsymbol{\pi}^*) + \eta_h(\mathbf{F}(\boldsymbol{\pi}_h^i) - \mathbf{F}(\boldsymbol{\pi}^*))\|_2^2$$
$$= \|(\boldsymbol{\pi}_h^i - \boldsymbol{\pi}^*) + \eta_h(\mathbf{F}(\boldsymbol{\pi}_h^i) - \tau\boldsymbol{\pi}_h^i - \mathbf{F}(\boldsymbol{\pi}^*) + \tau\boldsymbol{\pi}^*)\|_2^2$$
$$\leq \left(1 - 2(\lambda + \tau)\eta_h + (L + \tau)^2\eta_h^2\right)\|\boldsymbol{\pi}_h^i - \boldsymbol{\pi}^*\|_2^2.$$

However, this time, the exploration parameter $\varepsilon$ will cause additional bias in the term $(b)$. Define $\widetilde{\mathbf{r}}_h^i = \mathbb{E}[\widehat{\mathbf{r}}_h^i|E_h^i, \mathcal{F}_h]$.

$$(b) = 2\eta_h\alpha_h(\boldsymbol{\pi}_h^i - \boldsymbol{\pi}^*)^\top(\widehat{\mathbf{r}}_h^i - \mathbf{F}(\boldsymbol{\pi}_h^i))$$
$$= 2\eta_h\alpha_h(\boldsymbol{\pi}_h^i - \boldsymbol{\pi}^*)^\top(\widehat{\mathbf{r}}_h^i - \widetilde{\mathbf{r}}_h^i) + 2\eta_h\alpha_h(\boldsymbol{\pi}_h^i - \boldsymbol{\pi}^*)^\top(\widetilde{\mathbf{r}}_h^i - \mathbf{F}(\bar{\boldsymbol{\mu}}_h))$$
$$+ 2\eta_h\alpha_h(\boldsymbol{\pi}_h^i - \boldsymbol{\pi}^*)^\top(\mathbf{F}(\bar{\boldsymbol{\mu}}_h) - \mathbf{F}(\boldsymbol{\pi}_h^i))$$
$$\leq 2\eta_h\alpha_h\left(\frac{\lambda}{4}\|\boldsymbol{\pi}_h^i - \boldsymbol{\pi}^*\|_2^2 + \frac{1}{\lambda}\|\widetilde{\mathbf{r}}_h^i - \mathbf{F}(\bar{\boldsymbol{\mu}}_h)\|_2^2\right) + 2\eta_h\alpha_h\left(\frac{\lambda}{4}\|\boldsymbol{\pi}_h^i - \boldsymbol{\pi}^*\|_2^2 + \frac{1}{\lambda}\|\mathbf{F}(\bar{\boldsymbol{\mu}}_h) - \mathbf{F}(\boldsymbol{\pi}_h^i)\|_2^2\right) +$$
$$2\eta_h\alpha_h(\boldsymbol{\pi}_h^i - \boldsymbol{\pi}^*)^\top(\widehat{\mathbf{r}}_h^i - \widetilde{\mathbf{r}}_h^i)$$
$$\leq 2\eta_h\frac{\lambda}{2}\|\boldsymbol{\pi}_h^i - \boldsymbol{\pi}^*\|_2^2 + \frac{2\eta_h}{\lambda}\|\widetilde{\mathbf{r}}_h^i - \mathbf{F}(\bar{\boldsymbol{\mu}}_h)\|_2^2 + \frac{2\eta_h}{\lambda}\|\mathbf{F}(\bar{\boldsymbol{\mu}}_h) - \mathbf{F}(\boldsymbol{\pi}_h^i)\|_2^2$$
$$+ 2\eta_h\alpha_h(\boldsymbol{\pi}_h^i - \boldsymbol{\pi}^*)^\top(\widehat{\mathbf{r}}_h^i - \widetilde{\mathbf{r}}_h^i),$$

and similarly, if $\lambda = 0$, we have

$$
\begin{aligned}
(b) =& 2\eta_h\alpha_h(\boldsymbol{\pi}_h^i - \boldsymbol{\pi}^*)^\top(\widehat{\mathbf{r}}_h^i - \mathbf{F}(\boldsymbol{\pi}_h^i)) \\
=& 2\eta_h\alpha_h(\boldsymbol{\pi}_h^i - \boldsymbol{\pi}^*)^\top(\widehat{\mathbf{r}}_h^i - \widetilde{\mathbf{r}}_h^i) + 2\eta_h\alpha_h(\boldsymbol{\pi}_h^i - \boldsymbol{\pi}^*)^\top(\widetilde{\mathbf{r}}_h^i - \mathbf{F}(\bar{\boldsymbol{\mu}}_h)) \\
& + 2\eta_h\alpha_h(\boldsymbol{\pi}_h^i - \boldsymbol{\pi}^*)^\top(\mathbf{F}(\bar{\boldsymbol{\mu}}_h) - \mathbf{F}(\boldsymbol{\pi}_h^i)) \\
\leq& 2\eta_h\alpha_h\left(\frac{\tau\delta}{2}\|\boldsymbol{\pi}_h^i - \boldsymbol{\pi}^*\|_2^2 + \frac{1}{2\tau\delta}\|\widetilde{\mathbf{r}}_h^i - \mathbf{F}(\bar{\boldsymbol{\mu}}_h)\|_2^2\right) + 2\eta_h\alpha_h\left(\frac{\tau\delta}{2}\|\boldsymbol{\pi}_h^i - \boldsymbol{\pi}^*\|_2^2 + \frac{1}{2\tau\delta}\|\mathbf{F}(\bar{\boldsymbol{\mu}}_h) - \mathbf{F}(\boldsymbol{\pi}_h^i)\|_2^2\right) + \\
& 2\eta_h\alpha_h(\boldsymbol{\pi}_h^i - \boldsymbol{\pi}^*)^\top(\widehat{\mathbf{r}}_h^i - \widetilde{\mathbf{r}}_h^i) \\
\leq& 2\eta_h\tau\delta\|\boldsymbol{\pi}_h^i - \boldsymbol{\pi}^*\|_2^2 + \frac{\eta_h}{\tau\delta}\|\widetilde{\mathbf{r}}_h^i - \mathbf{F}(\bar{\boldsymbol{\mu}}_h)\|_2^2 + \frac{\eta_h}{\tau\delta}\|\mathbf{F}(\bar{\boldsymbol{\mu}}_h) - \mathbf{F}(\boldsymbol{\pi}_h^i)\|_2^2 \\
& + 2\eta_h\alpha_h(\boldsymbol{\pi}_h^i - \boldsymbol{\pi}^*)^\top(\widehat{\mathbf{r}}_h^i - \widetilde{\mathbf{r}}_h^i).
\end{aligned}
$$

Denote $\boldsymbol{\pi}_{unif} := \frac{1}{K}\mathbf{1}_K$. The remaining error terms we can bound by (using the auxiliary coupling random actions $\bar{a}_h^j$ from Step 1):

$$
\begin{aligned}
& |\,\mathbb{E}\left[2\eta_h\alpha_h(\boldsymbol{\pi}_h^i - \boldsymbol{\pi}^*)^\top(\widehat{\mathbf{r}}_h^i - \widetilde{\mathbf{r}}_h^i)|\mathcal{F}_h\right]| \\
& \leq |\,\mathbb{E}\left[2\eta_h\alpha_h(\boldsymbol{\pi}_h^i - \boldsymbol{\pi}^*)^\top(\widehat{\mathbf{r}}_h^i - \widetilde{\mathbf{r}}_h^i)|E_h^i,\mathcal{F}_h\right]\mathbb{P}(E_h^i) + \mathbb{E}\left[2\eta_h\alpha_h(\boldsymbol{\pi}_h^i - \boldsymbol{\pi}^*)^\top(\widehat{\mathbf{r}}_h^i - \widetilde{\mathbf{r}}_h^i)|\overline{E_h^i},\mathcal{F}_h\right]\mathbb{P}(\overline{E_h^i})| \\
& \leq \frac{2\tau^{-1}}{h+1}\,\mathbb{E}\left[\|\boldsymbol{\pi}_h^i - \boldsymbol{\pi}^*\|_2\|\widehat{\mathbf{r}}_h^i - \widetilde{\mathbf{r}}_h^i\|_2|\overline{E_h^i},\mathcal{F}_h\right]\mathbb{P}(\overline{E_h^i}) \\
& \leq \frac{8K^{3/2}\tau^{-1}\sqrt{1+\sigma^2}}{(h+1)^2}.
\end{aligned}
$$

$$
\begin{aligned}
\mathbb{E}[\|\widetilde{\mathbf{r}}_h^i - \mathbf{F}(\bar{\boldsymbol{\mu}}_t)\|_2^2|\mathcal{F}_h] \leq& 2\,\mathbb{E}\left[\|\widetilde{\mathbf{r}}_h^i - \mathbf{F}(\widehat{\boldsymbol{\mu}}(\bar{a}_{h,exp}^i, \bar{a}_h^{-i}))\|_2^2|\mathcal{F}_h\right] + 2\,\mathbb{E}\left[\|\mathbf{F}(\widehat{\boldsymbol{\mu}}(\bar{a}_{h,exp}^i, \bar{a}_h^{-i})) - \mathbf{F}(\bar{\boldsymbol{\mu}}_h)\|_2^2|\mathcal{F}_h\right] \\
\leq& \frac{8L^2}{N} + 2\,\mathbb{E}\left[\|\mathbf{F}(\widehat{\boldsymbol{\mu}}(\bar{a}_{h,exp}^i, \bar{a}_h^{-i})) - \mathbf{F}(\bar{\boldsymbol{\mu}}_h)\|_2^2|\mathcal{F}_h\right] \\
\leq& \frac{8L^2}{N} + 2L^2\,\mathbb{E}\left[\|\widehat{\boldsymbol{\mu}}(\bar{a}_{exp}^i, \bar{a}^{-i}) - \frac{1}{N}\sum_{j=1}^N\boldsymbol{\pi}_h^j\|_2^2|\mathcal{F}_t\right] \\
\leq& \frac{8L^2}{N} + 4L^2\,\mathbb{E}\left[\|\widehat{\boldsymbol{\mu}}(\bar{a}_{exp}^i, \bar{a}^{-i}) - \frac{1}{N}\sum_{j\neq i}(\varepsilon\boldsymbol{\pi}_{unif} + (1-\varepsilon)\boldsymbol{\pi}_h^j) - \frac{\boldsymbol{\pi}_{unif}}{N}\|_2^2|\mathcal{F}_t\right] \\
& + 4L^2\,\mathbb{E}\left[\|\varepsilon\frac{1}{N}\sum_{j\neq i}(\varepsilon\boldsymbol{\pi}_h^j - \boldsymbol{\pi}_{unif}) + \frac{\boldsymbol{\pi}_h^i - \boldsymbol{\pi}_{unif}}{N}\|_2^2|\mathcal{F}_t\right] \\
\leq& \frac{8L^2}{N} + \frac{8L^2}{N} + 4L^2(2\varepsilon^2 + \frac{4}{N}) \\
=& \frac{64L^2}{N} + 8L^2\varepsilon^2,
\end{aligned}
$$

$$
\|\mathbf{F}(\bar{\boldsymbol{\mu}}_h) - \mathbf{F}(\boldsymbol{\pi}_h^i)\|_2^2 \leq L^2\|\bar{\boldsymbol{\mu}}_h - \boldsymbol{\pi}_h^i\|_2^2 = L^2 e_h^i
$$

**Step 4: Main result.** Once again, for the strongly monotone case $\lambda > 0$ we have that

$$
\begin{aligned}
u_{h+1}^i \leq& \frac{2\tau^{-2}K^3(1+\sigma^2) + 8\tau^{-2}(L+\tau)^2 + 8K^{3/2}\tau^{-1}\sqrt{1+\sigma^2}}{(h+1)^2} \\
& + \frac{128\tau^{-1}\lambda^{-1}L^2N^{-1} + 16\tau^{-1}\lambda^{-1}L^2\varepsilon^2}{h+1} + \frac{2\tau^{-1}\lambda^{-1}L^2}{h+1}\,\mathbb{E}\left[e_h^i\right] \\
& + \left(1 - \frac{2\tau^{-1}(\lambda/2 + \tau)}{h+1}\right)u_h^i,
\end{aligned}
$$

leading to the recurrence

$$u_{h+1}^i \leq \frac{2\tau^{-2}K^3(\sigma^2+1)(1+8\tau^{-1}\lambda^{-1}L^2)+32\tau^{-2}\lambda^{-1}(2\tau^{-1}+1)+8\tau^{-2}(L+\tau)^2+8K^{3/2}\tau^{-1}\sqrt{1+\sigma^2}}{(h+1)^2}$$
$$+\frac{16\tau^{-1}\lambda^{-1}L^2(8N^{-1}+\varepsilon^2)+16\tau^{-2}\lambda^{-1}L^2N^{-2}}{h+1}$$
$$+\left(1-\frac{2\tau^{-1}(\lambda/2+\tau)}{h+1}\right)u_h^i.$$

In the case of a monotone operator, we have

$$u_{h+1}^i \leq \frac{2\tau^{-2}K^3(\sigma^2+1)+4\tau^{-2}(L+\tau)^2+8K^{3/2}\tau^{-1}\sqrt{1+\sigma^2}}{(h+1)^2}$$
$$+\frac{64\tau^{-2}\delta^{-1}L^2N^{-1}+8\tau^{-2}\delta^{-1}L^2\varepsilon^2}{h+1}+\frac{\tau^{-2}\delta^{-1}L^2}{(h+1)}\mathbb{E}\left[e_h^i\right]$$
$$+\left(1-\frac{2(1-\delta)}{h+1}\right)u_h^i,$$

which we can bound by (choosing $\delta = 1/4$):

$$u_{t+1}^i \leq \frac{(1+\sigma^2)K^3\tau^{-2}(2+32\tau^{-2}L^2)+4\tau^{-2}(L+\tau)^2+8K^{3/2}\tau^{-1}\sqrt{1+\sigma^2}+64\tau^{-2}L^2(2\tau^{-1}+1)}{(h+1)^2}$$
$$+\frac{256\tau^{-2}L^2N^{-1}+32\tau^{-2}L^2\varepsilon^2+32\tau^{-3}L^2N^{-2}}{h+1}$$
$$+\left(1-\frac{3/2}{h+1}\right)u_h^i.$$

We use Lemma 2 to obtain the statement of the theorem, once again choosing $\gamma = 3/2$.

The bound in the statement of the theorem in the main body of the paper follows from the fact that the lengths of the exploration epochs scale with $T_h = \mathcal{O}(\varepsilon^{-1}\log(h+1)) = \widetilde{\mathcal{O}}(\varepsilon^{-1})$.  $\square$

Finally, we note that while the dependence on $K$, the number of actions, is not discussed in the main body paper, as expected the algorithm for bandit feedback has a worse dependency on the number of actions. This is as expected due to the fact that (i) the importance sampling estimator increases variance on payoff estimators by a factor of $K$, and (2) in other words, a factor of $\mathcal{O}(K)$ is introduced in order to explore all actions.

## G    DETAILS OF EXPERIMENTS

**Hardware and compute.** All experiments were run on single core of an AMD EPYC 7742 CPU, a single experiment with 1000 independent agents and 100000 iterations takes roughly 1 hour. No GPU compute was used. All relevant code (withholding location details to preserve anonymity) has been shared in the supplementary material.

**Setup details.** For all experiments, we use parameters $\tau, \epsilon$ as implied by Corollaries 1,2. Projections to the probability simplex were implemented using the algorithm in Duchi et al. (2008). The particular Python implementation of the TRPA-Full and TRPA-Bandit operators are provided in the supplementary material.

### G.1    PROBLEM GENERATION DETAILS

We provide further details on how we generate/simulate the SMFG problems.

**Linear payoffs.** We generate a payoff map

$$\mathbf{F}_{lin}(\boldsymbol{\mu}) := (\mathbf{S}+\mathbf{X})\boldsymbol{\mu}+\mathbf{b}$$

for some $\mathbf{S} \in \mathbb{S}_{++}^{K \times K}$ and $\mathbf{X}$ anti-symmetric matrix, which makes monotone $\mathbf{F}_{lin}$ by Example 6. We randomly sample $\mathbf{S}$ from a Wishart distribution (which has support contained in positive definite matrices), generate $\mathbf{X}$ by computing $\frac{\mathbf{U}-\mathbf{U}^\top}{2}$ for a random matrix $\mathbf{U}$ with entries sampled uniformly from $[0, 1]$ and $\mathbf{b}$ having entries uniformly sampled from $[0, 1]$.

**Payoffs with KL potential.** Next, based on Example 5, we construct the following payoff operator $\mathbf{F}_{KL}$ for some reference distribution $\boldsymbol{\mu}_{\text{ref}} \in \Delta_{\mathcal{A}}$:

$$\Phi_{KL}(\boldsymbol{\mu}) := \mathrm{D}_{KL}(\gamma\boldsymbol{\mu} + (1-\gamma)\boldsymbol{\mu}_{\text{ref}} || \boldsymbol{\mu}_{\text{ref}})$$
$$\mathbf{F}_{KL}(\boldsymbol{\mu}) := \nabla\Phi_{KL}(\boldsymbol{\mu})$$
$$\mathbf{F}_{KL}(\boldsymbol{\mu}, a) = \gamma \log\left(\frac{\gamma\boldsymbol{\mu}(a) + (1-\gamma)\boldsymbol{\mu}_{\text{ref}}(a)}{\boldsymbol{\mu}_{\text{ref}}(a)}\right) + \gamma$$

Note that as $\Phi_{KL}$ is convex, $\mathbf{F}_{KL}$ is monotone. In our experiments, we use $\gamma = 0.1$, and we generate $\boldsymbol{\mu}_{\text{ref}}$ by sampling $K$ uniform random variables in $[0, 1]$ and normalizing.

**Beach bar process.** Following the example given in Perrin et al. (2020), we use the action set $\mathcal{A} = \{1, \ldots, K\}$ for potential locations at the beach and assume a bar is located at $x_{bar} := \lfloor \frac{K}{2} \rfloor$. Taking into the proximity to the bar and the occupancy measure over actions (i.e., the crowdedness of locations at the beach), the payoff map is given by:

$$\mathbf{F}_{bb}(\boldsymbol{\mu}, a) = 1 - \frac{|a - x_{bar}|}{K} - \alpha \log(1 + \boldsymbol{\mu}(a)).$$

Note that by Example 1, the above payoff map is monotone. We use $\alpha = 1$ for our experiments.

## G.2 Learning Curves - Full Feedback

We provide the learning curves under full feedback for various choices of the number of agents $N \in \{20, 100, 1000\}$. The errors in terms of maximum exploitability and distance to MF-NE are presented in Figure 3 and Figure 4 respectively.

As expected, the games with larger number of players $N$ converge to better approximate NE in the sense that the final maximum exploitability is smaller at convergence. Furthermore, in most cases the exploitability converges slightly slower with more agents, also supporting the theoretical finding that there is a dependence on $N$. As before, the exploitability curves have oscillations at later stages of the training, even though they remain upper bounded as foreseen the theoretical results. This does not contradict our results as long as for larger $N$, the upper bound on the oscillations us smaller. The confidence intervals plotted in figures support a high-probability upper bound on the maximum exploitability as one would expect.

## G.3 Learning Curves - Bandit Feedback

We provide the learning curves under bandit feedback for various choices of the number of agents $N \in \{50, 100, 1000\}$. The errors in terms of maximum exploitability and distance to MF-NE are presented in Figure 5 and Figure 6 respectively.

As in the case of full feedback, the curves converge to smaller values as $N$ increases. Furthermore, one straightforward observation is that the variance at early stages of learning is much higher than in the full feedback case. This can be due to the added variance of the importance sampling estimator constructed through exploration epochs. As exploration occurs in shorter duration at early epochs, the variance between agent policies will be high as well, explaining the initial increase in exploitability in certain toy experiments in Figure 5.

Furthermore, comparing the observations for bandit feedback (Figure 5) and full feedback cases (Figure 3), we empirically confirm that learning take more iterations in the bandit case. This is likely due to the fact that exploration occurs probabilistically, inducing additional variance in the policy updates that increases with $N$ and incorporates an additional logarithmic term in the theoretical bounds. However, the number of exploration epochs in the bandit case is comparable to the number of time steps in the full feedback case.

Finally, we also emphasize the fact that in earlier stages of training with bandit feedback, the cases with $N = 1000$ had much higher exploitability and $\ell_2$ distance to MF-NE at earlier time steps. This

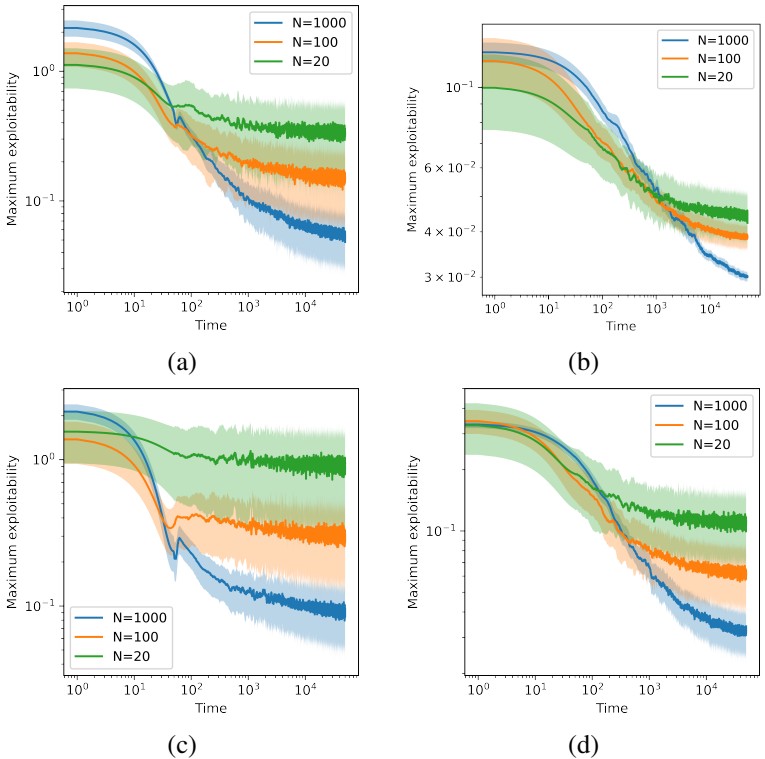

Figure 3: The (smoothed) maximum exploitability $\max_{i \in \mathcal{N}} \phi^i(\{\boldsymbol{\pi}^j\}_{j=1}^N)$ among $N$ agents throughout learning with full feedback for three different $N$, on the problems (a) linear payoffs, (b) exponentially decreasing payoffs, (c) payoffs with KL potential and (d) the beach bar payoffs.

is due to the fact that policy updates occur with larger intervals in between when $N$ is large, as can be seen in Algorithm 1. This can be explicitly observed in Figure 6, as the policies of agents are constant in between policy updates. However, at later stages as the time-dependent term in the bound on epxloitability in Corollary 2 disappears, we observe that the experiments for larger $N$ converge to a better policy (i.e., one with lower exploitability) than the cases with smaller $N$ as the theory suggests.

Finally, comparing Figures 5,6, we see that in certain experiments for some $N$ despite having a lower exploitability we observe a greater $\ell_2$ distance to MF-NE. This likely due to fact that the non-vanishing bias term in exploitability and $\ell_2$ distance have differing dependence on problem parameters such as $L, \lambda, K$. Therefore, for instance for the KL potential payoffs, we observe a greater mean $\ell_2$ distance to MF-NE but a smaller exploitability for $N = 1000$.

### G.4 COMPARISON WITH MULTIPLICATIVE WEIGHT UPDATES

The SMFG problem of interactive learning without communications and bandit feedback has not been studied in the literature previously, and there is no alternative algorithm with theoretical guarantees to the best of our knowledge in this setting. However, in Figure 7 we provide comparison with a heuristic extension of the OMD algorithm proposed by Pérolat et al. (2022) which was analyzed in the case of an infinite agent game. In general, independent learning with $N$ agents introduces unique challenges that cause the algorithm to cycle or diverge.

### G.5 EXPERIMENTS ON TRAFFIC CONGESTION

**UTD19 and closed-loop sensors.** The UTD19 dataset contains measurements by closed-loop sensors which report the fraction of the time a particular section of the route remains occupied (i.e., a car is located in between sensors placed on the sides). The data consists of measurements every 5 minutes,

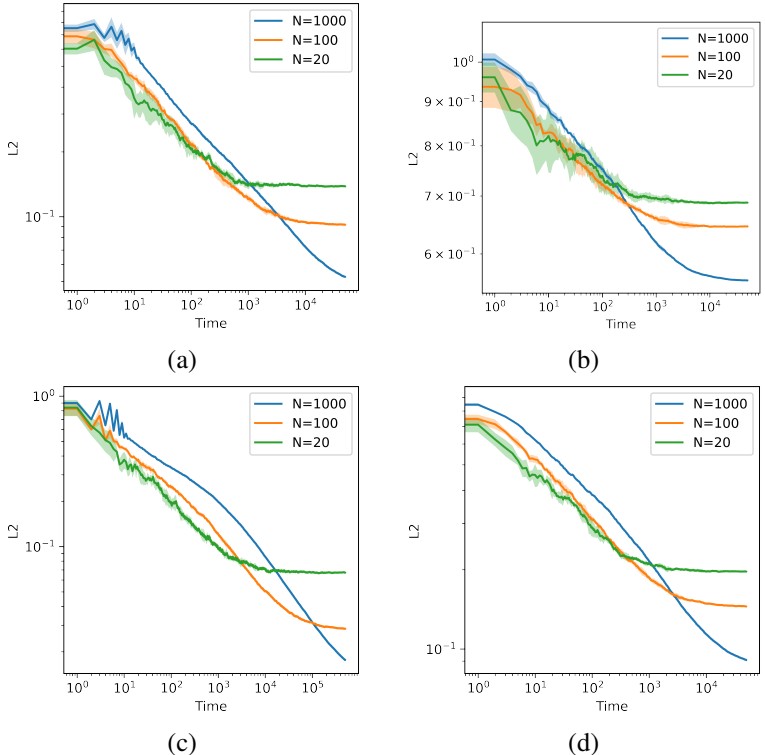

Figure 4: The mean $\ell_2$ distance to MF-NE given by $\frac{1}{N}\sum_{i\in\mathcal{N}}\|\boldsymbol{\pi}^i - \boldsymbol{\pi}^*\|_2$ with $N$ agents throughout learning with full feedback for three different $N$, on the problems (a) linear payoffs, (b) exponentially decreasing payoffs, (c) payoffs with KL potential and (d) the beach bar payoffs.

from various sensors across 41 European cities. The dataset contains 2 weeks of data collected by sensors placed around Zurich, the locations of sensors imposed on the map in Figure 8.

**Payoff models.** We fit kernelized ridge regression models to model the flow as a function of occupancy using the UTD19 dataset. We use an RBF kernel and a regularization of $\alpha = 1.0$ for all models. We compute a proxy for the travel velocity using the flow and occupancy measurements on each route, and a scaling factor $c_{\text{dist}}$ due to varying lengths of each route, leading to the estimated travel time

$$T_{travel} = c_{\text{dist}}\frac{\text{flow}}{\text{occupancy}}.$$

We use $-T_{travel}$ as the reward for each agent. The flows collected from the dataset and the corresponding fitted models are presented in Figure 9 and Figure 10.

In Figure 9, we emphasize that as expected, total flow peaks at a value of occupancy in $(0, 1)$, as congestion effects likely become dominant for high occupancy resulting in lower flow values. However, as expected, the travel speed (Figure 10) seems to be monotonically decreasing as a function of occupancy, also as expected. This suggests that up to minor deviations due to the kernelize regression model, the rewards $-T_{travel}$ decrease monotonically as occupancy increases. Hence Figure 10 provides empirical evidence of a monotone payoff operator.

We emphasize that our model of congestion is greatly simplified and does not take into account network effects in congestion due to interactions of various connected routes in the city. For instance, it is likely that correlations exist between the travel speed as a function of congestion of the three routes in reality. While more realistic simulations could be performed using the UTD19 dataset resulting in a more realistic evaluation of our methods, more intricate simulations of traffic remain outside the scope of this paper and we leave the evaluation of TRPA-Bandit in such realistic scenarios as future work.

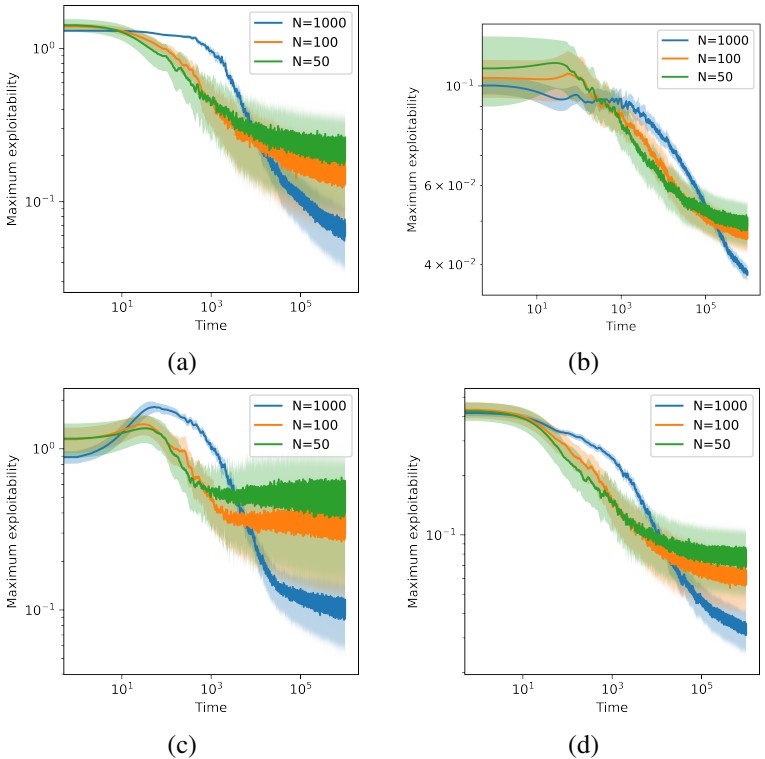

Figure 5: The (smoothed) maximum exploitability $\max_{i \in \mathcal{N}} \phi^i(\{\boldsymbol{\pi}^j\}_{j=1}^N)$ among $N$ agents throughout learning with bandit feedback for three different $N$, on the problems (a) linear payoffs, (b) exponentially decreasing payoffs, (c) payoffs with KL potential and (d) the beach bar payoffs.

## G.6 EXPERIMENTS ON NETWORK ACCESS

For the Tor network access experiment, we randomly chose 5 entry guard servers (the complete list available publicly The Tor Network) in various geographic locations, among the servers that have the longest recorded uptime. To simulate access to each server, we ping each 5 consecutive times and average the delays to compute the cost. As expected, due to varying bandwidths/computational power, each server has different sensitivities to load in terms of delay, as Figure 11 demonstrates for two. For the two servers plotted here for instance, we note that while on has waiting times somewhat sensitive to occupancy, the other is much less sensitive to additional agents accessing it, at least when simulating 100 agents. Hence, strong monotonicity is likely to not hold.

We use parameters $\tau = 0.01, \varepsilon = 0.3$ for the experiments in this section. The arbitrary choice is due to the fact that in the presence of external agents in the game that do not use TRPA-Bandit (in this case, thousands of other users accessing the Tor network), the theoretically optimal parameters implied by Corollary 2 can not be used. While more realistic simulations that are also closer to the theory could be run by keeping $N$ larger and simulating a Tor access rather than simple pings, we refrain from this in order to minimize the footprint of our experiments on the Tor network.

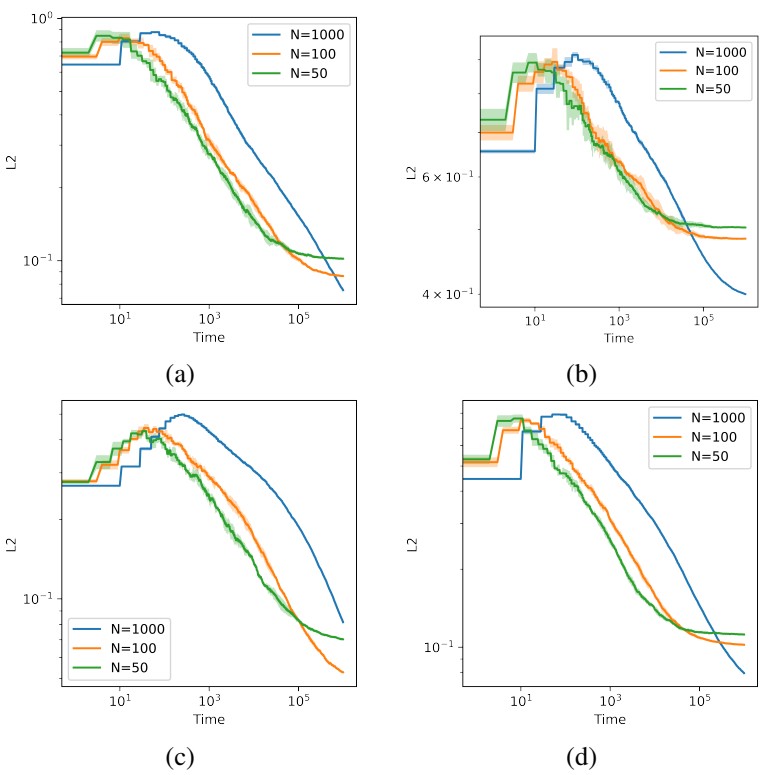

Figure 6: The mean $\ell_2$ distance to MF-NE given by $\frac{1}{N} \sum_{i \in \mathcal{N}} \|\boldsymbol{\pi}^i - \boldsymbol{\pi}^*\|_2$ with $N$ agents throughout learning with bandit feedback for three different $N$, on the problems (a) linear payoffs, (b) exponentially decreasing payoffs, (c) payoffs with KL potential and (d) the beach bar payoffs.

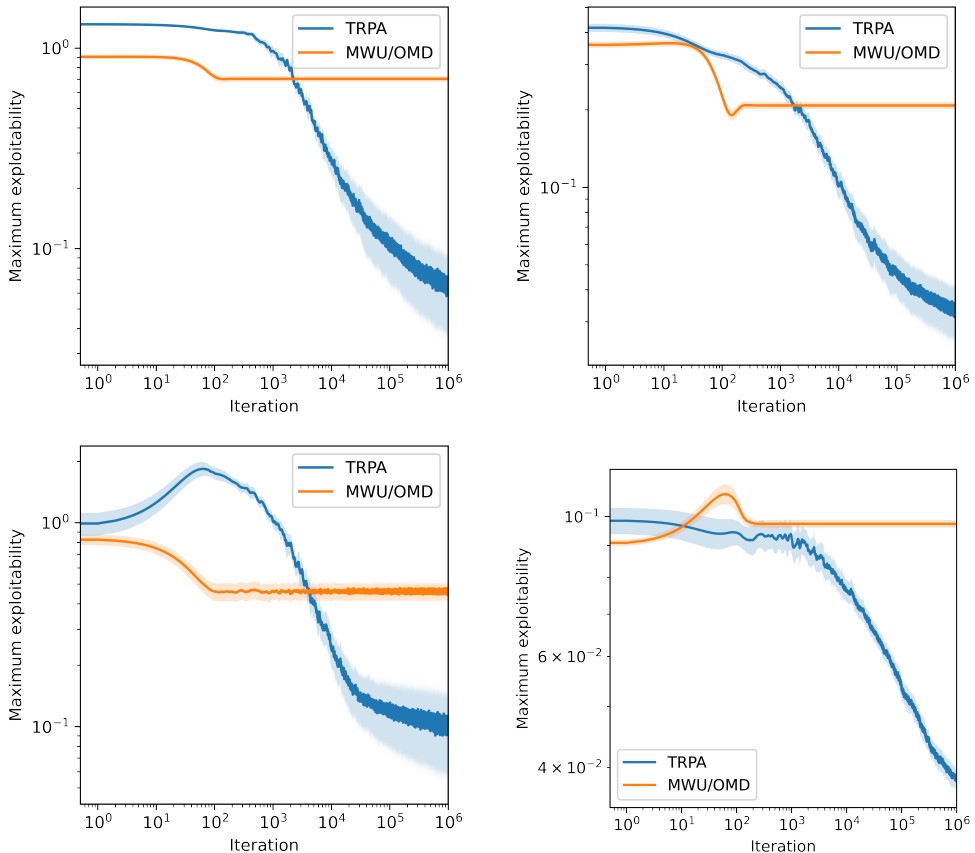

Figure 7: Comparison of maximum exploitability between our TRPA-Bandit and multiplicative weight updates (equivalently OMD of Pérolat et al. (2022)) in benchmarks LINEAR, BB, KL, EXP (left to right). $N = 1000$ agents, averaged over 10 experiments.

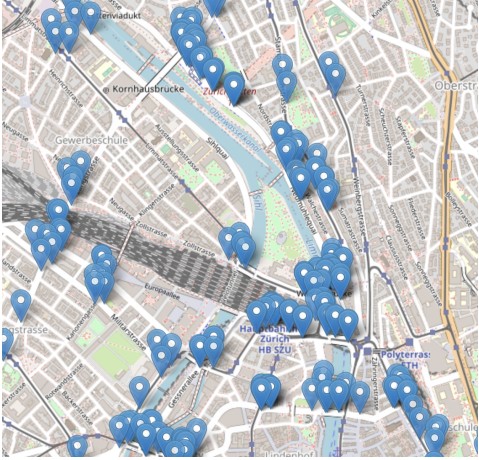

Figure 8: A portion of the sensors placed in the UTD19 dataset Loder et al. (2019) within the Zurich center. Map generated using OpenStreetMap contributors (2017).

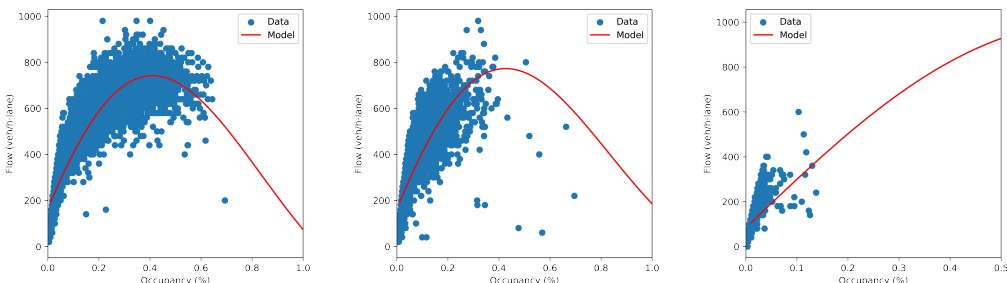

Figure 9: Data and fitted models of occupancy vs. flow on three different urban roads. The red line indicates the fitted model predictions.

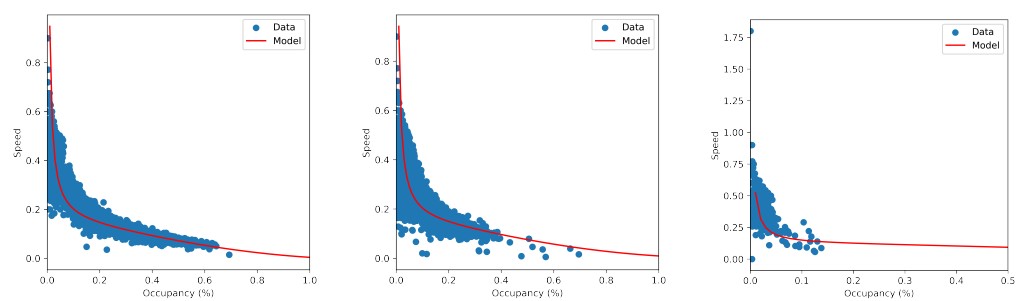

Figure 10: Data and fitted models of occupancy vs. speed (scaled) on three different urban roads. The red line indicates the fitted model predictions.

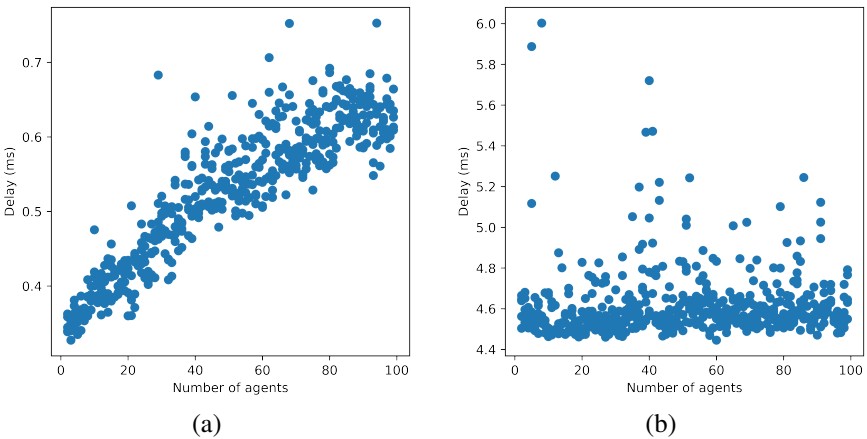

(a)                                    (b)

Figure 11: Access times (in terms of ping delays) of two Tor entry guard servers in terms of number of agents accessing simultaneously.

