# OpenReview forum: "Stateless Mean-Field Games: A Framework for Independent Learning with Large Populations"
_ICLR.cc/2024/Conference — Submitted to ICLR 2024_

### Official Review · Reviewer_qCUR · 2023-10-28

**Soundness:** 2 fair
**Presentation:** 3 good
**Contribution:** 2 fair
**Rating:** 5
**Confidence:** 2

**Summary:**

This paper focuses on the so-called stateless mean-field games. It considers the finite-agent cases, and the authors include two information feedback settings: full-feedback and bandit feedback. For each of the settings, the authors propose efficient algorithms to learn the Nash equilibrium of the game. The convergence rates of the algorithms and the numerical validations are provided.

**Strengths:**

This paper focuses on a new finite-agent setting for mean-field games. The work proposes independent learning algorithms with the performance guarantees to learn the Nash equilibrium. In addition, the numerical results are provided to verify the theoretical findings.

**Weaknesses:**

After reading the paper, some of my concerns are not addressed:

1. The results in Theorems 1 and 2 discount the effect of mean-field approximation and need more explanations. In my personal understanding, the mean-field approximation is implemented to avoid the dependency of the number of agents $N$ in sample complexity (I understand that in the finite-player setting, the complexity can scale as $\log N$ due to for example the union bound in concentrations). However, Theorems 1 and 2 suggest that $T$ should at least grow as $poly(N)$ for a meaningful result. This implies that the results in this paper cannot be extended to the setting $N=\infty$ considered in the previous papers, e.g., [1]. This may suggest that the algorithms proposed in this paper is not suitable for the large population problem. In addition, the justification for this polynomial dependency by the comparison with [2] below Corollary 2 may be unreasonable, since mean-field approximation is not considered in [2].

2. The bias term $O(1/\sqrt{N})$ needs more discussion. Although the authors explain that this potentially comes from the independent learning setting, such a term does not appear in other independent learning settings, e.g., the potential games and two-player zero-sum games. I do not fully understand why there is a bias term in the problem setting considered in this paper.

3. Based on the above points, I am not sure whether these concerns arise from the loose analysis of the upper bounds or from the problem formulation itself. It will be helpful to derive the lower bound for these two concerns.

[1] Guo X, Hu A, Xu R, et al. Learning mean-field games[J]. Advances in neural information processing systems, 2019, 32.

[2] Lugosi G, Mehrabian A. Multiplayer bandits without observing collision information[J]. Mathematics of Operations Research, 2022, 47(2): 1247-1265.

**Questions:**

The questions are the same as the weakness part.

---

> ### Author Response · Authors · 2023-11-20
>
> We thank reviewer qCUR for their comments.
>
> ## Question 1: (regarding dependence on $N$)
>
> We emphasize that (Guo et al. 2019) pointed out by the reviewer and many others in mean-field game literature rely on a very strong oracle/assumption: their algorithms are **centralized** and can **prescribe any behavior to an infinite population** and observe the resulting dynamics while learning (as in a “population generative model”). In a more realistic setting, however, the population distribution will be induced by the behavior of learners interacting with the game over many rounds leading to complicated interactions. We consider the much more challenging and realistic setting where $N$ decentralized learners repeatedly play the game while locally observing interactions. Namely, unlike past work, our setting is:
> - **Finite-agent**: learning occurs by playing the finite-agent game, not by centrally simulating an infinite player approximation,
> - **Independent/decentralized:** agents can not communicate and their observations are local,
> - **Limited feedback:** agents can only partially observe rewards leading to a more realistic learning model.
>
> All in all, the setting of SMFG is much more challenging compared to many works in MFG literature. Our analysis incurs a dependence on $poly(N)$, however, we solve a more challenging and realistic learning problem.
>
> We agree with the reviewer that the setting of multi-player MAB as in Lugosi et al., 2022 can not be directly compared to our setting: in fact, the collision model employed by Lugosi et al., 2022 and others in MMAB literature has a **much more restrictive structure than our setting**. The SMFG incorporates a wide range of complicated payoff functions (including the much simpler binary collision rewards) hence the SMFG problem is much more complicated than the MMAB setting. The reference to Lugosi et al., 2022 served as an analogy for how independent/decentralized learning can introduce a dependence on $N$ even for the much more restrictive setting of collision models.
>
> Finally, while the scaling of the number of iterations to $\mathcal{O}(N^{3/2})$ is indeed a polynomial dependence in the decentralized learning with bandit feedback, it might not be prohibitively large for the large class of payoff functions considered under decentralized learning. The MFG approximation is introduced to tractably solve a difficult problem, but the decentralized learning model nevertheless can introduce a dependence on $N$. Once again, while not a directly comparable result, we point out for example the polynomial $\mathcal{O}(N^{12})$ dependence in the decentralized case for congestion games [5].
>
> ## Question 2: (regarding non-vanishing bias term)
>
> The bias term is standard for mean-field games [1,2,3] and can be considered as trading off a small amount of bias/exploitability (when $N$ is large) for a **tractable approximate solution of an otherwise intractable problem**. Our work can be seen as proof that this approximation can be performed without explicitly constructing the $N\rightarrow\infty$ approximate problem by a centralized algorithm but rather independently by participants of the real-world $N$-player game.
>
> We also note that SMFG solves a different class of problems compared to congestion games or potential games as we show in Section B.2 that SMFG in general does not admit a potential structure. While in the highly structured and restrictive settings of potential games and two-player zero-sum games, it is possible to solve the problem exactly, even for symmetric games the task of finding exact NE can be computationally intractable [4].
>
> ## Question 3: (regarding lower bounds)
>
> Developing such lower bounds for multi-agent systems is an active topic of research; such results are sparse even for structured games such as potential games. One result we are aware of is https://arxiv.org/pdf/2303.12287.pdf which provides hardness results of independent learning for Markov games. More relevantly, the works https://arxiv.org/pdf/1511.00785.pdf and https://arxiv.org/pdf/1606.04550.pdf provably show a lower bound on the query complexity of computing $\varepsilon$-NE growing exponentially in the number of players $N$, demonstrating that the mean-field approximation of our SMFG framework for large is relevant. Such lower bound results for SMFG, while being of fundamental interest, are out of the scope of this paper.
>
> ### References:
>
> [1] Cui K, Koeppl H. Approximately solving mean field games via entropy-regularized deep reinforcement learning. AISTATS 2021
>
> [2] Anahtarci B, Kariksiz CD, Saldi N. Q-learning in regularized mean-field games. Dynamic Games and Applications. 2023.
>
> [3] Yardim B, Cayci S, Geist M, He N. Policy mirror ascent for efficient and independent learning in mean field games. ICML 2023.
>
> [4] A Survey of PPAD-Completeness for Computing Nash Equilibria, Paul W. Goldberg, 2011
>
> [5] Cui, et al. "Learning in congestion games with bandit feedback." NeurIPS 2022.

---

> ### Comment · Reviewer_qCUR · 2023-11-22
> **Thanks for the Response**
>
> Thanks the authors for the detailed response.
>
> For the concern regarding the dependence on $N$, it is not clear to me why the setting in Lugosi et al., 2022 has a much more restrictive structure. The reason is that they do not implement the mean-field approximation, and the number of arms in their setting grows as number of agents grows.
>
> In the seminal work of Lasry and Lions (2007), the mean-field game is formulated to study a “continuum limit”. The previous works also achieve this goal by deriving learning bounds that are independent of the number of agents. I understand that the $poly(N)$ appears since the authors consider the independent learning setting. But I am not sure whether this comes from the suboptimality of the algorithm or the problem setting itself.
>
> I will keep my scores at this stage due to these reasons.

---

> ### Author Response · Authors · 2023-11-23
>
> We thank the reviewer for their response.
>
> ## Comparison with Lugosi et al.:
>
> Regarding the work of Lugosi et al., the reason **we consider their model restrictive is due to the collision model**. In their case, if multiple agents choose a particular arm $k$, the reward obtained is a constant zero, and if an agent alone plays the arm $k$, it obtains a constant reward $r_k$ associated with the arm. In fact, it’s easy to formulate an NE in the setting of Lugosi et al. in the centralized case: we can simply place the $N$ agents to the arms with the highest rewards. There is no need for a mean-field approximation to make the problem tractable, their game model is simple enough.
>
> In our case, the reward map $\mathbf{F}$ is an **unknown nonlinear function of action occupancies** (see beginning of Section 2.1). For instance, the payoff of arm $k$ can arbitrarily depend on how many agents played another arm $k’$ as well as how many agents played $k$. Therefore our game model is much more complicated to solve, even in the centralized learning case it is a computational challenge to formulate an NE. The mean-field approximation makes our problem tractable.
>
> ## Regarding MFG and the work of Lasry and Lions
>
> Regarding the seminal work of Lasry and Lions and subsequent research in mean-field RL, we point out that past MFG works solve the continuum limit (the MFG) directly without consideration of the real-world $N$ player game. This is considerably easier.
>
> In other words, there is no dependence on $N$ in past works because there is **no concept of agents**, rather they are simplified to a distribution $\mu$. In this regard past work is **theoretically convenient** but **not realistic**: in the real world, regardless of the mean-field approximation the game played is always an $N < \infty$ player game. This is precisely the contribution of our paper: we could solve MF-VI (page 6) directly easily to obtain a bound similar to past work without a notion of independent agents, but instead for the first time we prove that efficient independent learning with $N < \infty$ agents is possible.
>
> We refer the reviewer to the table provided above to reviewer qRij for a comparison with the past line of work that has followed the seminal paper by Lasry and Lions.
>
> ## Sublinear dependence on $N$
>
> We would also like to point out that $N$ appears **sublinearly** in our bounds in Theorem 1 and 2, which is much more tractable compared to an arbitrary polynomial dependence. **Even in games such as potential games or congestion games**, a $poly(N)$ cost is incurred when solving the game, which we reference in our original response. The $\log N$ dependence of Lugosi et al. is possible in the much simple collision model for the rewards, but not in the case when the game has more complicated reward dynamics such as ours.

---

### Official Review · Reviewer_rayu · 2023-10-31

**Soundness:** 3 good
**Presentation:** 3 good
**Contribution:** 3 good
**Rating:** 8
**Confidence:** 2

**Summary:**

To deal with the challenges of space complexity of multiagent RL, they propose stateless mean field games. They are able to use sampling algorithms to find approximate Nash equilibrium sufficiently quick.

**Strengths:**

They propose a (seemingly) novel way to deal with massive state space. The paper is well written.

**Weaknesses:**

It seems they find an approximate nash equilibrium rather than an exact one. They also impose lipschitz and monotonous payoffs as their restrictions. However they make a good case that these assumptions are not too limiting.

**Questions:**

Is finding a exact nash equilibrium possible with this methodology?

---

> ### Author Response · Authors · 2023-11-20
>
> We thank reviewer rayu for their comments.
>
> As the reviewer points out, our algorithms produce an approximate NE, however, this bias term is standard for mean-field games [1,2,3] and can be considered as trading off a small amount of bias/exploitability (when $N$ is large) for a **tractable approximate solution of an otherwise intractable problem**. In general, as $N$ gets large (as in many real-world problems) the bias will vanish as our bounds explicitly show.
>
> Regarding the question of whether an exact NE can be computed, intractability results exist even in the case the reward function is exactly known and learning is not decentralized [4].
>
>
> [1] Cui K, Koeppl H. Approximately solving mean field games via entropy-regularized deep reinforcement learning. AISTATS 2021
>
> [2] Anahtarci B, Kariksiz CD, Saldi N. Q-learning in regularized mean-field games. Dynamic Games and Applications. 2023.
>
> [3] Yardim B, Cayci S, Geist M, He N. Policy mirror ascent for efficient and independent learning in mean field games. ICML 2023.
>
> [4] A Survey of PPAD-Completeness for Computing Nash Equilibria, Paul W. Goldberg, 2011

---

### Official Review · Reviewer_qRij · 2023-11-01

**Soundness:** 3 good
**Presentation:** 4 excellent
**Contribution:** 2 fair
**Rating:** 3
**Confidence:** 3

**Summary:**

This paper studies a generalization of congestion games where cost functions need only be monotone (rather than monotone increasing), which they term a "stateless mean-field game". Since the game can be written as a VI with a monotone operator, they propose the use of (what is effectively) L2-regularized gradient descent as an uncoupled game dynamic to recover a solution. The experiments of the paper implement the proposed L2-regularized GD and claim an empirical improvement over unregularized GD.

**Strengths:**

The Tor network access experiment (which I read to be a live real-world experiment using real Tor latencies) is a very creative and interesting experiment design that I have not seen before. The empirical results that online mirror descent does not perform as well as L2 regularized online mirror descent is not surprising or completely new (see below) but real-world experiments comparing the efficacy of different game dynamics is always interesting and valuable. The paper is also generally well-written.

**Weaknesses:**

It seems that there might be some significant overlap between the theoretical claims of the paper and what is already known about congestion games, variational inequalities, and game dynamics. First, approaching congestion games with variational inequalities instead of explicitly exploiting game potentials (e.g. via best response-dynamics) is somewhat standard; the fact that such approaches still work when one relaxes cost functions from being monotone increasing to just monotone is a fairly direct implication---although this is assuming that my understanding that SMFG = Congestion Game but with both monotone increasing/decreasing costs is correct.
One of the main technical contributions claimed by the paper is that, instead of running (what is effectively) gradient descent, the paper proposes to run (what is effectively) L2-regularized gradient descent. It seems there were three claimed benefits: 1) it makes the problem strongly monotone (to avoid needing the extragradient method), 2) it helps with stochasticity (convergence despite noise seems to just follow from normal stochastic approximation though and nothing to do with the regularization in particular), and 3) it allows them to implement the algorithm in a decentralized way across the players. The 3rd point seems to be the primary emphasis. However, I don't believe this is a new observation: a standard way of learning equilibria in online learnable (such as monotone) setups is no-regret dynamics. As a side note, Theorem 1 should probably state the dependence on $K$ instead of hiding it as a constant---it seems like the correct dependence should be $\log(K)$ at least in the full information setting (e.g. using exponentiated gradient descent).

**Questions:**

See above.

---

> ### Author Response · Authors · 2023-11-20
>
> We thank reviewer qRij for their comments.
>
> ## SMFG vs. congestion games:
>
> As Appendix B.2 demonstrates, in general, SMFGs are not congestion (or more generally potential) games. While combinations of VI and no-regret dynamics are thoroughly analyzed in structured games such as potential games or for the subclass congestion games, for our class of symmetric games independent learning guarantees have been largely absent. Our work is thus more directly comparable to mean-field games literature.
>
> One parallel can be drawn with non-atomic congestion games which also approximate the infinite-agent limit (Roughgarden et al., 2004), however, we are not aware of any algorithms in this setting where learning is performed by $N$ agents in a decentralized manner. One of the main contributions of the work is utilizing the infinite-player approximation to understand independent learning in the more realistic finite-player setting with explicit finite-sample and finite-agent bounds.
>
> In case the reviewer has particular works for comparison with our setting, we would be happy to provide further differences.
>
> ## Independent learning and no-regret dynamics:
>
> While no-regret dynamics are analyzed to provide learning guarantees for various classes of games, they can in general cycle or fail to converge. Despite many centralized learning results with strong oracles (e.g. Perrin et al., 2020 and others mentioned in the paper), independent learning guarantees in MFG literature have so far been understudied. We fill the gap in MFG literature by showing that certain algorithms (combined with particular exploration strategies) can be used to approximately solve large-scale games without communication.
>
> Finally, we note that due to the particular regularization induced by $\tau$, our algorithms are not direct adaptations of no-regret methods. Instead, a particular regularization term depending on population size is required for optimal rates. A similar case can be made for the bandit feedback case: to obtain optimal rates, a particular exploration probability $\varepsilon$ depending on $N$ must be used.
>
> ## Summary of theoretical contributions in MFG literature:
>
> We provide the table below to compare the theoretical results of our paper to existing work in mean-field games literature.
>
> |   |$N$ agent learning| True NE  |  Bandit Feedback | Sample Complexity Guarantees  | Independent Learning  | Sample Efficient IL  |
> |---|---|---|---|---|---|---|
> |  Guo et al., 2019 |  No |  No |  No | Yes  |  No |  - |
> |  Perrin et al., 2020 |  No | Yes  | No  |  No |  No |  - |
> | Cui et al., 2021  | No  | No  |  No | No  |  No |  - |
> |  Anahtarci et al., 2023 | No  |  No |  No |  Yes |  No |  - |
> | Parise et al., 2023  | No  |  Yes | No  | No  |  No | -  |
> | Yardim et al., 2023 | Yes   | No  |  No | Yes  |  Yes |  No  |
> | **Ours (SMFG)**  | **Yes**  |  **Yes** |  **Yes** | **Yes**  | **Yes**  | **Yes**  |
>
> ## Dependence on $K$:
>
> Finally, we thank the reviewer for pointing out the dependence on $K$ in our bounds. Since the primary focus of our complexity results is to establish the dependencies on the number of agents $N$ and the number of iterations $T$, we move the explicit dependency on $K$ to the appendix due to space considerations. We will make the dependence explicit in the main paper as the reviewer suggested.
>
> ### References:
>
> Parise F, Ozdaglar A. Graphon games: A statistical framework for network games and interventions[J]. Econometrica, 2023, 91(1): 191-225.
>
> Cui K, Koeppl H. Approximately solving mean field games via entropy-regularized deep reinforcement learning. In International Conference on Artificial Intelligence and Statistics 2021 Mar 18 (pp. 1909-1917). PMLR.
>
> Anahtarci B, Kariksiz CD, Saldi N. Q-learning in regularized mean-field games. Dynamic Games and Applications. 2023 Mar;13(1):89-117.
>
> Perrin S, Pérolat J, Laurière M, Geist M, Elie R, Pietquin O. Fictitious play for mean field games: Continuous time analysis and applications. Advances in Neural Information Processing Systems. 2020;33:13199-213.
>
> Guo X, Hu A, Xu R, Zhang J. Learning mean-field games. Advances in neural information processing systems. 2019;32.
>
> Yardim B, Cayci S, Geist M, He N. Policy mirror ascent for efficient and independent learning in mean field games. In International Conference on Machine Learning 2023 Jul 3 (pp. 39722-39754). PMLR.
>
> Perolat J, Perrin S, Elie R, Laurière M, Piliouras G, Geist M, Tuyls K, Pietquin O. Scaling up mean field games with online mirror descent. arXiv preprint arXiv:2103.00623. 2021 Feb 28.
>
> Roughgarden, T., & Tardos, É. (2004). Bounding the inefficiency of equilibria in nonatomic congestion games. Games and economic behavior, 47(2), 389-403.

---

> ### Comment · Reviewer_qRij · 2023-11-23
> **Reviewer response**
>
> Thanks for the response! I appreciate the pointer to Appendix B.2---I understand that SMFGs are not exactly congestion games, but it does seem to be the case (based on B.2 and the author's response) that the only additional generality afforded by SMFGs is allowing for monotonically decreasing costs.
> Regarding no-regret dynamics, I'm not entirely sure what the authors mean by no-regret dynamics will tend to cycle---perhaps the authors are conflating no-regret dynamics with e.g. fictitious play or similar dynamics?
> That the regularization term depends on the number of players is not something unique to the algorithm of this paper: even in the usual no-regret dynamics, the regularization that one applies to each player also depends on population size, since regularization depends on the number of iterations that one needs to run game dynamics which itself depends on the number of players.
> I'm still concerned regarding the significance of the submission's results and maintain my score.

---

### Meta-Review · Area_Chair_9kpV · 2023-12-07

**Metareview:**

This paper studies a generalization of congestion games where cost functions need only be monotone (rather than monotone increasing), which they term a "stateless mean-field game". Since the game can be written as a VI with a monotone operator, they propose the use of (what is effectively) L2-regularized gradient descent as an uncoupled game dynamic to recover a solution. The experiments of the paper implement the proposed L2-regularized GD and claim an empirical improvement over unregularized GD.
The major concern is the novelty: there is a significant overlap between the theoretical claims of the paper and what is already known about congestion games, variational inequalities, and game dynamics.
The AC agrees and thus recommends rejection.

**Justification For Why Not Higher Score:**

There is a significant overlap between the theoretical claims of the paper and what is already known about congestion games, variational inequalities, and game dynamics.

**Justification For Why Not Lower Score:**

N/A

---

### Decision · Program_Chairs · 2024-01-16

Reject